# Dynamic spreading of chromatin-mediated gene silencing and reactivation between neighboring genes in single cells

**Sarah Lensch[1], Michael H Herschl[2], Connor H Ludwig[1], Joydeb Sinha[3], Michaela M Hinks[1], Adi Mukund[4], Taihei Fujimori[1], Lacramioara Bintu[1]\***

[1]Department of Bioengineering, Stanford University, Stanford, United States; [2]University of California, Berkeley—University of California, San Francisco Graduate Program in Bioengineering, Berkeley, United States; [3]Department of Chemical and Systems Biology, Stanford University, Stanford, United States; [4]Biophysics Program, Stanford University, Stanford, United States

**Abstract** In mammalian cells genes that are in close proximity can be transcriptionally coupled: silencing or activating one gene can affect its neighbors. Understanding these dynamics is important for natural processes, such as heterochromatin spreading during development and aging, and when designing synthetic gene regulation circuits. Here, we systematically dissect this process in single cells by recruiting and releasing repressive chromatin regulators at dual-gene synthetic reporters, and measuring how fast gene silencing and reactivation spread as a function of intergenic distance and configuration of insulator elements. We find that silencing by KRAB, associated with histone methylation, spreads between two genes within hours, with a time delay that increases with distance. This fast KRAB-mediated spreading is not blocked by the classical cHS4 insulators. Silencing by histone deacetylase HDAC4 of the upstream gene can also facilitate background silencing of the downstream gene by PRC2, but with a days-long delay that does not change with distance. This slower silencing can sometimes be stopped by insulators. Gene reactivation of neighboring genes is also coupled, with strong promoters and insulators determining the order of reactivation. Our data can be described by a model of multi-gene regulation that builds upon previous knowledge of heterochromatin spreading, where both gene silencing and gene reactivation can act at a distance, allowing for coordinated dynamics via chromatin regulator recruitment.

**\*For correspondence:**
lbintu@stanford.edu

**Competing interest:** The authors declare that no competing interests exist.

## Editor's evaluation

This study describes a novel approach to investigate how the transcriptional repressors KRAB and HDAC4 repress gene expression, how repression spreads and the role of insulator elements in blocking the spread of repression and in reactivation of repressed genes. Despite some inherent limitations resulting from the use of artificial reporters compared to previous genomic studies, it addresses the question of repression in a rather novel manner adding a dimension of time and single cell resolution. The results allow modeling of the coordinated repression or activation of closely linked genes and should be of wide interest to researchers interested in chromatin, gene expression and synthetic biology. The authors have made a thorough revision of the paper and have addressed all of the major issues raised by the referees. They have better discussed their results in the context of previous studies and they have added new data with EZH2 inhibitors that support a role for PRC2 in HDAC4-mediated repression.

## Introduction

Eukaryotic gene expression is regulated by chromatin regulators (CRs) and the histone and DNA modifications they read, write, and remove (*Bannister and Kouzarides, 2011*; *Zhang et al., 2016*). Chromatin-mediated gene regulation is crucial in development, aging, and disease (*Jambhekar et al., 2019*; *Völker-Albert et al., 2020*; *Zhao and Shilatifard, 2019*), with classical examples of X-chromosome inactivation and the spatial-temporal control of Hox genes (*Soshnikova and Duboule, 2009*; *Payer, 2017*). It is also important in synthetic biology, where precise control of gene expression is necessary for probing gene regulatory networks with CRISPRi type screens, for better understanding mechanisms of epigenetic regulation such as cellular reprogramming, and for therapeutic applications such as gene therapy (*Keung et al., 2015*; *Thakore et al., 2016*; *Lienert et al., 2014*). Due to the limit in length of DNA constructs that can be successfully delivered and integrated into cells (*Lukashev and Zamyatnin, 2016*; *Liu et al., 2017*), multiple genes are often placed close together, such as an antibiotic resistance selective marker next to a gene of interest. In these synthetic systems, a common method of regulating gene expression is through site-specific recruitment of CRs (*Margolin et al., 1994*; *Deuschle et al., 1995*; *Gilbert et al., 2013*). CRs modulate gene expression with varying kinetics and can establish long-term epigenetic memory through positive feedback mechanisms (*Bintu et al., 2016*; *Ayyanathan et al., 2003*; *Zhang et al., 2015*; *Uckelmann and Davidovich, 2021*), which enable spreading of epigenetic effects beyond the target locus and lead to undesirable changes in gene expression, sometimes implicated in aging and cancer (*Sedivy et al., 2008*; *Wang et al., 2015*). This phenomenon of spreading of epigenetic effects was discovered in *Drosophila* and was originally coined position effect variegation (*Muller, 1930*; *Wang et al., 2014a*). The mechanism of action has since been elucidated to involve readers and writers of histone modifications forming a feedback loop that causes modifications to spread (*Elgin and Reuter, 2013*), and has also been shown to occur in mammalian cells in vivo (*Groner et al., 2012*). However, neither the temporal dynamics nor the spatial extent of this process are well characterized. Here, we seek to understand the effects of intergenic distance and insulators on the dynamics of spreading of gene silencing and activation between two neighboring genes. This understanding will be useful for building more robust synthetic systems and epigenetic therapies.

To understand the effects of intergenic distance and insulators on spreading, we studied the dynamics of gene silencing and reactivation after recruitment and release, respectively, of different types of CRs to a dual-gene reporter (*Figure 1A&B* and *Figure 1—figure supplement 1*). This reporter consists of two fluorescent genes separated by increasing DNA distances or insulator elements derived from the chicken hypersensitivity site 4 (cHS4) (*Chung et al., 1993*; *Chung et al., 1997*; *Guye et al., 2013*). The cHS4 insulator is a commonly used insulator in synthetic biology in mammalian cells (*Recillas-Targa et al., 2004*), as it has been shown act as a barrier against heterochromatin (*Recillas-Targa et al., 2002*; *Burgess-Beusse et al., 2002*), thus preventing transgene silencing in many cell lines including CHO and K562 (*Recillas-Targa et al., 2004*; *Pikaart et al., 1998*; *Walters et al., 1999*; *Mutskov et al., 2002*; *Zhang et al., 2017*). At this reporter, we recruited two different chromatin regulators: Kruppel associated box (KRAB) and histone deacetylase 4 (HDAC4).

The KRAB repressive domain from zinc finger 10 is commonly used in synthetic biology applications (*Nakamura et al., 2021*), is one of the strongest repressor domains in human cells (*Margolin et al., 1994*; *Witzgall et al., 1994*; *Cong et al., 2012*; *Tycko et al., 2020*), and is associated with spreading of heterochromatin and epigenetic memory through positive feedback mechanisms (*Bintu et al., 2016*; *Ayyanathan et al., 2003*; *Groner et al., 2010*; *Amabile et al., 2016*; *O'Geen et al., 2019*). KRAB-mediated gene silencing operates through recruitment of cofactors, including KAP1, HP1, and SETDB1, that read and write the repressive histone modification, histone 3 lysine 9 trimethylation (H3K9me3), creating a positive feedback loop that establishes stable gene silencing (*Ayyanathan et al., 2003*). Through this type of feedback mediated by the ability of KAP1 to recruit HP1, spreading of epigenetic effects beyond the target locus can affect nearby genes in a distance dependent manner (*Groner et al., 2010*). Targeted recruitment of the KRAB domain from ZNF10 has been shown to repress gene expression, and lead to loss in histone 3 acetylation and gain of H3K9me3 across several tens to hundreds of kilobases around the target gene depending on the method and duration of recruitment (*Groner et al., 2010*; *Amabile et al., 2016*; *Feng et al., 2020*). However, the spatial-temporal dynamics of these effects and the capacity of the commonly used cHS4 insulator to influence KRAB-mediated spreading of silencing have not been systematically characterized. In

addition, the dynamics of reactivation at neighboring genes after removal of KRAB has also not been well characterized in a synthetic system with variable distance or insulators. Understanding the drivers of reactivation could also be important for diseases, where developmentally silenced genes reactivate (*Das and Chadwick, 2021*).

We wanted to compare the effects of KRAB with those of another fast silencer that is not associated with histone methylation positive feedback, and instead is only associated with removal of acetylation, such as HDAC4 We have previously shown that HDAC4 recruitment leads to fast gene silencing, comparable to that of KRAB (*Bintu et al., 2016*). Therefore, we expected silencing of the targeted gene to occur at a similar rate as with KRAB, while differences in the downstream gene silencing would be due to the lack of positive feedback with HDAC4 recruitment.

We investigated how silencing by these two rapid-acting chromatin regulators with and without positive feedback mechanisms, KRAB and HDAC4, respectively, affects gene expression of neighboring genes when separated by either distance or the cHS4 insulator, as well as how the dynamics of reactivation are affected after removal of each chromatin regulator. We used single-cell time-lapse microscopy and flow cytometry to monitor gene expression during and after KRAB or HDAC4 recruitment to the upstream gene. We found that transcriptional silencing of the upstream gene by recruitment of either KRAB or HDAC4 affects the downstream gene even when separated by up to 5 kb of distance or with the cHS4 insulator and its core region. For KRAB, the time required for spreading to occur is short and largely distance-dependent. For HDAC4, however, the delay time between silencing of the two genes is longer, does not increase with intergenic distance, and can be influenced by insulators. Reactivation also spreads between the two genes with a delay that is distance-dependent, and is affected by promoters and insulators. We can summarize our findings with a simple kinetic model that describes the dynamics of silencing and reactivation as a competition between: (1) the silencing rates associated with each repressive CR, and (2) the activation rates associated with strong promoters and insulators, where both of these rates decrease with genomic distance. Our results show that targeted transcriptional silencing affects neighboring genes with different dynamics and provide insight for designing novel synthetic systems in mammalian cells.

## Results

### Spreading of transcriptional silencing between genes

To study the spreading of transcriptional silencing and chromatin modifications, we site-specifically integrated reporters consisting of two neighboring fluorescent reporter genes (mCitrine and mCherry) with each gene driven by a constitutive promoter (*Figure 1A&B*, *Figure 1—figure supplement 1*). Both fluorescent proteins are fused to H2B; this approach increases the signal to noise ratio due to nuclear localization and is commonly used for studying chromatin dynamics (*Hadjantonakis and Papaioannou, 2004*; *Stewart et al., 2009*; *Fraser et al., 2005*; *Kanda et al., 1998*). In each reporter cell line, we also stably integrated a recruitable rTetR fused to either KRAB or HDAC4. The addition or removal of doxycycline (dox) induces the recruitment or release, respectively, of the rTetR-CR upstream of the first reporter gene, enabling precise control over the duration of recruitment (*Figure 1A&B*). To probe the effect of intergenic distance on spreading dynamics, we separated the two fluorescent genes by either no spacer (NS) (*Figure 1A*) or 5 kb lambda phage DNA (5 kb) (*Figure 1B*), which was used as a neutral spacer (*Chung et al., 1997*; *Walters et al., 1999*; *Belozerov et al., 2003*; *Majumder and Cai, 2003*; *Hily et al., 2009*; *Di Simone et al., 2001*). To mitigate variability due to genomic position, we site-specifically integrated our reporters in three cell types: in CHO-K1 (Chinese Hamster Ovarian) at the phiC31 integration site on the multi-integrase human artificial chromosome (MI-HAC) (*Yamaguchi et al., 2011*) and in human K562 and HEK293T at the AAVS1 safe harbor locus on chromosome 19 (*Figure 1—figure supplement 1*; *Hockemeyer et al., 2011*).

To investigate the spreading of transcriptional silencing, we first recruited KRAB upstream of the mCitrine gene for 5 days in CHO-K1, K562, and HEK293T cell lines with the addition of dox (*Figure 1A&B*). Using flow cytometry to measure fluorescence intensity, we observed silencing of both the upstream mCitrine gene and the downstream mCherry gene in the NS (*Figure 1C&D*) and 5 kb (*Figure 1E&F*, *Figure 1—figure supplement 2A*) reporter lines for the cell types tested. Tuning the probability of KRAB recruitment at the reporter by varying dox concentration, we found that spreading of silencing still happens, in both the NS and the 5 kb reporters, and the percentage of cells with both genes silenced increases with dox concentration (*Figure 1—figure supplement 3*). These

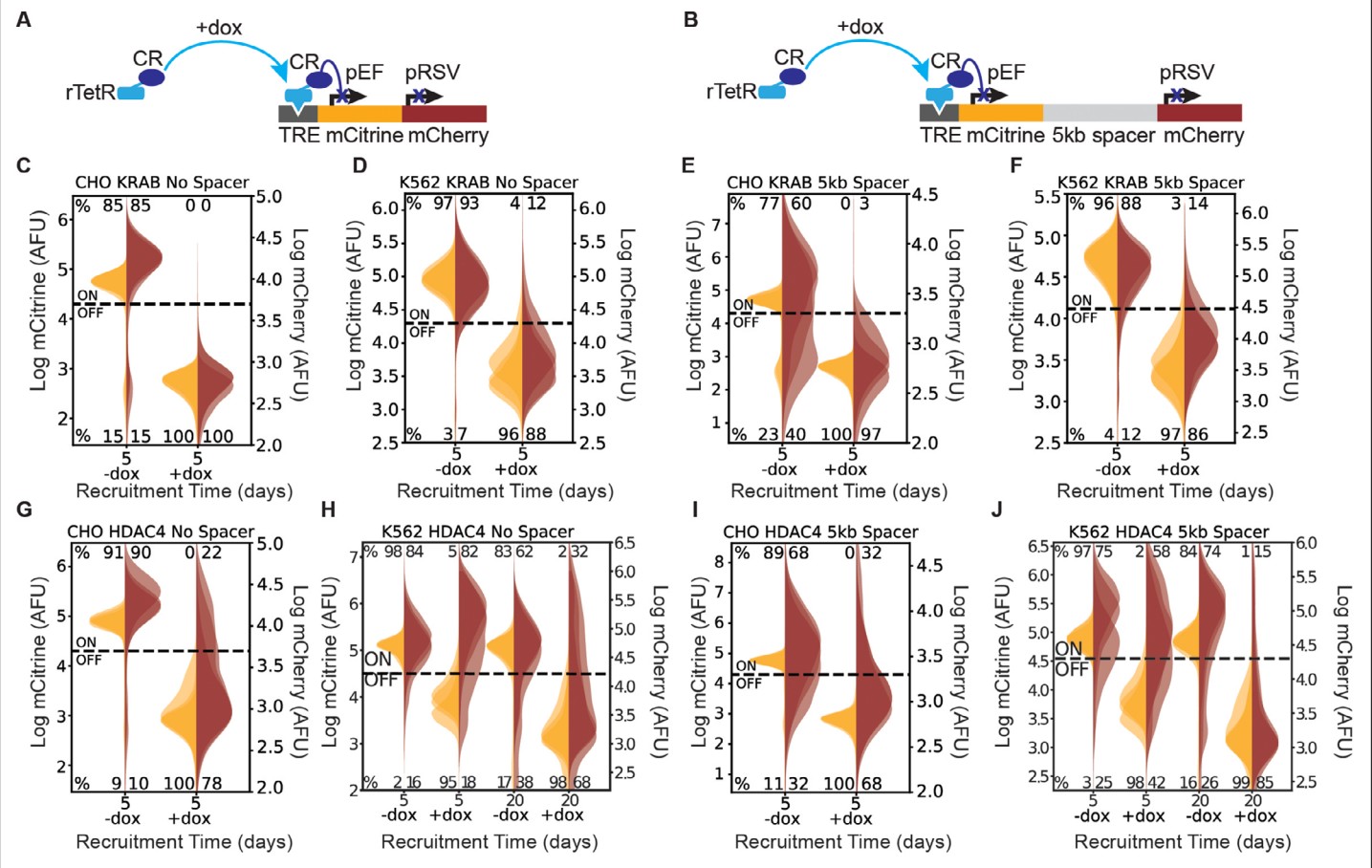

**Figure 1.** Recruitment of chromatin regulators to synthetic dual-fluorescent reporters results in transcriptional silencing of both genes. (**A, B**) Recruitment of a chromatin regulator (dark blue oval) via addition of dox allows for the binding of the rTetR-CR fusion to TRE (Tet Responsive Element, dark grey), upstream of a dual-gene reporter expressing mCitrine (yellow) and mCherry (red) separated by either (**A**) no spacer or (**B**) 5 kb of lambda DNA (gray). (**C–J**) Fluorescence distributions of mCitrine and mCherry measured by flow cytometry either without CR recruitment (-dox), or after 5 or 20 days of recruitment ( + dox) of either KRAB or HDAC4 at the NS or 5 kb reporters in either CHO-K1 or K562 as indicated in each title. Percentages of cells ON (high-fluorescence, top) or OFF (low-fluorescence, bottom) are calculated based on a threshold (dotted line). Data from independent clonal cell lines for CHO-K1 or biological replicates of multiclonal populations for K562 are shown as overlaid semi-transparent distributions (n = 3). A Welch's unequal variances T-test comparing the percent of cells off in -dox versus + dox for each gene, showed that the percent of cells silenced were statistically significant (p < 0.05) and reproducible for NS and 5 kb reporters in all cell types tested (**Appendix 1—table 1**).

The online version of this article includes the following source data and figure supplement(s) for figure 1:

**Figure supplement 1.** Reporter constructs used in different cell lines for analyzing spreading of transcriptional changes.

**Figure supplement 2.** Recruitment of KRAB and HDAC4 to 5 kb reporter in HEK293T.

**Figure supplement 3.** Silencing by KRAB at lower dox concentrations.

**Figure supplement 4.** Transcriptional run-on from pEF over pRSV.

**Figure supplement 4—source data 1.** Original gel images from RT-PCR.

observations are consistent with previous reports of spreading of silencing upon KRAB recruitment (*Groner et al., 2010*; *Amabile et al., 2016*; *Hathaway et al., 2012*).

Surprisingly, HDAC4 recruitment also resulted in silencing of both genes (*Figure 1G&J*), albeit not in all cells in the population: only 68–78% of CHO-K1 cells silenced mCherry with HDAC4 (*Figure 1G&I*) compared to 97–100% with KRAB (*Figure 1C&E*). Moreover, in K562 cells, HDAC4-mediated silencing of mCherry was much slower, taking up to 20 days, compared to 5 days in CHO-K1 cells (*Figure 1H&J*). With HDAC4-mediated recruitment at the 5 kb dual reporter in HEK293T at the AAVS1 locus, the majority of cells that silence mCitrine also silence the downstream mCherry within 7 days (*Figure 1—figure supplement 2B*), similar to the CHO-HAC system, and much faster than in

K562. These results suggest that cellular background can play a strong role in the speed of silencing and spreading, as all other factors in the systems are held constant such as reporter set up and genomic locus. A Welch's unequal variances T-test comparing the percent of cells off in -dox versus + dox for each gene, showed that the percent of cells silenced were statistically significant and reproducible for NS and 5 kb reporters in all cell types tested (*Appendix 1—table 1*). These results show that in a synthetic reporter system, where genes are separated by short distances up to 5 kb, silencing mediated by a histone deacetylase can also affect a nearby gene.

Additionally, we noticed that in CHO-K1 cells in the absence of dox, mCherry expression is lower and has a wider distribution in the 5 kb reporter compared to NS (*Figure 1E&I*, vs. *Figure 1C&G*). This observation suggests that the two promoters interfere, as observed before (*Shearwin et al., 2005*), in a distance-dependent manner. Surprisingly, in K562 we saw an increase in pRSV-mCherry expression after 5 days of dox-mediated HDAC4 recruitment and pEF-mCitrine silencing (*Figure 1H*). We found that this transcriptional interference between pEF and pRSV results from transcriptional run-on from the pEF promoter which is resolved upon mCitrine silencing (*Figure 1H*, *Figure 1—figure supplement 4A&B*, Appendix 2). Despite this type of transcriptional interference, we can still measure the silencing dynamics of the downstream gene.

## Changes in histone modifications upon transcriptional silencing of neighboring genes

To see if the changes in gene expression were accompanied by the expected changes in chromatin modifications, we used CUT&RUN (*Skene et al., 2018*; *Meers et al., 2019*) to map activating and repressive histone modifications (*Lensch et al., 2022*). Previous studies looking at histone modifications across the genome in different cell types have shown that active promoters and strong enhancers in euchromatin have high levels of acetylation (*Ernst et al., 2011*), which is in agreement with the general histone code by which acetylation correlates with open and active regions of euchromatin, while methylation results in silenced and compact heterochromatin (*Jenuwein and Allis, 2001*). KRAB recruitment is known to result in loss of acetylation and gain of H3K9me3 (*Ayyanathan et al., 2003*; *Groner et al., 2010*; *Amabile et al., 2016*; *Feng et al., 2020*), while HDAC4 is associated with loss of acetylation (*Wang et al., 1999*; *Wang et al., 2014b*). Recruitment of KRAB for 5 days in K562 cells at the 5 kb reporter resulted in loss of the activating histone 3 lysine acetylation (H3Kac) and a strong increase in the repressive modification H3K9me3 across the recruitment site, both fluorescent protein genes, and the 5 kb spacer (*Figure 2A&C*). This increase in H3K9me3 and decrease in H3Kac was also observed in the NS reporter line (*Figure 2—figure supplement 1A*) and was accompanied by a concomitant depletion of histone 3 lysine 4 trimethylation (H3K4me3), which is associated with active genes, in both reporter lines (*Figure 2—figure supplement 1&B*). Moreover, we observed an increase in H3K9me3 beyond the AAVS1 integration site into neighboring genes (*Figure 2B*), corresponding to a region of enrichment of approximately 50–60 kb beyond the reporter (*Figure 2—figure supplement 1C&D*). This too was accompanied by a depletion of H3Kac and H3K4me3 in the immediate vicinity of the AAVS1 integration site. The magnitude of histone modification changes in this vicinity was comparable between the NS and 5 kb reporter lines (*Figure 2—figure supplement 1E*). We observed similar changes in chromatin modifications in CHO-K1 cells after 5 days of KRAB recruitment to the 5 kb reporter (*Figure 2—figure supplement 2A*), showing that these changes in histone modifications coincide with the spreading of transcriptional silencing at the reporter in different cell lines. This loss of acetylation and gain of H3K9me3 and its spreading across a large domain including over neighboring genes is in line with what has been previously shown (*Groner et al., 2010*; *Amabile et al., 2016*), confirming that our system works as expected.

Recruitment of HDAC4 for 5 days in CHO-K1 cells to the 5 kb and NS reporters showed a decrease in acetylation levels, and no change in H3K9me3 (*Figure 2D&E*, *Figure 2—figure supplement 2B&C*), as expected. However, surprisingly histone 3 lysine 27 trimethylation (H3K27me3) was also detected throughout the 5 kb (*Figure 2D&E*) and the NS reporters (*Figure 2—figure supplement 2B*). This repressive modification is associated with polycomb repressive complex 2 (PRC2) and not with HDAC4. We have previously observed at a pEF-mCitrine reporter flanked by insulators in the same locus of CHO-K1 cells that HDAC4 recruitment does not lead to deposition of H3K27me3 (*Bintu et al., 2016*). Therefore, we hypothesized that the pRSV promoter gets silenced not by HDAC4, but by PRC2 after the pEF-mCitrine gene is silenced. Once the H3K27me3 modification is present, like

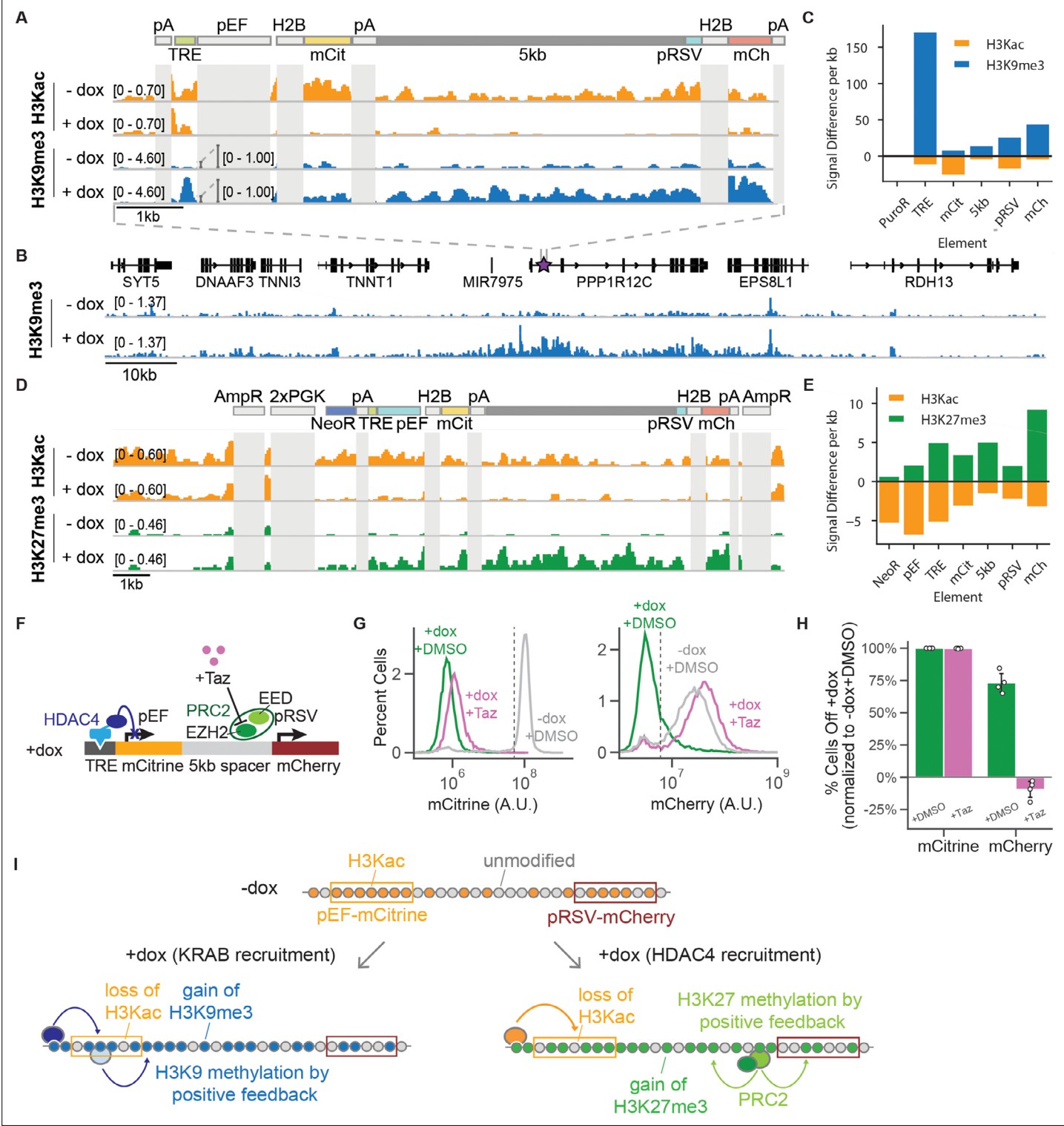

**Figure 2.** CUT& RUN data shows changes in histone modifications at silenced genes. (**A**) Genome browser tracks showing counts per million (CPM)-normalized reads after CUT& RUN for histone 3 acetyl-lysine (H3Kac) and histone 3 lysine 9 trimethylation (H3K9me3) with (+ dox) and without (-dox) recruitment of rTetR-KRAB to the 5kb-spacer reporter in K562 cells. Non-unique regions resulting in ambiguous alignment, including pEF, H2B, and polyA tails (pA), are masked in light gray. (**B**) Genome browser tracks of H3K9me3 with (+ dox) and without (-dox) recruitment of rTetR-KRAB, looking at the surrounding locus where the 5 kb reporter is integrated in cells (purple star within first intron of PPP1R12C). This snapshot does not include an in situ representation of the reporter, which instead was appended to the end of the reference genome to preserve gene annotations. (**C**) Quantification of signal difference for H3K9me3 and (H3Kac) with (+ dox) and without (-dox) recruitment of KRAB for 5 days to the 5 kb reporter in K562 cells. (**D**) Genome

*Figure 2 continued on next page*

*Figure 2 continued*

browser tracks showing CPM-normalized reads after CUT& RUN for H3Kac and histone 3 lysine 27 trimethylation (H3K27me3) with (+ dox) and without (-dox) recruitment of rTetR-HDAC4 to the 5kb-spacer reporter in CHO-K1 cells. Non-unique regions resulting in ambiguous alignment, including AmpR, PGK, H2B, and polyA tails (pA), are masked in light gray. (**E**) Quantification of signal difference for H3Kac and H3K27me3 with (+ dox) and without (-dox) recruitment of HDAC4 for 5 days to the 5 kb reporter in K562 cells. (**F**) Addition of Tazemetostat (Taz) inhibits the EZH2 methyltransferase from PRC2. (**G**) Fluorescence distributions of mCitrine (left) and mCherry (right) measured by flow cytometry either without CR recruitment (-dox, gray) or after 7 days of HDAC4 recruitment (+ dox + DMSO, green) and with Taz(+ dox + Taz, pink) at the 5 kb reporter in CHO-K1. (**H**) Percentages of cells OFF normalized by the -dox + DMSO condition, based on threshold shown in panel G (dotted line)(n = 4). (**I**) In the absence of CR recruitment, both genes have H3Ac across the reporter (top). Upon recruitment of KRAB (bottom left), we see a loss of H3Kac and gain of H3K9me3 across the dual-gene reporter through both DNA looping from rTetR-KRAB as well as positive feedback loops for spread of methylation, resulting in a distance-dependent delay of transcriptional silencing between two genes. Upon recruitment of HDAC4 (bottom right), we see a loss of H3Kac as well as a gain of H3K27me3 across the reporter through positive feedback loops for spread of methylation by PRC2, resulting in distance-independent delay of transcriptional silencing between two genes.

The online version of this article includes the following figure supplement(s) for figure 2:

**Figure supplement 1.** Changes in chromatin modifications at the two-gene reporter and surrounding AAVS1 locus after recruitment of KRAB for 5 days in K562 cells.

**Figure supplement 2.** Recruitment of KRAB or HDAC4 for 5 days in CHO-K1 cells induces changes in chromatin modifications at the two-gene reporters.

**Figure supplement 3.** EZH2 inhibitor Tazemetostat affects silencing after recruitment of EED and HDAC4.

H3K9me3, H3K27me3 also has reader-writer positive feedback (*Uckelmann and Davidovich, 2021*), that can lead to spreading of these chromatin modifications (*Figure 2I*). To test whether the silencing of mCherry was mediated by PRC2, we recruited HDAC4 in the presence of Tazemetostat (Taz), a small molecule inhibitor of EZH2, which is the H3K27 methyltransferase in PRC2 (*Figure 2F*). First, we tested Taz in the NS reporter line with recruitment of EED, the H3K27 reader in PRC2, and showed that inhibition reduced silencing of mCitrine and mCherry (*Figure 2—figure supplement 3A&B*). We found that in the presence of dox and Taz, while the mCitrine gene was still silenced by HDAC4, the mCherry gene was not silenced (and even slightly increased in expression) in both the 5 kb reporter (*Figure 2G&H*) and the NS reporter (*Figure 2—figure supplement 3C*). This observation suggests that silencing of the pRSV promoter is due to PRC2 action, but it can only take place once HDAC4 removes acetylation at the locus. In accordance with our hypothesis, previous studies have shown that H3K27me3 and H3Ac are negatively correlated as measured by co-ChIP in primary cells (*Weiner et al., 2016*), that lack of transcription and histone deacetylation both drive an increase in H3K27me3 deposition steadily over time (*Hosogane et al., 2016*), and that recruitment of HDAC1 leads to repressed chromatin associated with a strong increase in H3K27me3 (*Song et al., 2016*). Additionally, knockdown of p300, a histone acetyltransferase, resulted in a global decrease of H3K27Ac in tandem with an increase in H3K27me3 (*Martire et al., 2020*). Our system allows us to study the temporal and distance dependence dynamics of this type of indirect spreading of silencing, where silencing of one gene opens the door for other repressive complexes to silence neighboring genes (*Figure 2I*, right). We can compare these dynamics with those associated with direct silencing by KRAB, where repressive modifications that are associated with the recruited chromatin regulator spread across the locus (*Figure 2I*, left).

## Dynamics of spreading of transcriptional silencing—delay between two genes

We measured the spreading dynamics of transcriptional silencers at finer temporal resolution by first taking time-lapse movies of CHO-K1 cells with the NS and 5 kb spreading reporters (*Figure 3— videos 1–4*). Cells were imaged every 20 min for 4 – 5 days and cumulative fluorescence traces were used to determine transcriptional silencing times after dox addition (*Lensch et al., 2022*; *Figure 3A*, *Figure 3—figure supplement 1*). Since H2B fusions to fluorescent proteins have long half-lives on the order of weeks, their loss is mostly driven by cell division (*Morcos et al., 2020*), which allows the silencing of the gene to be detected based on the change in fluorescent protein accumulation. Stitching together single-cell traces across cell divisions results in cumulative traces of fluorescent protein levels over time for specific lineages (*Figure 3A*, *Figure 3—figure supplement 1*). The slope of these traces is proportional to promoter activity, and its decrease can be used to determine

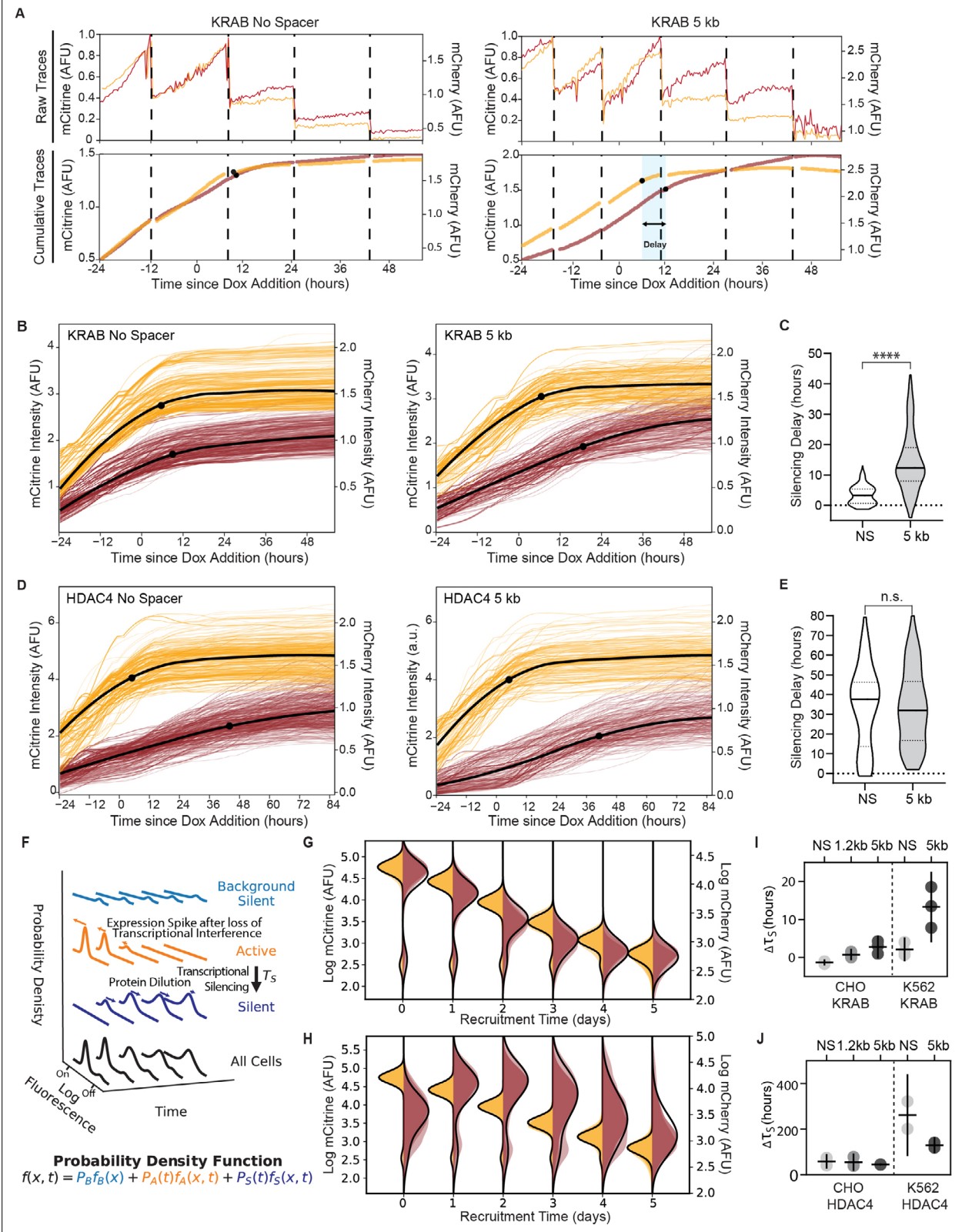

**Figure 3.** Single-cell data measures delay in transcriptional silencing of the two genes. (**A**) (Top) Example raw traces measured by time-lapse microscopy showing total fluorescence of mCitrine (yellow) and mCherry expression (red) as a function of time in an individual cell lineage in CHO-K1 with the NS reporter (left) and 5 kb reporter (right). Dotted lines denote cell divisions. Recruitment of KRAB starts after 24 hours by dox addition. (Bottom) Cumulative single-cell traces stitched across cell divisions showing estimated silencing time points (black dots, Materials and methods) and silencing

*Figure 3 continued on next page*

*Figure 3 continued*

delay (blue shading). (**B**) Cumulative single-cell traces of mCitrine (yellow) and mCherry (red) in CHO-K1 cells with NS reporter (left, n = 296) and 5 kb reporter (right, n = 218) with the mean trace (black curves) and median silencing times (dots) as a function of time since recruitment of KRAB by dox addition at time 0. (**C**) Distribution of delay times between silencing of mCitrine and mCherry in individual cells as shown in (**A**) after recruitment of KRAB in (NS reporter median silencing delay = 3.3 hr; 5 kb reporter median silencing delay = 12.3 hr; statistically significant difference by Welch's unequal variances T-test). (**D**) Cumulative single-cell traces as a function of time relative to HDAC4 recruitment (NS, n = 211; 5 kb, n = 291), as in (**B**). (**E**) Distribution of delay times after recruitment of HDAC4 (NS reporter median silencing delay = 38 hours; 5 kb reporter median silencing delay = 32 hours; no statistically significant difference by Welch's unequal variances T-test). (**F**) A probabilistic model consisting of three states: background silent (light blue), active (orange), silent (dark blue). Each state has its own weight and distribution, and all states are summed to a final probability density function (black) that describes the probability $f$ of finding a cell with fluorescence $x$ at time $t$. This model is used to fit daily flow cytometry data to extract transcriptional silencing times ($\tau_S$) upon CR recruitment, while taking into account: stochastic transitions of cells from the active to the silent state upon CR-mediated silencing, spike in expression after loss of transcriptional interference, and mRNA and protein degradation and dilution (Materials and methods). (**G, H**) Overlaid daily distributions of mCitrine (transparent yellow) and mCherry (transparent red) fluorescence from flow cytometry during recruitment of (**G**) KRAB and (**H**) HDAC4 with average model fit (black line) (n = 3). (**I**) Silencing delays between mCitrine and mCherry after KRAB recruitment extracted from daily flow cytometry time-courses using the model in (**F**) for different spreading reporters: NS, 1.2 kb, 5 kb. Each dot represents a clone for CHO-K1 (left) and a biological replicate for K562 (right); horizontal bar is mean delay, vertical bar is 90% confidence interval from the fit estimated using the t-distribution (n = 3). (**J**) Silencing delays between mCitrine and mCherry after HDAC4 recruitment (CHO-K1, n = 3 clones; K562 n = 2 biological replicates); same notation as (**I**).

The online version of this article includes the following video and figure supplement(s) for figure 3:

**Figure supplement 1.** Example images and single-cell analysis of silencing dynamics from time-lapse microscopy of CHO-K1 cells.

**Figure supplement 2.** Dynamics of silencing measured by flow cytometry and fit by gene expression model.

**Figure supplement 3.** Steady-state expression in the absence of dox in CHO-K1 versus K562 cells.

**Figure 3—video 1.** Spreading of silencing movie for KRAB NS.
https://elifesciences.org/articles/75115/figures#fig3video1

**Figure 3—video 2.** Spreading of silencing movie for KRAB 5 kb.
https://elifesciences.org/articles/75115/figures#fig3video2

**Figure 3—video 3.** Spreading of silencing movie for HDAC4 NS.
https://elifesciences.org/articles/75115/figures#fig3video3

**Figure 3—video 4.** Spreading of silencing movie for HDAC4 5 kb.
https://elifesciences.org/articles/75115/figures#fig3video4

---

silencing of a gene (Materials and methods). Upon KRAB recruitment, we observed only a slight delay of 2.3 ± 3.1 hr between mCitrine and mCherry silencing in the NS reporter and a longer delay of 12.3 ± 9.8 hr in the 5 kb reporter (*Figure 3B&C*, *Figure 3—figure supplement 1A&B*). In contrast, spreading of HDAC4-mediated transcriptional silencing was slower, and no appreciable difference in silencing delays was observed between reporter lines (NS: 37 ±21 hr; 5 kb: 32 ±19 hr) (*Figure 3D&E*, *Figure 3—figure supplement 1C&D*). Taken together, these data suggest that KRAB-mediated spreading of transcriptional silencing is distance-dependent in CHO-K1 cells while HDAC4 mediated spreading is distance-independent at these length scales.

Since time-lapse microscopy measurements are limited to hundreds of cells and hard to extend to non-adherent cell lines like K562, we developed an alternative approach to extracting the delay times between silencing of the two reporter genes from flow cytometry data. We developed a mathematical model that describes the evolution of mCitrine and mCherry fluorescence distributions after CR recruitment (*Lensch et al., 2022*; *Figure 3F*, Materials and methods). We used our daily flow cytometry measurements of fluorescence distributions to fit silencing times for each gene (*Figure 3G–J*), along with other parameters associated with gene expression including a spike in mCherry mRNA production due to transcriptional interference (as shown by qPCR in *Figure 1—figure supplement 4C&D*) and mRNA and protein degradation and dilution (Materials and methods). This approach allowed us to estimate the delay between mCitrine and mCherry silencing, $\Delta \tau_s$, at the gene rather than protein level for all spreading reporters in CHO-K1 and K562 with different CRs recruited (*Figure 3G–J*, *Figure 3—figure supplement 2*, *Appendix 1—table 2*). Overall, the delay between mCitrine and mCherry silencing are similar between movie traces and fits of the cytometry data (*Figure 3C&E*, vs. *Figure 3I&J*), especially if we consider the additional time necessary for mRNA degradation of about ~ 4 hours included in the flow fits (Materials and methods), as determined previously by inhibiting transcription with actinomycin D and quantifying the decay of H2B-mCitrine mRNA over time

(*Bintu et al., 2016*). Since we observed a delay time between mCherry and mCitrine silencing averaging less than 4 hours after KRAB recruitment to the NS reporter (*Figure 3B*), we assumed that the H2B-mCherry mRNA degrades at a similar rate as H2B-mCitrine.

These delays show similar trends with distance in CHO-K1 and K562 cells for a given CR. KRAB recruitment results in silencing delays that increase with intergenic distance (*Figure 3I*). HDAC4 recruitment results in delays that are not statistically different among different distances in either cell type, but show a trend where delay decreases with increased distance in K562 from around 10 days in the NS reporter to around 5 days in the 5 kb reporter (*Figure 3J*). We have determined that pRSV-mCherry silencing after HDAC4 recruitment is mediated by PRC2 (*Figure 2F–H*). If PRC2 were recruited directly by HDAC4 near the pEF promoter, we would expect the silencing delay to increase with distance, as we see with KRAB; however, we see that delay with HDAC4 recruitment is distance-independent. Therefore, the observation that the delays of pRSV-mCherry silencing do not increase with distance upon HDAC4-mediated silencing (*Figure 3E&J*), together with the loss of mCherry silencing but not mCitrine silencing upon HDAC4 recruitment in the presence of Taz (*Figure 2F&H*), suggests that the silencing of mCherry is initiated at the pRSV promoter, by the action of PRC2 (*Figure 2F&I*).

Silencing spreads slower in K562 versus CHO-K1 cells after both KRAB and HDAC4 recruitment (*Figure 3I&J*), consistent with the 5 days results (*Figure 1*). This is most likely because the pRSV promoter is stronger in K562 cells, as indicated by higher mCherry expression compared to CHO-K1 cells for both the NS and 5 kb reporters (*Figure 3—figure supplement 3*). These small differences in temporal dynamics are likely due to differences in context of the reporters, both different genomic locus and cell type.

## The role of the cHS4 insulators in spreading of transcriptional silencing between genes

Insulators are sequences shown to prevent transgene silencing and believed to stop spreading of repressive chromatin modifications (*Barkess and West, 2012*; *Gaszner and Felsenfeld, 2006*). The cHS4 insulator is a commonly used insulator for protecting transgene silencing in many cell lines (*Recillas-Targa et al., 2004*); most of its insulator activity has been attributed to a 250 bp core region (*Chung et al., 1997*; *Recillas-Targa et al., 2002*; *West et al., 2004*; *Yusufzai and Felsenfeld, 2004*; *Gowher et al., 2012*), that is associated with increased histone acetylation (*Mutskov et al., 2002*; *Zhao and Dean, 2004*). To measure the role of the cHS4 insulator on blocking the spreading of targeted gene silencing, we built four different insulator configurations: a 'single insulator' between the two genes using either the full-length 1.2 kb cHS4 (SH) or its 250 bp core region (SC) (*Figure 4A*), or insulators flanking the mCitrine gene, referred to as 'double insulator', with either full cHS4 (DH) or core region (DC) (*Figure 4B*). This double configuration is commonly used in mammalian cell engineering to prevent background silencing of transgenes due to position effect variegation (*Chung et al., 1993*).

Surprisingly, no cHS4 insulator configuration was capable of stopping KRAB-mediated spreading of silencing in either CHO-K1 or K562 cells (*Figure 4C&D*, *Figure 4—figure supplement 1A&B*). Moreover, cHS4 insulators do not delay KRAB-mediated silencing either: the estimated delay times in the insulator reporters are close to the delays in reporters with spacers (*Figure 4F*, *Appendix 1— table 2*). Therefore, these cHS4 insulator configurations do not have a strong effect on silencing mediated by KRAB recruitment.

In CHO-K1 cells, no cHS4 insulator configuration was able to inhibit spreading of silencing upon HDAC4 recruitment (*Figure 4E*, *Figure 4—figure supplement 1C*), and the small fraction of cells where mCherry is not silenced by day 5 of recruitment is similar to the spacer reporters (*Figure 3— figure supplement 2B*). Also similar to the spacer reporters, the fitted delay times ranged from 42 to 85 hr (*Figure 4F* right, *Figure 4—figure supplement 1C*). These delay times fall within the 90% confidence interval of the 1.2 kb lambda spacer (*Appendix 1—table 2*). Therefore, insulators do not appear to have a strong effect on HDAC4-mediated silencing at our reporters in CHO-K1 cells.

However, cHS4 insulators do appear to attenuate HDAC4-mediated mCherry silencing in K562 cells, even with 20 days of recruitment (*Figure 4G&H*, *Figure 4—figure supplement 2*). Rather than seeing complete silencing of the mCherry reporter, we see a broadening of the mCherry fluorescence distribution with the majority of cells expressing lower levels of mCherry than the population without

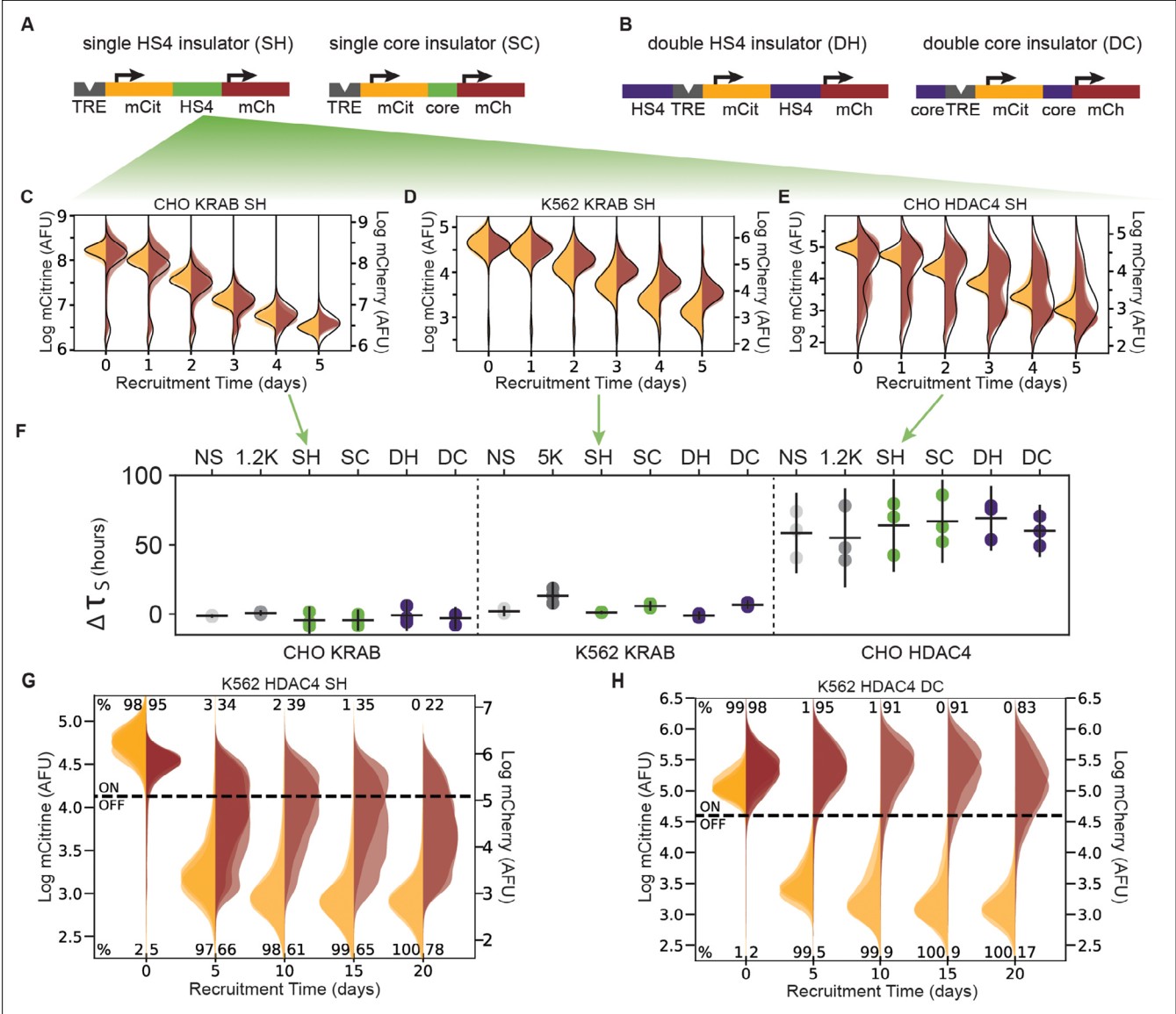

**Figure 4.** The role of the cHS4 insulators in spreading of transcriptional silencing across genes. (**A**) Single insulator geometries between mCitrine (mCit) and mCherry (mCh) fluorescent genes, with full 1.2 kb HS4 insulator (SH, left) and 250 bp core insulator (SC, right). (**B**) Double insulator geometries with full-length 1.2 kb HS4 insulator (DH, left) and 250 bp core insulator (DC, right) flanking the TRE-pEF-mCitrine region. (**C–E**) Overlaid replicates of daily distributions of mCitrine (yellow) and mCherry (red) fluorescence from flow cytometry during recruitment of (**C**) KRAB in CHO-K1, (**D**) ) KRAB in K562, or (**E**) HDAC4 in CHO-K1, to SH reporters with average model fit (black line) (n = 3). (**F**) Silencing delay times between mCitrine and mCherry in different insulator reporters after chromatin regulator recruitment for 5 days (each dot is a replicate, horizontal bar is mean delay, vertical bar is 90 % confidence interval estimated using the t-distribution). (**G–H**) Overlaid replicates of daily distributions of mCitrine (yellow) and mCherry (red) fluorescence from flow cytometry during extended recruitment of HDAC4 to (**G**) SH reporter or (**H**) DC reporter in K562 (n = 3 ).

The online version of this article includes the following figure supplement(s) for figure 4:

**Figure supplement 1.** The effect of all insulator configurations on spreading dynamics of transcriptional silencing in CHO-K1 and K562.

**Figure supplement 2.** Insulators attenuate spreading of silencing via HDAC4 in K562.

**Figure supplement 3.** Dynamics of spreading upon weaker gene targeting insulator reporters with HDAC4 at lower dox concentrations in CHO-K1.

**Figure supplement 4.** Insulators do not block spreading of silencing with weaker gene targeting at lower dox concentrations.

**Figure supplement 5.** Insulators prevent background silencing of reporter genes.

dox, both at 5 and 20 days of HDAC4 recruitment to cHS4 insulator constructs in K562 (*Figure 4G*, *Figure 4—figure supplement 2*). An exception to this observation is the DC configuration, where the mCherry distribution remains in the ON range even after 20 days of HDAC4 recruitment (*Figure 4H*). The lack of complete mCherry silencing and broader mCherry distribution is in contrast to the silencing seen at the NS and 5 kb reporters after 20 days of HDAC4 recruitment in K562 (*Figure 1H&J*). These results suggest that in K562, where silencing by HDAC4 recruitment has slower dynamics compared to CHO, the cHS4 insulators can help the downstream gene remain active.

To test if cHS4 insulators can block weaker gene targeting by HDAC4 in CHO-K1 cells, we performed silencing experiments at non-saturating dox concentrations. At lower levels of dox, silencing of the two genes is slower; fits of daily flow cytometry data show silencing delay times that decrease upon increasing dox concentrations for all cHS4 insulator configurations (*Figure 4—figure supplement 3*). At lower dox concentrations, fewer cells silence mCitrine by the end of 5 days, but those that do are likely to also silence mCherry for both HDAC4 and KRAB (*Figure 4—figure supplement 4*), showing that the insulators do not block spreading of silencing in CHO-K1 even with weaker CR targeting.

Although the cHS4 insulators do not generally prevent spreading of silencing during recruitment of CRs, the insulators are able to prevent spontaneous background silencing of the reporters in the absence of dox in CHO-K1 cells (*Figure 4—figure supplement 5*, Appendix 3) consistent with previous transgene silencing reports (*Recillas-Targa et al., 2002*; *Pikaart et al., 1998*). Overall, levels of mCherry expression are higher in the cHS4 insulator constructs (*Figure 3—figure supplement 3*), and there is no transcriptional run-on mRNA across the SH insulator (Appendix 2, *Figure 1—figure supplement 4E&F*), suggesting it helps terminate transcription and prevent transcriptional interference as shown before (*Tian and Andreadis, 2009*).

## Reactivation of genes

To understand the dynamics of neighboring gene coupling during transcriptional activation, we investigated the order of gene reactivation and the degree of epigenetic memory in our various two-gene constructs. We first silenced our various reporters with different spacers or insulator configurations for 5 days, then removed dox to release the CR and monitored gene expression every few days by flow cytometry (*Figure 5A*). In CHO-K1 cells, while both genes reactivated simultaneously in the NS reporter (*Figure 5B*, *Figure 5—figure supplement 1A&B*), mCitrine reactivated first in the 5 kb reporter (*Figure 5C*, *Figure 5—figure supplement 1C&D*), both after release of KRAB or HDAC4. This pattern of a distance-dependent delay in reactivation between mCitrine and mCherry also holds for the insulator configurations for both KRAB and HDAC4: the full 1.2 kb long cHS4 configurations featured delayed mCherry reactivation (*Figure 5D*, *Figure 5—figure supplement 2*), compared to the more synchronous reactivation patterns observed in the 250 bp long core insulator configurations (*Figure 5—figure supplement 2*). The order of reactivation suggests that in CHO-K1 cells reactivation initiates at the stronger pEF promoter that drives mCitrine, and spreads in a distance-dependent manner to the weaker pRSV-mCherry gene.

However, when looking at order of reactivation in K562 cells after release of KRAB, we found that for the NS reporter, mCherry reactivates first (*Figure 5E*, *Figure 5—figure supplement 3A*), while with the 5 kb reporter we observe three different scenarios in individual cells: either mCherry reactivates first, mCitrine reactivates first, or they reactivate together (*Figure 5F*, *Figure 5—figure supplement 3B*). In the cHS4 insulator constructs, mCherry reactivates before mCitrine in K562 cells (*Figure 5G*, *Figure 5—figure supplement 3C*). This change in reactivation pattern in K562 cells compared to CHO-K1 is likely due to the pRSV promoter being stronger in K562 (*Figure 3—figure supplement 3*).

Despite not preventing the spreading of silencing by KRAB, the cHS4 insulators do play a role in the reactivation rate and level of memory. After release of rTetR-KRAB, reporters with insulators reactivated more rapidly compared to the NS and 5 kb reporters, with the double core insulator exhibiting the highest degree of gene reactivation in both CHO-K1 and K562 cells (*Figure 5H&I*). This is in line with the double core insulating elements also having the strongest insulating effect against mCherry silencing after HDAC4 recruitment in K562. These results show that the extent of epigenetic memory in our system depends on the chromatin regulator recruited and the configurations of promoters and insulators at the target locus, with the stronger pEF promoter closely surrounded by cHS4 insulators diminishing memory.

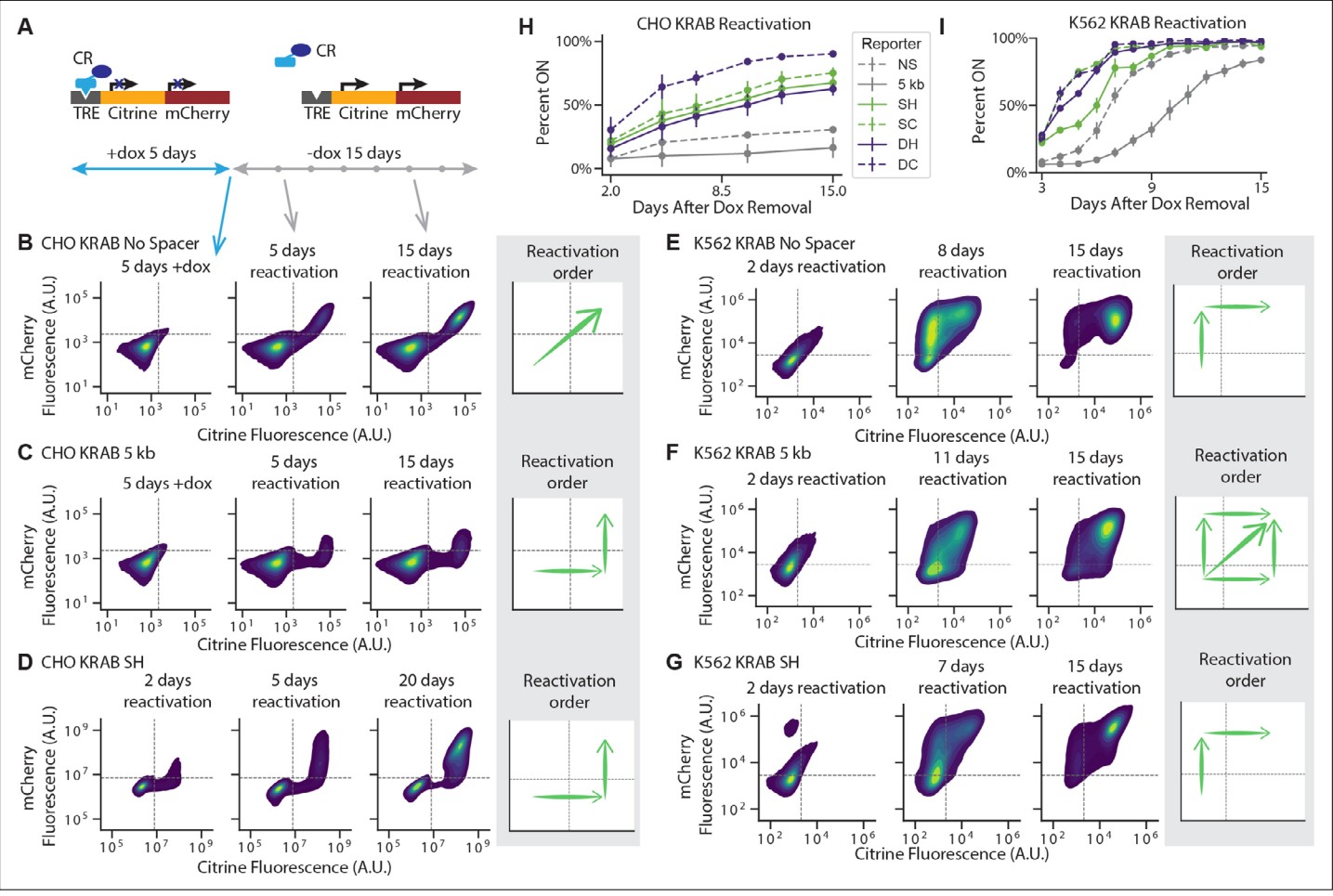

**Figure 5.** Reactivation of gene expression spreads between the two genes. (**A**) Schematic of experimental setup: each CR is recruited upstream of the mCitrine gene for 5 days. Recruitment is then stopped by removing dox and reactivation is monitored every few days by flow cytometry. (**B-G**) 2D density plots of mCitrine and mCherry fluorescence from flow cytometry show pattern of gene reactivation in: (**B**) CHO-K1 KRAB NS (n = 3 clones), (**C**) CHO-K1 KRAB 5 kb (n = 4 clones), (**D**) CHO-K1 KRAB SH (n = 3), (**E**) K562 KRAB NS (n = 3), (**F**) K562 KRAB 5 kb (n = 3), and (**G**) K562 KRAB SH (n = 3). (**H-I**) Percent of cells in which at least one reporter gene reactivated over time after dox removal after KRAB release in (**H**) CHO-K1 and (**I**) K562. Replicates are from either independent clonal cell lines, where indicated, or from biological replicates of multiclonal populations.

The online version of this article includes the following figure supplement(s) for figure 5:

**Figure supplement 1.** Reactivation of gene expression in CHO-K1 NS and 5 kb reporter lines.

**Figure supplement 2.** Reactivation of gene expression in CHO-K1 insulator reporter lines.

**Figure supplement 3.** Reactivation of gene expression in K562 cell reporter lines.

## Model connecting chromatin states to gene expression dynamics

We developed a kinetic model that summarizes our observations of gene dynamics and chromatin modifications as a competition between the distance-dependent silencing rates associated with each tethered chromatin regulator and the reactivation rates driven by our promoters and insulators. In this model, each gene can be either active or silent, leading to four possible states in our two-gene reporters (*Figure 6A*). Each gene can transition between active and silent states with rates that depend on the distance between itself and other DNA elements that recruit chromatin regulators: the CR recruitment sites, promoters, and insulators. Under different experimental conditions, different rates in this kinetic pathway dominate, governing the transitions from one state to another. When KRAB is recruited upstream of pEF-mCitrine, silencing of the downstream pRSV-mCherry gene happens quickly (over hours) at a rate that decreases as the distance between pRSV-mCherry and pEF-mCitrine increases (*Figure 6B*, $k_{s\_KRAB}(d)$), suggesting the distance-dependent silencing rates induced

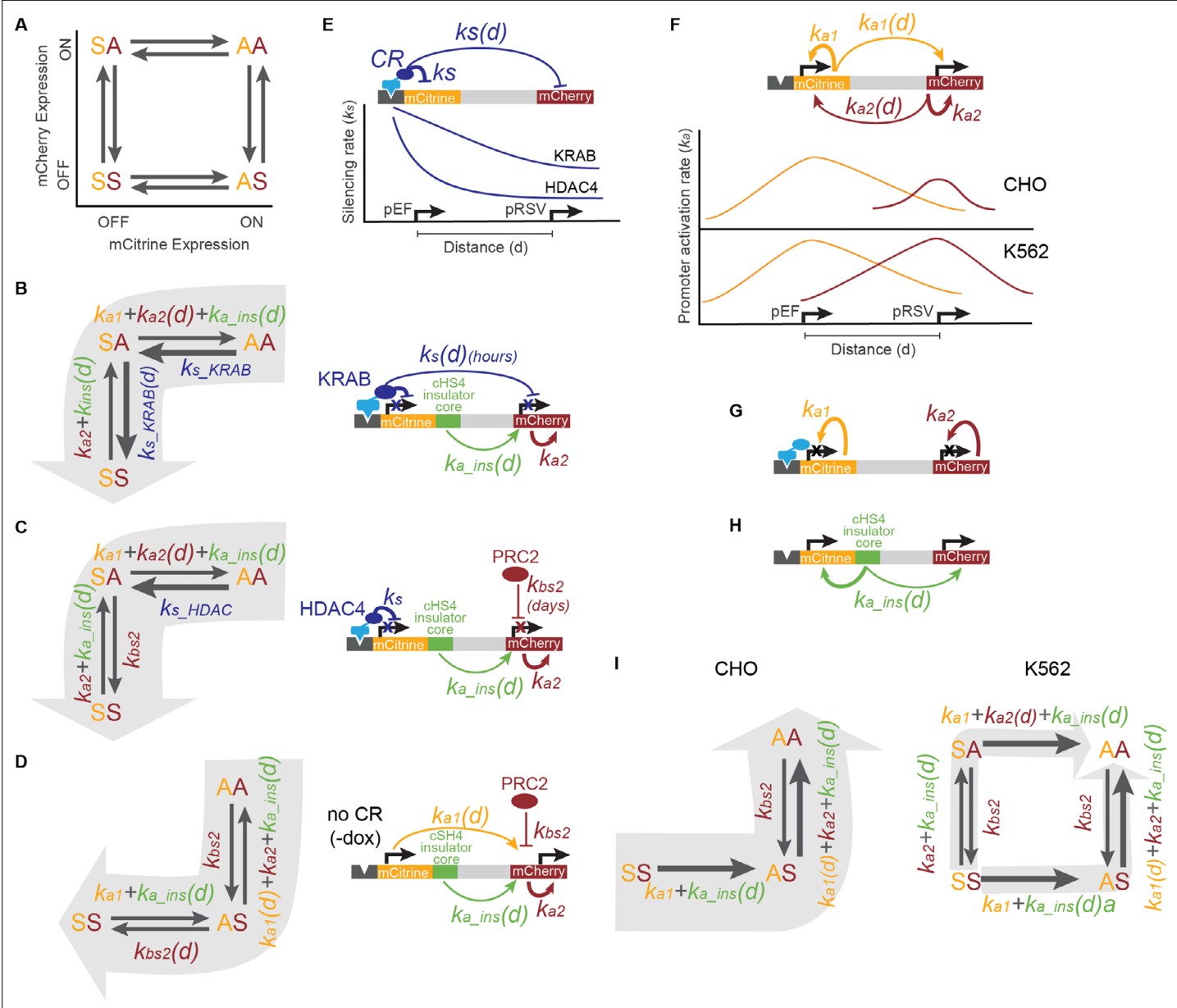

**Figure 6.** Model of multi-gene control coordinated by the action of CRs, promoters, and insulators. (**A**) Four states for a two-gene reporter, where the first letter (yellow) represents the mCitrine state and the second letter (red) represents mCherry state, as either active (A) or silent (S). Arrows represent the rates at which each gene is turned on or off. (**B**) During KRAB recruitment, cells transition from both genes active (AA) to both gene silent (SS), with mCitrine silencing first (SA intermediate state). The silencing rates of KRAB at the nearby pEF-mCitrine ($k_{s\_KRAB}$) and pRSV-mCherry ($k_{s\_KRAB}(d)$, where $d$ is the distance between the pEF and pRSV promoters) dominate over the activation rates of promoters ($k_{a1}$, $k_{a2}$) and insulators ($k_{a\_ins}(d)$, where here $d$ is the distance between the core insulator and a nearby promoter). (**C**) During HDAC4 recruitment, silencing of mCitrine (AA to SA) is driven by the silencing rate of HDAC4 ($k_{s\_HDAC4}$), while silencing of mCherry (SA to SS) is driven by background silencing rate ($k_{bs2}$) due to PRC2. Both the pRSV promoter ($k_{a2}$) and insulator reactivation rates ($k_{a\_ins}$) can compete with pRSV-mCherry silencing. (**D**) Before CR recruitment (no dox), pEF as well as insulators can act from a distance on pRSV ($k_{a1}(d)$, $k_{a\_ins}(d)$) preventing background silencing of mCherry ($k_{bs2}$). (**E**) KRAB can act on genes over a long distance ($k_s(d)$), silencing both mCitrine and mCherry, while the range of HDAC4 silencing is much smaller ($k_s$), only directly affecting mCitrine. (**F**) In the absence of CR recruitment, promoters can activate themselves ($k_{a1}$, $k_{a2}$), and maintain activity of genes at a distance ($k_{a1}(d)$, $k_{a2}(d)$). In CHO-K1, pRSV is weaker than in K562. (**G**) When genes are silenced, their promoters cannot act on neighboring genes, instead only reactivate themselves ($k_{a1}$, $k_{a2}$). (**H**) The core element of the cHS4 insulator can maintain activity of the nearby genes and drive reactivation after CR-mediated silencing ($k_{a\_ins}(d)$). (**I**) Reactivation after silencing is driven by gene activation rates ($k_{a1}$, $k_{a2}$) and core cHS4 insulators ($k_{a\_ins}(d)$) if present. In CHO-K1 cells where pEF is stronger than pRSV, mCitrine reactivates first followed by mCherry (left). However, in K562 where both promoters are equally strong, either gene can reactivate first (right).

by KRAB dominate over the activation rates associated with the promoters and insulators (***Figure 6B***, $k_{a1/2}$ and $k_{a\_ins}$, respectively).

However, when HDAC4 is recruited upstream of pEF-mCitrine, we observe delayed silencing of pRSV-mCherry (over many days, compared to hours in KRAB) at a rate that does not change as a function of the distance between the two genes. We hypothesize that pRSV-mCherry silencing is not due to the direct action of the HDACs recruited upstream of the pEF, but rather due to background silencing (***Figure 6C***, $k_{bs2}$) of pRSV by endogenous polycomb complexes as indicated by the appearance of H3K27me3 (***Figure 2D&E***, ***Figure 2—figure supplement 2B&C***), loss of pRSV-mCherry silencing upon PRC2 inhibition (***Figure 2G–H***), and the fact that the silencing delay is distance-independent. In line with our hypothesis, previous studies have shown that the pRSV promoter is more prone to transgene background silencing than the pEF promoter when integrated into mammalian genomes (***Garrison et al., 2007***).

Because we see much less pRSV-mCherry background silencing when the upstream pEF- mCitrine is active (in the absence of dox), we conclude that when pEF is active, it can increase activity of the downstream pRSV and prevent it from background silencing (***Figure 6D***, $k_{a1}(d)$). Once the pEF-mCitrine gene is silenced by HDAC4, it can no longer bolster the overall rate of reactivation at the pRSV-mCherry gene ($k_{a1}(d)$ is missing in ***Figure 6C*** compared to 6D) and protect it against background silencing. This reasoning is also consistent with the observation that we see less background silencing of the pRSV promoter when it is closer to the pEF in the no spacer construct compared to the 5 kb reporter in the absence of CR recruitment (***Figure 1***).

In summary, in our system both silencing and reactivation rates are affected by the distance between two genes. Upon recruitment of KRAB, a CR associated with reader-writer feedback and spreading of silencing histone modifications, the silencing rate is maximal near its recruitment sites and decreases slowly with distance (***Figure 6E***). However, HDAC4 recruitment leads to histone deacetylation, which is not associated with reader-writer positive feedback, and therefore is expected to decrease quickly over distance (***Figure 6E***). We hypothesize that CRs that are associated with positive reader-writer feedback loops can directly affect nearby genes, while CRs without feedback can only indirectly affect neighboring genes by changing the state of promoters very close to the recruitment site, which can in turn influence farther genes.

In our system, promoters can be thought of as regions that are associated with high rates of gene activation and can drive reactivation after silencing. In the absence of chromatin regulator recruitment, each gene drives its own activation and in addition drives the reactivation of the neighboring gene in a distance-dependent manner (***Figure 6F***, top). In CHO-K1 cells, the pEF is a stronger promoter than the pRSV, while in K562 their strengths are closer (***Figure 6F*** bottom, ***Figure 3—figure supplement 3***). However, when a gene is silent, we are led to assume that the promoter can only reactivate itself and has no effect on reactivation of the adjacent gene until it is active again (***Figure 6G***). This assumption is necessary to explain why the silencing of pRSV after HDAC4 recruitment does not depend on its distance to the upstream silenced pEF promoter, even though an active pEF can increase pRSV reactivation in a distance dependent manner.

Similar to promoters, the core region of the cHS4 insulator can be modeled as a DNA element that increases the rate of reactivation of nearby genes in our setup in a distance-dependent manner (***Figure 6H***). This action, combined with our promoter reactivation rates, can prevent background gene silencing (***Figure 6D***) and drive reactivation after targeted CR-mediated silencing (***Figure 6B&C***). For example, because insulators increase reactivation rates, they can fight background silencing of the pRSV-mCherry both upon HDAC4 recruitment (***Figure 6C***), and under conditions without dox (***Figure 6D***). In K562, where the rate associated with pRSV activation is higher than in CHO-K1, the core insulators can bring the overall activation rate above background silencing when they are closer to pRSV. However, because the activation rates associated with insulators are small compared to KRAB-mediated silencing rates, they cannot insulate well against KRAB.

Thinking of the promoters and core insulators in our synthetic constructs as elements associated with higher reactivation rates also explains the distance-dependent gene reactivation after release of the chromatin regulators. When dox is removed after both genes are silenced, the reactivation dynamics are determined by the strength of the promoters and presence of insulators. In CHO-K1 cells, the stronger pEF gene and presence of any insulators leads to the reactivation of the mCitrine gene, which then can act at a distance to help reactivate the weaker pRSV-mCherry gene (***Figure 6I***,

left). This leads to insulated reporters having the fastest reactivation and also explains the order of gene reactivation in the NS and 5 kb reporters. In K562, where the strength of the pEF and pRSV promoters are more balanced, either gene can reactivate first and along with insulators drive reactivation of both genes (*Figure 6I*, right).

## Discussion

Characterizing how transcriptional silencing mediated by CRs affects neighboring genes is important for understanding interactions between adjacent genes in the genome and in synthetic circuits, as well as for developing safe gene therapy and epigenetic editing applications. To tease out the rules associated with these interactions, we engineered a series of synthetic reporters with different configurations, changing the distance between genes and testing different insulator arrangements in two mammalian cell lines. These reporters allowed us to monitor the extent and dynamics of spreading of silencing during recruitment of a chromatin regulator, as well as reactivation patterns and dynamics after its release. Using time-lapse microscopy and flow cytometry, we found that transcriptional silencing of a gene following recruitment of either KRAB or HDAC4 can affect a downstream gene even when separated by 5 kb of distance or different cHS4 insulator arrangements. KRAB silencing, associated with both histone deacetylation and H3K9 methylation, spreads with a delay that is very short (hours), increases with the distance between the genes, and is not affected by insulators. HDAC4 spreading, associated with histone deacetylation and H3K27me3, is slower (days) and does not depend on the distance between the genes. HDAC4-mediated silencing can also spread past insulators, though insulators seem to have a stronger effect on it, even blocking it in the DC configuration.

KRAB was previously shown to lead to spreading of both gene silencing and histone methylation when recruited near a gene as a fusion to either TetR (*Groner et al., 2012*; *Amabile et al., 2016*) or to programmable DNA-binding domains, such as dCas9 (*Feng et al., 2020*; *Thakore et al., 2015*) or TALENs (*Cong et al., 2012*). However, the extent of reported spreading varied in different contexts between tens to hundreds of kilobases. Our results help frame the extent of spreading of KRAB-mediated silencing as a dynamic process that depends on the time of recruitment as well as the level of activation at the target locus, which can be influenced by neighboring promoters and insulators. For example, when KRAB is recruited for a short period at an enhancer in the hemoglobin locus in K562, the extent of spreading of methylation is very small ( < 1 kb) (*Thakore et al., 2015*). In contrast, when KRAB is recruited for a long period (41 days) at a site with moderate acetylation, silencing can spread over 200 kb (*Amabile et al., 2016*). Similarly, recruitment of HP1-alpha, a CR in the same silencing pathway as KRAB, has been shown to lead to silencing and spreading of H3K9me3 across ~ 10 kb over time (*Hathaway et al., 2012*). The strength of recruitment also affects silencing and spreading as we saw from low dox recruitment (*Figure 1—figure supplement 3*). Additionally, in our system, we see that silencing is slower and occasionally inhibited by the presence of the cHS4 insulator, which contains high levels of acetylation. In our model, we propose that genes in active regions can also cooperate with neighboring genes in addition to insulators to oppose silencing by chromatin regulators. However, there are other factors at play in targeting of endogenous loci which affect gene silencing. In previous studies, targeting dCas9-KRAB to hundreds of repeated sgRNA sites forms a large domain of H3K9me3 heterochromatin on the order of megabases in a few days, but does not result in widespread gene silencing, rather the silencing of genes is controlled by the loss of H3K27Ac and H3K4me3 (*Feng et al., 2020*). Additionally, previous studies have also shown that while genes that are more susceptible to KRAB-mediated silencing with KAP1 have higher levels of repressive histone marks at the promoter and gene body, although surprisingly they have higher levels of active histone marks surrounding the gene (*Meylan et al., 2011*), indicating that the genomic locus affects silencing by KRAB. Therefore, more work needs to be done to understand how the genomic and epigenetic environment surrounding a group of genes can affect their repressibility.

The fact that KRAB-mediated silencing can spread to neighboring genes quickly raises two concerns for synthetic gene control and epigenetic editing. First, in mammalian genetic circuits where multiple genes need to be integrated together in a cell, genes controlled by KRAB need to either be placed far away from other genes to avoid unwanted interference and feedback, or other mechanisms of escaping silencing potentially through recruitment of H3K9me demethylases such as PHF8 (*Kleine-Kohlbrecher et al., 2010*), as may be the case in regulation of zinc finger gene clusters which are

auto- and co-regulated by KAP1 recruitment (*O'Geen et al., 2007*). Second, when targeting endogenous genes with KRAB (as in CRISPRi), the possibility of silencing spreading beyond the target gene should be considered. Previous work has shown that dCas9-KRAB targeting of promoters or enhancers results in silencing of the associated gene along with enrichment of H3K9me3 at the targeted locus (*Das and Chadwick, 2021*; *Thakore et al., 2015*; *Kearns et al., 2015*; *Klann et al., 2017*; *O'Geen et al., 2017*). Targeting of promoters of endogenous genes appears to result in spreading of H3K9me3 across the gene body (*Klann et al., 2017*), while targeting enhancers can result in either H3K9me3 only at the enhancer (*Thakore et al., 2015*) or only at the target gene (*Kearns et al., 2015*). However, we are not aware of ChIP-seq data after dCas9-KRAB recruitment to an endogenous promoter to determine the extent of spreading of H3K9me3 beyond the gene body.

In contrast to KRAB, silencing by HDAC4 (and in general by HDACs) is not traditionally associated with spreading of heterochromatin in mammalian cells, so we were initially surprised to observe that silencing mediated by HDAC4 can affect neighboring genes. In general, spreading of heterochromatin-mediated silencing is commonly associated with reader-writer feedback, which has been shown to contribute to heterochromatin spreading in theoretical models (*Dodd et al., 2007*; *Hodges and Crabtree, 2012*; *Erdel and Greene, 2016*; *Erdel, 2017*) and synthetic experimental systems (*Park et al., 2019*). HDAC4 is not directly associated with CRs that can bind deacetylated histone tails; therefore, we did not expect to see new histone modifications after silencing. However, the appearance of H3K27me3, a PRC2 modification known to have reader-writer feedback, after HDAC4 recruitment led us to believe that the silencing of pEF-mCitrine allowed background silencing of pRSV-mCherry, which was confirmed by the loss of mCherry silencing when PRC2 was inhibited (*Figure 2F–H*). In accordance with our observations, previous studies have shown that H3K27me3 and H3Ac are negatively correlated on individual nucleosomes (*Weiner et al., 2016*), and specifically that deacetylation by either lack of transcription or recruitment of HDAC1 leads to increases in H3K27me3 (*Hosogane et al., 2016*; *Song et al., 2016*). Additionally, studies where cells are treated with HDAC inhibitor show in an increase in histone acetylation which is correlated with reduced recruitment of EZH2, EED and Suz12 proteins from the PRC2 as well as a decrease in H3K27me3 (*Reynolds et al., 2012*; *Wang et al., 2012*), showing the bidirectionality of the inverse correlation between H3K27me and H3K27Ac (*Pasini et al., 2010*). This supports our observation that low levels of acetylation allow PRC2 to work more effectively. This indirect silencing scenario could also arise upon silencing of endogenous promoters (for example in development, aging, or synthetic gene control), where their silencing would allow natural silencer elements in the gene neighborhood to suddenly start working. Testing reporters with inverted promoters as well as testing other promoter combinations and configurations would be informative for building synthetic constructs where genes are in close proximity.

We were surprised to discover that the cHS4-based insulators were inefficient at blocking the spreading of chromatin-mediated targeted silencing, since they are traditionally thought to prevent spreading of heterochromatin and are commonly used in synthetic biology constructs to protect them from heterochromatin encroachment. None of the insulator variants derived from the cHS4 region of the beta-globin locus prevented spreading of silencing after KRAB recruitment (*Figure 4*). However, these insulators did reduce background silencing of our reporters in CHO-K1 (*Figure 4—figure supplement 5*), consistent with results from classical transgene insulator assays (*Recillas-Targa et al., 2002*; *Pikaart et al., 1998*). We also showed that even a single cHS4 insulator can stop transcriptional run-on (*Figure 1—figure supplement 4*), consistent with previous reports (*Tian and Andreadis, 2009*). We can explain the function of insulators on spreading and background silencing by thinking of them as areas in an active chromatin state, that can for example recruit additional writers of acetylation (*Zhao and Dean, 2004*), and other properties similar to promoters (*Raab and Kamakaka, 2010*). It has been shown that the core region of the HS4 insulator has high levels of acetylation (*Mutskov et al., 2002*; *Zhao and Dean, 2004*), and it has been previously proposed that the HS4 insulator protects from spreading of H3K9me3 through maintenance of high levels of acetylation (*Mutskov et al., 2002*; *Litt et al., 2001*; *Ghirlando et al., 2012*). A recent study (*Rudina and Smolke, 2019*) used ChIP-seq ENCODE data of histone modifications, including the presence of active acetylation marks and absence of repressive methylation marks, to identify candidate insulator sequences which

were then screened for their ability to prevent background transgene silencing. Additionally, when these insulators were tested in the presence of an HDAC inhibitor, the expression of the insulated transgene increased (*Rudina and Smolke, 2019*).

Previous models have proposed the idea that insulators, similar to enhancers and transcription factors, are anti-silencers which oppose repressive forces (*Fourel et al., 2004*), and transcriptional activators have been shown to insulate against heterochromatin and even drive reactivation (*Sutter et al., 2003*). Our findings are in line with these models, and we further propose that active regions, such as insulators and promoters, cooperate with each other to inhibit gene silencing and drive gene reactivation. According to this model, insulators fail when the gene reactivation driven by insulators is slower than the KRAB-induced repression and spreading of silencing. However, insulators can work against a much slower silencing process such as background silencing. In other words, at these length scales, the process of 'insulation' is better thought of as a dynamic fight between activation and repression. Screening different sequences with insulating function, such as Matrix Attachment Regions (MARs) and ubiquitous chromatin opening elements (UCOEs) (*Rudina and Smolke, 2019*), along with other insulators such as H19-Igf2 and SNS (*Di Simone et al., 2001*; *Kaffer et al., 2000*) in a similar recruitment assay in the future could help go beyond the binary classification of sequences as insulators or non-insulators and help specify and quantify how well they perform against different mechanisms of silencing. This process would also help identify more reliable insulators for synthetic biology. Additionally, it would also be interesting to test enhancers between the two reporter genes to study whether they can also block spreading of silencing to the downstream gene.

By monitoring gene reactivation dynamics after release of the CRs, we conclude that activation can also spread to nearby genes in a distance-dependent manner. However, which gene reactivates first depends not only on the distance between the genes, but also on their promoter size and strength and overall reactivation propensity of the locus. These results led us to modeling promoters and insulators as regions with increased rates of reactivation. In this model, acetylation from strong promoters, such as pEF, can spread via looping and positive feedback to the downstream weaker pRSV promoter, as seen in CHO-K1 cells. This model can be used in other contexts in which spreading of active modifications is relevant, such as enhancer-promoter interactions, activation of nearby endogenous genes after epigenetic editing and perhaps in understanding aging where loss of heterochromatin can lead to changes in gene expression (*Villeponteau, 1997*) or in diseases such as facioscapulohumeral dystrophy (FSHD). FSHD is a neuromuscular disorder that is potentially caused by loss of spreading of heterochromatin causing aberrant gene activation (*Kleinjan and van Heyningen, 1998*; *Gabellini et al., 2002*; *Hahn et al., 2010*).

Together, our experimental results and the model based on them have broad implications for understanding chromatin mediated gene regulation and building mammalian synthetic biology applications. They suggest that genes in close proximity (< 5 – 10 kb) in a synthetic gene system can respond to changes in chromatin regulators recruited at the promoter of one gene in a coordinated manner, during both silencing and activation, similar to endogenous gene clusters where neighboring genes tend to be co-regulated, including cell cycle, circadian rhythm, and housekeeping genes (*Cho et al., 1998*; *Cohen et al., 2000*; *Boutanaev et al., 2002*; *Ueda et al., 2002*; *Lercher et al., 2002*). In our model, promoters, enhancers, and insulators can be represented simply as regions with increased rates of reactivation. It would be interesting to extend the system to scenarios when these regulatory regions are farther apart in linear space but nevertheless close in 3D-space, as is the case of many promoters and their corresponding enhancers or silencers. In terms of synthetic biology applications, this gene coupling can be detrimental when we want to deliver compact circuits of genes that need to be controlled independently. However, the length-dependent time delay in gene response can also be used to build more sophisticated temporal population responses. Finally, our experimental results and theoretical framework highlight the need for further measuring and modeling the dynamics of gene expression after targeting various epigenetic editors at endogenous loci systematically (for example, with KRAB-based tools such as

CRISPRi) in order to determine the time necessary for efficient on-target effects without unwanted off-target spreading.

# Materials and methods

## Key resources table

| Reagent type (species) or resource | Designation | Source or reference | Identifiers | Additional information |
|---|---|---|---|---|
| Cell line (*Cricetulus griseus*) | CHO-K1 (with Human artificial chromosome) | Oshimura Lab, *Yamaguchi et al., 2011* | | |
| Cell line (*Homo sapiens*) | K562 (lymphoblast, adult leukemia) | ATCC | CCL-243 | |
| Cell line (*Homo sapiens*) | HEK293T (epithelial, embryo kidney) | ATCC | CRL-3216 | |
| Recombinant DNA reagent | pSL007_PB_pCMV-H2B-mIFP-T2A-rTetR-ratKRAB-zeo (plasmid) | This paper | RRID: Addgene_179438 | Piggybac rTetR-CR: rat KRAB |
| Recombinant DNA reagent | pLB62_PB_pGK-H2B-mIFP-T2A-rTetR-humanKRAB-zeo (plasmid) | This paper | RRID: Addgene_179439 | Piggybac rTetR-CR: human KRAB |
| Recombinant DNA reagent | pLB37_PB_pGK-H2B-mIFP-T2A-rTetR-HDAC4-zeo (plasmid) | This paper | RRID: Addgene_179440 | Piggybac rTetR-CR: human HDAC4 |
| Recombinant DNA reagent | pSL006_phiC31_neo-5xTetO-pEF-H2B-mCitrine-pRSV-H2B-mCherry (plasmid) | This paper | RRID: Addgene_179425 | NS reporter phiC31 |
| Recombinant DNA reagent | pMH010_phiC31_neo-5xTetO-pEF-H2B-mCitrine-1.2kb,lambda-pRSV-H2B-mCherry (plasmid) | This paper | RRID: Addgene_179427 | 1.2 kb reporter phiC31 |
| Recombinant DNA reagent | pSL008_phiC31_neo-5xTetO-pEF-H2B-mCitrine-5kb,lambda-pRSV-H2B-mCherry (plasmid) | This paper | RRID: Addgene_179426 | 5 kb reporter phiC31 |
| Recombinant DNA reagent | pSL011_AAVS1-puro-9xTetO-pEF-H2B-mCitrine-pRSV-H2B-mCherry (plasmid) | This paper | RRID: Addgene_179428 | NS reporter AAVS1 |
| Recombinant DNA reagent | pSL012_AAVS1-puro-9xTetO-pEF-H2B-mCitrine-5kb,lambda-pRSV-H2B-mCherry (plasmid) | This paper | RRID: Addgene_179429 | 5 kb reporter AAVS1 |
| Recombinant DNA reagent | pMH002_phiC31_neo-5xTetO-pEF-H2B-mCitrine-HS4-pRSV-H2B-mCherry (plasmid) | This paper | RRID: Addgene_179430 | Single cHS4 (SH) reporter phiC31 |
| Recombinant DNA reagent | pMH003_phiC31_neo-5xTetO-pEF-H2B-mCitrine-coreHS4-pRSV-H2B-mCherry (plasmid) | This paper | RRID: Addgene_179431 | Single core cHS4 (SC) reporter AAVS1 phiC31 |
| Recombinant DNA reagent | pMH004_phiC31_neo-HS4-5xTetO-pEF-H2B-mCitrine-HS4-pRSV-H2B-mCherry (plasmid) | This paper | RRID: Addgene_179432 | Double cHS4 (DH) reporter phiC31 |
| Recombinant DNA reagent | pMH005_phiC31_neo-core-5xTetO-pEF-H2B-mCitrine-core-pRSV-H2B-mCherry (plasmid) | This paper | RRID: Addgene_179433 | Double core cHS4 (DC) reporter phiC31 |
| Recombinant DNA reagent | pMH011_AAVS1_puro-pA-9xTetO-pEF-H2B-mCitrine-HS4-pRSV-H2B-mCherry (plasmid) | This paper | RRID: Addgene_179434 | Single cHS4 (SH) reporter AAVS1 |
| Recombinant DNA reagent | pMH012_AAVS1_puro-pA-9xTetO-pEF-H2B-mCitrine-core-pRSV-H2B-mCherry (plasmid) | This paper | RRID: Addgene_179435 | Single core cHS4 (SC) reporter AAVS1 |
| Recombinant DNA reagent | pMH014_AAVS1_puro-pA-HS4-9xTetO-pEF-H2B-mCitrine-HS4-pRSV-H2B-mCherry (plasmid) | This paper | RRID: Addgene_179436 | Double cHS4 (DH) reporter AAVS1 |
| Recombinant DNA reagent | pMH015_AAVS1_puro-pA-core-9xTetO-pEF-H2B-mCitrine-core-pRSV-H2B-mCherry (plasmid) | This paper | RRID: Addgene_179437 | Double core cHS4 (DC) reporter AAVS1 |
| Antibody | Anti-rabbit IgG H&L chain (guinea pig, polyclonal) | Antibodies-Online | Cat #ABIN101961 | CUT&RUN (1:100) |
| Antibody | Anti-H3ac (rabbit, polyclonal) | Active Motif | Cat #39,139 | CUT&RUN (1:100) |
| Antibody | Anti-H3K4me3 (rabbit, polyclonal) | Active Motif | Cat #39,159 | CUT&RUN (1:100) |
| Antibody | Anti-H3K27me3 (rabbit, monoclonal) | Cell Signaling | Cat #9,733 S | CUT&RUN (1:50) |
| Peptide, recombinant protein | pA-MNase | Henikoff Lab | | |
| Commercial assay or kit | CUTANA ChIC/CUT&RUN Kit | EpiCypher | Cat #14 – 1048 | CUT&RUN |

*Continued on next page*

Continued

| Reagent type (species) or resource | Designation | Source or reference | Identifiers | Additional information |
|---|---|---|---|---|
| Chemical compound, drug | Doxycycline | Tocris Bioscience | 4,090 | |
| Chemical compound, drug | Tazemetostat-6438 | Selleck Chemicals | S7128 | EZH2 inhibitor |
| Software, algorithm | MACKtrack | *Taylor, 2019* | | https://github.com/brookstaylorjr/MACKtrack |

## Plasmid construction

The CHO-K1 PhiC31 reporters (*Figure 1—figure supplement 1A*) were assembled as follows: First, a PhiC31-Neo-5xTetO-pEF-H2B-mCitrine reporter construct was assembled using a backbone containing the PhiC31 attB site, a neomycin resistance gene, and a multiple cloning site (*Yamaguchi et al., 2011*). The elements of the reporter constructs were PCR amplified from the following sources: five Tet binding sites from the TRE-tight plasmid (Clontech), pEF from pEF/FRT/V5-Dest (Life Technologies), and H2B- mCitrine from pEV2-12xCSL-H2B-mCitrine (*Sprinzak et al., 2010*). These components were first sequentially cloned into the pExchange1 backbone using standard molecular biology techniques. The entire TRE-pEF-H2B-mCitrine was then PCR-amplified and combined by Gibson assembly with the phiC31-Neomycin-MCS backbone cut by AvrII. This construct was designed such that after integration, the neomycin gene would be expressed from a PGK promoter situated upstream of the phiC31 site in the MI-HAC (*Yamaguchi et al., 2011*). The second fluorescent reporter was added by digesting the mCitrine only plasmid with NdeI and adding: the pRSV-H2B from R4-blast-pRSV-H2B- mTurquoise (gift from Teruel Lab), mCherry from pEx1-pEF-H2B-mCherry-T2A-rTetR-EED (Addgene #78101), and polyA from PhiC31-Neo-ins-5xTetO-pEF-H2B-Citrine-ins (Addgene #78099) using Gibson Assembly to generate the NS construct. The lambda spacers were amplified from lambda phage DNA (NEB, N3011) and inserted via Gibson Assembly after digesting the NS reporter plasmid with Bsmb1. Similarly, the full-length cHS4 was amplified from PhiC31-Neo-ins-5xTetO-pEF-H2B-Citrine-ins (Addgene #78099) or the core insulator was amplified from PB CMV-MCS-EF1α-Puro PiggyBac vector backbone (System Biosciences #PB510B-1), and inserted into the NS reporter with Gibson assembly after digestion with BsmBI for insulators between mCitrine and mCherry, and BsiWI (restriction digestion site added by site directed mutagenesis) for the insulators upstream of TRE. The PhiC31 integrase was a gift from the Oshimura Lab (*Yamaguchi et al., 2011*).

The K562 and HEK293T AAVS1 reporter constructs (*Figure 1—figure supplement 1B*) were assembled as follows: First, a 9xTetO-pEF-H2B-Citrine reporter was cloned into a AAVS1 donor vector backbone (Addgene #22212) containing a promoter-less splice-acceptor upstream of a puromycin resistance gene and homology arms against the AAVS1 locus. The elements of the reporter were amplified from the following sources: the 9XTetO sites were ordered from IDT, and the pEF-H2B-mCitrine was PCR amplified from the PhiC31 construct. These components were cloned into the AAVS1 donor vector backbone using Gibson Assembly. The mCherry components, spacers and insulators were added to the mCitrine only base plasmid in the same manner as the phiC31 reporters (see above), except for insulators where plasmids were digested with BstBI and MluI-HF.

The Piggybac plasmids containing the rTetR-CR were assembled into the PB CMV-MCS- EF1α-Puro PiggyBac vector backbone (System Biosciences #PB510B-1), which was modified via Gibson Assembly to add H2B-rTetR-Zeo from pEx1-pEF-H2B-mCherry-T2A-rTetR-KRAB- Zeo (Addgene #78352), mIFP from pSLQ2837-1, and for K562 plasmids the pGK promoter from pSLQ2818 (the latter two gifted from Tony Gao & Stanley Qi, Stanford). The CRs used were: rat KRAB from pEx1-pEF-H2B-mCherry-T2A-rTetR-KRAB (Addgene #78348) for CHO-K1, human KRAB ZNF10 from pSLQ2815 CAG-Puro-WPRE_PGK-KRAB-tagBFP-dCas9 (gifted from Tony Gao & Stanley Qi, Stanford) for K562, and human HDAC4 from pEx1-pEF-H2B-mCherry- T2A-rTetR-HDAC4 (Addgene #78349) for both CHO-K1 and K562.

All plasmids used in this study have been deposited to Addgene (see Key Resources Table).

## Cell line construction

The CHO-K1 cell line with the human artificial chromosome (*Yamaguchi et al., 2011*) was a gift from Mitsuo Oshimura's lab. It was profiled for the presence of the HAC by genomic PCR and also had the correct morphology by microscopy. The K562 and HEK293T human cell lines were from ATCC, and were authenticated by STR profiling as a match for ATCC lines CCL-243 and CRL-11268, respectively. We did regular mycoplasma testing three times per year and did not detect the presence of mycoplasma in any of our cell lines.

Reporter lines in CHO-K1 cells were created by integrating the reporter plasmids at the phiC31 integration site on the MI-HAC (human artificial chromosome, gifted by Oshimura Lab) by co-transfecting 750 ng of reporter plasmid with 250 ng of the phiC31 integrase using Lipofectamine 2000 (Invitrogen, 11668027). Cells were plated 24 hr before transfection and media change was performed 12 hr after transfection. Selection was started 48–72 hr after transfection with 600 ng/mL geneticin (Gibco, 10131027) for 1–2 weeks. CRs were integrated randomly with the Piggybac system (System Biosystems, PB210PA-1) by co-transfecting 750 ng of CR plasmid and 250 ng PiggyBac transposase with Lipofectamine 2000. Cells were selected with 400 ng/mL Zeocin starting 24 hr or later after transfection for 1–2 weeks. Single clones were isolated for each lambda reporter, and correct integration into the MI-HAC was verified by PCR of genomic DNA. Note that rat KRAB was used for CHO-K1cells and human KRAB ZFN10 was used for K562; both are driven by CMV promoter and have mIFP as a fluorescent marker followed by T2A before the rTetR-CR fusion.

Reporters were integrated in K562 cells at the AAVS1 safe harbor site using TALENs: AAVS1-TALEN-L (Addgene #35431) targeting 5'-TGTCCCCTCCACCCCACA-3' and AAVS1-TALEN-R (Addgene #35432) targeting 5'-TTTCTGTCACCAATCCTG-3' (*Sanjana et al., 2012*). Roughly 1.2M K562 cells mixed with 5000 ng of reporter plasmid and 1000 ng of each left and right TALENS in a nucleofection cuvette (Mirus Bio, 50121), were transfected by nucleofection (Lonza 2B Nucleofector) with program T-16. Selection was started 48–72 hr after transfection with 3 µg/mL Puromycin (Invivo Gen, ant-pr) for 1–2 weeks. We performed genomic PCR on K562 multiclonal populations to confirm the presence of proper integration at the AAVS1 site, using primers from *Oceguera-Yanez et al., 2016*. CRs were integrated randomly with the Piggybac system by nucleofecting 1000 ng of CR plasmid and 300 ng PiggyBac. Cells were selected with 400 ng/mL Zeocin starting 24 hr or later after transfection for 1–2 weeks.

Both CHO-K1 and K562 reporter cell lines with CRs were sorted (Sony SH800) for triple positive fluorescence: mCitrine and mCherry of the dual gene reporter and mIFP transcribed along with the CR.

## Cell culture conditions

Cells were cultured at 37°C in a humidified incubator (Panasonic MCO-230AICUVL) with 5% $CO_2$. CHO-K1 cells were grown in Alpha MEM Earle's Salts media with 10% Tet Approved FBS (Takara Bio 631367, Lots # A16039 & #17033) and 1 X Penicillin/Streptomycin/L-glutamine (Gibco 10378016). CHO-K1 cells were passaged by rinsing with Dulbecco's Phosphate-Buffered Saline (DPBS, Gibco 14190250), and incubating at room temperature with 0.25% Trypsin (Gibco, 25200056). K562 cells were grown in RPMI 1640 medium (Gibco, 11875119) with 10% Tet Approved FBS (Takara Bio 631367, Lot #17033) and 1 X Penicillin/Streptomycin/L-glutamine (Gibco 10378016). For long-term storage, cells were frozen in growth media with 10% DMSO (Sigma Aldrich, D2650) and placed at – 80°C (for up to a month), and then transferred to liquid nitrogen for long term storage.

## Flow cytometry of recruitment and release assays

During recruitment assays, doxycycline (Tocris Bioscience, 4090) was added to the media to a final concentration of 1000 ng/mL unless otherwise stated. Fresh dox diluted from frozen aliquots was added to the media before each cell passage during recruitment, every 2 – 3 days. Flow cytometry data was collected on two different flow cytometers due to machine access: CytoFLEX S (Beckman Coulter) and ZE5 Cell Analyzer (BioRad). Cells were filtered through 40 µm strainers to remove clumps before flow, and 10,000 – 30,000 cells were collected for each time point. For CHO-K1 NS and 5 kb clonal lines were used for replicates; for the remaining multiclonal constructs, data for each time course was collected from experiments started on different days for use as biological replicates. For EZH2 inhibitor experiments, Tazemetostat-6438 (Selleck Chemicals, S7128) was dissolved in DMSO

(Sigma-Aldrich, D2650) at 1 mM and was added to media at final concentration of 5 µM. Data was analyzed using a Matlab program called EasyFlow (https://antebilab.github.io/easyflow/) and a Python-based package, Cytoflow (https://cytoflow.github.io/). All cells were gated for mIFP positive cells based on fitting the background fluorescence of wildtype cells to a sum of Gaussians, and setting the threshold 2 standard deviations away from the mean of the main peak. The percentages of cells with mCitrine and mCherry active (*Figure 1*) were determined using a manual threshold based on the no dox sample. To quantify the percent of cells with mCitrine or mCherry ON during reactivation (*Figure 5* and *Figure 5—figure supplements 1–3*), we fit the background fluorescence of wildtype cells in each fluorescent channel to a sum of Gaussians and set the ON threshold 3 standard deviations away from the mean of the main peak.

## CUT&RUN experiments

CUT&RUN for the K562 KRAB and CHO-K1 KRAB cell lines was performed according to the third version of the published protocol on protocols.io (*Janssens and Henikoff, 2019*). For each antibody condition, 250,000 K562 or CHO-K1 cells were harvested and incubated with 10 uL of activated Concanavalin A beads. Cells were permeabilized with 0.05% digitonin and incubated at room temperature for 2 hr with a 1:100 dilution of one of the following antibodies: guinea pig anti-rabbit IgG H&L chain (Antibodies-Online; Cat. No.: ABIN101961); rabbit anti-H3ac (Active Motif; Cat. No.: 39139); rabbit anti-H3K4me3 (Active Motif; Cat. No.: 39159); rabbit anti-H3K9me3 (Abcam; Cat. No.: ab8898). Unbound antibody was washed out, and cells were incubated with 140 ng/mL of a protein A and micrococcal nuclease fusion (pA-MNase; generously provided by the Henikoff lab). Unbound pA-MNase was washed out, and cells were resuspended in a low-salt buffer. Tubes were placed in a thermal block chilled to 0°C, and 10 mM $CaCl_2$ was added to trigger DNA cleavage by pA-MNase. After 5 min, the reaction was stopped by chelating calcium ions with the addition of 20 mM EGTA, and tubes were incubated at 37°C for 30 min under physiological salt conditions to promote diffusion of fragmented chromatin into the supernatant. Fragmented chromatin in the supernatant was collected and purified using the DNA Clean & Concentrator-5 kit (Zymo; Cat. No.: D4004). CUT&RUN for the CHO-K1 HDAC4 cell lines was performed using the EpiCypher CUTANA ChIC/CUT&RUN Kit (Cat. No.: 14 – 1048) according to the User Manual Version 2.0 with the following conditions: an input of 500,000 CHO-K1 cells per antibody condition; a final digitonin concentration of 0.01%; an overnight antibody incubation at 4°C with 1:100 dilutions of the anti-H3Kac and anti-H3K9me3 antibodies described above along with a 1:50 dilution of rabbit anti-H3K27me3 (Cell Signaling Technologies, Cat. No: 9,733 S); and DNA purification with the provided spin columns. Dual-indexed sequencing libraries were prepared using the NEBNext Ultra II DNA Library Prep Kit for Illumina (New England Biolabs; Cat. No.: E7645L) with 12 – 14 cycles of PCR, and size selection was performed with either Agencourt AMPure XP (Beckman Coulter; Cat. No.: A63880) or SPRIselect (Beckman Coulter; Cat. No.: B23318) magnetic beads. Library concentrations were quantified with the Qubit 1 X dsDNA HS Assay Kit (Invitrogen; Cat. No.: Q33231), and library fragment sizes were assessed with the High Sensitivity D1000 ScreenTape (Agilent; Cat. No.: 5067 – 5584) and High Sensitivity D1000 Reagent (Agilent; Cat. No.: 5067 – 5585) on an Agilent 4,200 TapeStation System. Libraries were pooled for genome-wide sequencing.

## CUT&RUN sequencing, data processing, and data analysis

Genome-wide libraries were paired-end sequenced by the Stanford Center for Genomics and Personalized Medicine on a HiSeq 4000 with 2 × 101 cycles. A custom genome with our reporter sequence appended to the end of the hg19 human genome assembly was constructed with bowtie2-build. Paired-end alignment was performed with the following bowtie2 command: bowtie2 --local --very-sensitive-local --no-unal --no-mixed --no-discordant --phred33 -I 10 X 700 x {reference genome} -p 8 – 1 {first mate of pair} – 2 {second mate of pair} -S {output SAM file name}. Fragments that mapped completely within non-unique reporter elements (i.e. pEF, H2B, PGK, AmpR, or SV40 polyA) were ambiguous and thus removed to avoid confounding. Samtools was used to convert SAM files to BAM files and to subsequently sort and index BAM files. Picard was used to mark and remove duplicates with the following command: java -jar {picard tool} MarkDuplicates -I {input sorted BAM file} -O {output deduplicated BAM file} -M {output metrics file} --REMOVE_DUPLICATES true. Reads were normalized to counts per million (CPM) and bedgraph files were generated with the following bamCoverage

command: bamCoverage --bam {input deduplicated BAM file} -o {output bedgraph file} --outFile-Format bedgraph --extendReads --centerReads --binSize 10 --normalizeUsing CPM. CUT&RUN data were visualized via the Broad Institute and UC San Diego Integrative Genomics Viewer. Additional data analyses and visualization were performed with custom scripts in Python.

ChromaBlocks analysis was performed in R using the Repitools package. Normalized bedgraph files were read in as dataframes and converted to GRangesList objects with the annoDF2GR() and GRangesList() methods. GRangesList objects for H3K9me3, H3Kac, or H3K4me3 samples were specified for the rs.ip argument within the ChromaBlocks() method, and the respective dox-treated or untreated IgG sample was specified as the rs.input argument. ChromaBlocks() was called with the 'large' preset for H3K9me3 and H3Kac and with the 'small' preset for H3K4me3 to identify regions of enrichment. Start coordinates and widths of enriched regions were used to construct bed files for visualization in IGV and for signal integration to compute the $\log_2$ fold-change in *Figure 2—figure supplement 1E*.

## qPCR and PCR on cDNA

Each cell line was treated with dox (1 ug/mL) for the indicated number of days before harvesting. RNA was harvested using RNeasy Mini Kit (Qiagen, 74106) using the on-column RNase-Free DNase (Qiagen, 79254). cDNA was synthesized using the iScript Reverse Transcription Supermix (Biorad, 1708840) for qPCR experiments and qScript cDNA SuperMix (VWR, 101414 – 106) for PCR experiments. qPCR on cDNA was performed using SsoFast EvaGreen Supermix on a CFX Connect Real-Time PCR System (both from Bio-Rad Laboratories). For qPCR primer sequences, see *Appendix 2—table 1*. The values are reported as delta Ct to Beta-actin. PCR on cDNA was performed using Q5 Hot Start High-Fidelity 2 X Master Mix (NEB, M0494L) on a C1000 Touch Thermal Cycler from Bio-Rad. PCR products were run on a 1–1.5% TAE agarose gel at room temperature and imaged using Biorad Gel Doc EZ Imager. For PCR primer sequences, see Appendix 2-Table 1.

## Time-lapse microscopy

Fluorescent time-lapse imaging was conducted using a DMi8 inverted epifluorescence microscope (Leica, Germany) with a 20 X Plan Apo 0.8 NA objective and a Leica DFC9000 GT CMOS Camera using 2 × 2 binning. The microscope was enclosed in a cage incubator (Okolab Bold Line, custom for DMi8) held at 37 °C, and samples were placed in a stage top chamber (Okolab H301-K-FRAME) also held at 37°C, 5 % $CO_2$ (Praxair Cat. BI NICD5O6B-K), and with humidity control. CHO-K1 cells were seeded 24 hr prior to the start of movie acquisition on optically clear 96-well μ-Plates (Ibidi Cat. 89626). Border wells were filled with PBS (Thermo Fisher Scientific Cat. 10010023) and plates were sealed with Breath-Easy gas-permeable membranes (Diversified Biotech Cat. BEM-1) to reduce evaporation and maintain humidity during image acquisition. Cells were imaged in alpha MEM media with no phenol red (Thermo Fisher, Cat. 41061029) with 10% Tet-Free FBS (Takara Cat. 631106; Lot #A17033), and 1% Penicillin/Streptomycin/Glutamine (Fisher Scientific Cat. 10378 – 016) to reduce background fluorescence. Images were acquired every 20 min in three different fluorescent channels: YFP, RFP and CY5, corresponding to the mCitrine, mCherry and mIFP fluorophores, respectively. The light source was a Sola Light Engine, and a Lumencor control pod was used to attenuate the light to < 1 mW range. The FIM was set to 10% and Lumencore was set to 5%. Exposure times were 150 ms, 200 ms, and 100 ms for the CY5, RFP and YFP channels, respectively (an additional trigger cable between the light source and microscope was necessary to achieve precise exposure times). One to two time-lapse movies were analyzed for each cell line, where each movie consisted of cells in dozens of different wells, and in each well 3 – 5 non-overlapping sites were imaged. Cell culture media was changed every 24 hr to replenish doxycycline and remove toxic reactive oxygen species which may have formed due to prolonged imaging. LasX software was used to control the microscope in the Mark and Find module to image up to 200 positions in total and AFC on demand mode adjusted focus for every cycle and position of imaging.

## Analysis of time-lapse movies

Analysis of all time lapse movies was conducted in MATLAB R2016a (MathWorks) unless otherwise stated. Fluorescent imaging data were obtained by automated image segmentation and single-cell tracking using the MACKtrack package (https://github.com/brookstaylorjr/MACKtrack). Nuclei were segmented and tracked using H2B-mIFP signal or H2B-mCherry signal (on a case-by-case basis based

on expression level) and single-cell intensities for integrated nuclear intensity of mCitrine, mCherry, and mIFP channels were measured at each frame. Tracked single-cell output traces were then filtered in a semi-automated fashion to remove incomplete or mistracked cells which were defined as traces exhibiting numerous aberrant drops in fluorescence intensity inconsistent with cell divisions. Additionally, traces consisting of background silenced cells with low mCitrine expression prior to the 24-hr time point were omitted from analysis. In order to call transcriptional silencing events, the gradient of the cumulative intensity traces for mCherry and mCitrine were used. To generate cumulative traces, mitotic events were first computationally identified based on periodic dilution of the stable H2B-fluorophore signal by approximately 50% upon cell division. Using these points as reference, raw cumulative traces for mCherry and mCitrine were generated by computationally re-adding the lost fluorescence due to cell division events and adding these values to all subsequent time points after a division event to 'stitch together' intensity values across divisions (see *Figure 3—figure supplement 1*). The raw stitched traces were smoothed to obtain smoothed stitched traces by applying: first a moving average filter ('smoothrows' function, Matlab) with a window size of 2 frames, then a 'smoothingspline' ('fit' function,Matlab) with the smoothing parameter set to 1.0e-04, and finally a median filter ('medfilt1' function, Matlab) with a window-size of 3 frames. The gradient of these smoothed stitched traces were used to call silencing when it fell below a threshold. mCitrine or mCherry silencing was called in individual cells at the first frame where the gradient of the smooth cumulative trace fell below a threshold (25% for KRAB and 30% for HDAC4) of its initial value (taken to be the maximum gradient of the same trace between frames 50 and 70 prior to dox addition). Furthermore, cells must have stably remained below that threshold through at least one more cell division to be considered silenced. To account for the spike in mCherry signal following mCitrine silencing in certain cells (*Figure 1—figure supplement 4*), the gradient threshold was called in these instances relative to a 10% drop from its absolute maxima which represented the spike directly prior to transcriptional silencing. Single-cell silencing delays were calculated by subtracting the frame number at which silencing was called in mCitrine from the corresponding silencing frame number of mCherry in the same cell and converting to hours.

## Statistical analysis of movie delay times

Welch's unequal variances T-test was performed on GraphPad Prism 8.4 on the calculated time delays from movies to determine significance (p-values) between chromatin-regulator spreading rates between NS and 5 kb reporters.

## Model for the time evolution of fluorescence distributions

We have developed a model that describes the time dependent evolution of the fluorescence distributions of our two fluorescent reporters after the recruitment of CRs. This model has two components: a deterministic component describing the decay of fluorescence due to mRNA and protein degradation and dilution, and a probabilistic component taking into account that cells can stochastically transition between the active and silent states at different times. In the deterministic component, we will derive equations that describe the fluorescence of cells in either the active or silent state, beginning with the simpler upstream *mCitrine* gene, followed by the downstream *mCherry* gene. The deterministic equations are then used in the probabilistic component to derive a probability density function that describes flow cytometry data. By deriving this model and fitting it to our data, we estimate the silencing parameters of our system such as the time delay between reporter silencing and the magnitude of transcriptional interference.

### Deterministic component with mRNA and protein degradation and dilution

In this component of the model, we derive deterministic equations for the fluorescence of cells in the active and silent state. First, we will derive these equations for the simpler upstream *mCitrine* gene. We begin at the level of transcription where *m* represents the concentration of reporter mRNA. The recruitment of the CR begins at *t=0* with the addition of doxycycline, but transcriptional silencing of *mCitrine* does not occur until some time later, $t_{s,C}$. By assuming a constant mRNA production rate, $\alpha_m$, and a constant coefficient of mRNA degradation and dilution, $\beta_m$, we can write a piecewise differential equation for mRNA concentration:

$$\frac{dm}{dt} = \begin{cases} \alpha_m - \beta_m m & , t \le t_{S,C} \\ -\beta_m m & , t > t_{S,C} \end{cases} \tag{1}$$

This differential equation describes transcription of the upstream *mCitrine* gene. We assume a steady-state initial condition and continuity among the pieces to arrive at the following solution for *mCitrine* mRNA as a function of time:

$$\mathrm{m}(t) = \begin{cases} \frac{\alpha_m}{\beta_m} & , t \le t_{S,C} \\ \frac{\alpha_m}{\beta_m} e^{-\beta_m(t-t_{S,C})} & , t > t_{S,C} \end{cases} \tag{2}$$

Next, we move on to the level of translation, where mRNA is translated into protein. The protein concentration is represented by $p$. Here we assume a constant coefficient of protein production, $\alpha_p$, and a constant coefficient of protein degradation and dilution, $\beta_p$. These assumptions allow us to write the following general differential equation to represent translation:

$$\frac{dp}{dt} = \alpha_p m - \beta_p p \tag{3}$$

Substituting our solution for *m(t)* into the equation above and combining $\alpha_p$ and $\alpha_m$ into a single integrated production parameter such that $\alpha = \alpha_p\,\alpha_m$, we arrive at the following piecewise differential equation for the concentration of the mCitrine protein over time:

$$\frac{dp}{dt} = \begin{cases} \frac{\alpha}{\beta_m} - \beta_p p & , t \le t_{S,C} \\ \frac{\alpha}{\beta_m} e^{-\beta_m(t-t_{S,C})} - \beta_p p & , t > t_{S,C} \end{cases} \tag{4}$$

Again, we assume a steady-state initial condition and continuity among the pieces to arrive at the following solution for mCitrine concentration:

$$\mathrm{p}(t) = \begin{cases} A & , t \le t_{S,C} \\ Be^{-\beta_p(t-t_{S,C})} - Ce^{-\beta_m(t-t_{S,C})} & , t > t_{S,C} \end{cases} \tag{5}$$

here

$$A = \frac{\alpha}{\beta_p \beta_m} \quad B = \frac{\alpha}{\beta_p \beta_m - \beta_p^2} \quad C = \frac{\alpha}{\beta_m^2 - \beta_p \beta_m} \tag{6}$$

Now that we have derived equations for the concentration of protein over time, we need to convert from concentration to fluorescence, represented by *x* in our model. We assume that fluorescence increases proportionally to concentration after the addition of some background fluorescence $x_B$. The proportionality constant is absorbed into the production parameter, α. Therefore, fluorescence can simply be expressed as protein concentration with the addition of the background fluorescence in Arbitrary Fluorescence Units (AFU):

$$x(t) = x_B + p(t) \tag{7}$$

Using this relationship and the solution to $p(t)$, we define two separate equations for mCitrine fluorescence, one for the active state where $t \le t_{S,C}$ and one for the silent state where $t > t_{S,C}$:

$$x_A(t) = x_B + A \tag{8}$$

$$x_S(t, t_{S,C}) = x_B + Be^{-\beta_P(t-t_{S,C})} - Ce^{-\beta_m(t-t_{S,C})} \tag{9}$$

These two equations describe the fluorescence for the upstream *mCitrine* gene, but they are too simple to describe silencing at the downstream *mCherry* gene, as we often observe an increase in mCherry after the recruitment of the CR. We hypothesize that this upwards spike in mCherry production is due to the loss of transcriptional interference upon silencing of the upstream *mCitrine* gene. Therefore, we modified the differential equation for mRNA with the addition of a new term, γ, which represents the factor by which mRNA production is reduced by transcriptional interference. This factor

is removed after the average silencing time of the *mCitrine* gene, $t_{S,C}$, causing a delayed spike in the mRNA concentration of *mCherry* before it silences and begins to degrade at its own $t_{S,M}$:

$$\frac{dm}{dt} = \begin{cases} \frac{\alpha_m}{\gamma} - \beta_m m & , t \leq t_{S,C} \\ \alpha_m - \beta_m m & , t_{S,C} < t \leq t_{S,M} \\ -\beta_m m & , t > t_{S,M} \end{cases} \tag{10}$$

To solve this differential equation, we again assume a steady-state initial condition and continuity among the pieces. The solution for *mCherry* mRNA over time is:

$$m(t) = \begin{cases} \frac{\alpha_m}{\beta_m \gamma} & , t \leq t_{S,C} \\ \frac{\alpha_m}{\beta_m} \left[ 1 - \Gamma e^{-\beta_m (t - t_{S,C})} \right] & , t_{S,C} < t < t_{S,M} \\ \frac{\alpha_m}{\beta_m} \left[ 1 - \Gamma e^{-\beta_m (t_{S,M} - t_{S,C})} \right] e^{-\beta_m (t - t_{S,M})} & , t > t_{S,M} \end{cases} \tag{11}$$

where

$$\Gamma = 1 - \frac{1}{\gamma} \tag{12}$$

Substituting this solution for *m(t)* into the general translation equation and again combining $\alpha$'s into an integrated production parameter $\alpha$, we arrive at a piecewise differential equation for the concentration of mCherry over time:

$$\frac{dp}{dt} = \begin{cases} \frac{\alpha}{\beta_m \gamma} - \beta_p p & , t \leq t_{S,C} \\ \frac{\alpha}{\beta_m} \left[ 1 - \Gamma e^{-\beta_m (t - t_{S,C})} \right] - \beta_p p & , t_{S,C} < t < t_{S,M} \\ \frac{\alpha}{\beta_m} \left[ 1 - \Gamma e^{-\beta_m (t_{S,M} - t_{S,C})} \right] e^{-\beta_m (t - t_{S,M})} - \beta_p p & , t > t_{S,M} \end{cases} \tag{13}$$

Again, we assume a steady-state initial condition and continuity among the pieces to arrive at the following solution for *mCherry* concentration over time:

$$p(t) = \begin{cases} \frac{A}{\gamma} & , t \leq t_{S,C} \\ A - B\Gamma e^{-\beta_p (t - t_{S,C})} + C\Gamma e^{-\beta_m (t - t_{S,C})} & , t_{S,C} < t \leq t_{S,M} \\ \left[ D + CE \right] e^{-\beta_p (t - t_{S,M})} - CE e^{-\beta_m (t - t_{S,M})} & , t > t_{S,M} \end{cases} \tag{14}$$

where

$$D = A - B\Gamma e^{-\beta_p (t_{S,M} - t_{S,C})} + C\Gamma e^{-\beta_m (t_{S,M} - t_{S,C})} \tag{15}$$

$$E = 1 - \Gamma e^{-\beta_m (t_{S,M} - t_{S,C})} \tag{16}$$

While the previous equation for *p(t)* assumes the more likely scenario that the upstream *mCitrine* gene silences before the downstream *mCherry* gene ($t_{S,M} > t_{S,C}$), we must also consider that *mCherry* may silence before *mCitrine* ($t_{S,M} \leq t_{S,C}$). In this scenario, there is no spike of mCherry expression because the *mCherry* promoter silences without the loss of transcriptional interference concomitant with *mCitrine* silencing. Therefore, the following solution for mCherry concentration over time when mCherry silences first is the same as mCitrine, only divided by γ:

$$p(t) = \begin{cases} \frac{A}{\gamma} & , t \leq t_{S,M} \\ \frac{B}{\gamma} e^{-\beta_p (t - t_{S,M})} - \frac{C}{\gamma} e^{-\beta_m (t - t_{S,M})} & , t > t_{S,M} \end{cases} \tag{17}$$

Now that we have defined two equations of mCherry concentration, we again combine these equations with the fluorescence equation and split them such that there is one equation for the active state where $t \leq t_{S,M}$ and one equation for the silent state where $t > t_{S,M}$. The active state fluorescence combines the first two parts of *equation 14* with the first part of *equation 17* while the silent state fluorescence combines the third part of *equation 14* with the second part of *equation 17*. These

combinations reflect that in either the active or silent state, either reporter can silence first. The fluorescence equations for mCherry are:

$$x_A(t) = \begin{cases} x_B + \frac{A}{\gamma} & ,t \leq t_{S,C} \\ x_B + A - B\Gamma e^{-\beta_p(t-t_{S,C})} + C\Gamma e^{-\beta_m(t-t_{S,C})} & ,t > t_{S,C} \end{cases} \tag{18}$$

$$x_S(t, t_{S,M}) = \begin{cases} x_B + \frac{B}{\gamma}e^{-\beta_p(t-t_{S,M})} - \frac{C}{\gamma}e^{-\beta_m(t-t_{S,M})} & ,t_{S,M} \leq t_{S,C} \\ x_B + [D + CE]\, e^{-\beta_p(t-t_{S,M})} - CEe^{-\beta_m(t-t_{S,M})} & ,t_{S,M} > t_{S,C} \end{cases} \tag{19}$$

These two equations describe the downstream mCherry fluorescence at any given point in time and with any given $t_{S,C}$ and $t_{S,M}$. They will be useful in the next section of the model. The constant parameters of the model thus far are as follows:

- $\alpha$ - the integrated fluorescence production parameter in $AFU/day^2$
- $\beta_p$ - the protein degradation and dilution coefficient in $days^{-1}$
- $\beta_m$ - the mRNA degradation and dilution coefficient in $days^{-1}$
- $x_B$ - the background fluorescence in $AFU$
- $\gamma$ - the interference factor (only applicable for mCherry)

## Probabilistic component with stochastic transcriptional silencing

In the probabilistic component of the model, we include a stochastic transition between active and silent states in the derivation of a probability density function of fluorescence $x$ at time $t$, $f(x,t)$. The distribution evolves over time and describes the flow cytometry data that we collect over the course of our silencing experiments. This probability density function of fluorescence is a weighted sum of three independent probability density functions, one for each of three possible cell states: background silent cells, $f_B(x)$; active cells, $f_A(x,t)$; and silent cells, $f_S(x,t)$. The time dependence in active and silent cells reflects the time dependence in the deterministic equations describing fluorescence in the active and silent states. Furthermore, because we have observed that the transition from the active to the silent state is a stochastic process with variability between cells (**Bintu et al., 2016**), we incorporate a time dependence into the weights of active and silent cells, $P_A(t)$ and $P_S(t)$ respectively. The probability density function can hence be written:

$$f(x,t) = P_B\, f_B(x) + P_A(t)\, f_A(x,t) + P_S(t)\, f_S(x,t) \tag{20}$$

To consider all cells and for $f(x,t)$ to meet the definition of a probability density function, the weights of each state, represented by P's, must sum to 1. For simplicity, we assume that the fraction of background silent cells, $P_B$, does not significantly change over the silencing time course. Therefore, $P_B$ does not have a time dependence in the following relationship:

$$P_B + P_A(t) + P_S(t) = 1 \tag{21}$$

To determine the fraction of active cells and silent cells at any point in time, we consider that a cell transitions from an active state to a silent state after a time of $t_S$. We add stochasticity to this transition by allowing cells to transition at different times and defining random variable $T_S$. We choose a gamma distribution for $T_S$ as this distribution is defined to be strictly greater than zero and can assume various shapes between an exponential distribution with $k = 1$ and a normal distribution as $k$ approaches infinity. In the shape-rate parameterization of the gamma distribution, the rate, typically represented as $\theta$, can be expressed by dividing the mean, $\tau_S$, by the shape $k$. Therefore, we attain the following definition and probability density function for $T_S$:

$$T_S \sim \Gamma\left(k, \frac{\tau_S}{k}\right) \tag{22}$$

$$g(t_S) = \frac{k^k t_S^{k-1}}{\tau_S^k \Gamma(k)} e^{-\frac{kt_S}{\tau_S}} \tag{23}$$

Using this definition, we will attain solutions for $P_A(t)$ and $P_S(t)$ by first integrating over $T_S$'s probability density function from 0 to $t$ to yield the following cumulative density function:

$$G\left(t\right) = \int_0^t g\left(t_S\right) dt_S \tag{24}$$

This cumulative density function represents the fraction of active cells that have transitioned into the silent state until $t$ and must be multiplied by $\left(1 - P_B\right)$ to yield the overall weight of silent cells:

$$P_S\left(t\right) = \left(1 - P_B\right) G\left(t\right) \tag{25}$$

To get the weight of active cells, we simply subtract the cumulative density function from 1 to find the fraction of active cells that have not silenced until $t$ before multiplication by $\left(1 - P_B\right)$:

$$P_A\left(t\right) = \left(1 - P_B\right) \left(1 - G\left(t\right)\right) \tag{26}$$

With analytical expressions for each weight, we now consider each independent probability density function of fluorescence for each of the three cell states. First, the probability density function for the fluorescence of background silent cells does not have a time dependence and is log-normal with a median of the background fluorescence, $x_B$, and a shape parameter, or log-space standard deviation, $\sigma$:

$$f_B\left(x\right) = \frac{1}{x\sigma\sqrt{2\pi}} e^{\frac{-\ln\left(\frac{x}{x_B}\right)^2}{2\sigma^2}} \tag{27}$$

Next, the probability density function for the fluorescence of active cells is also log-normal, but the median of this distribution is given by $x_A\left(t\right)$ derived in the previous section. For simplicity, we assume that the shape parameter of this distribution, $\sigma$, is the same as the background distribution:

$$f_A\left(x,t\right) = \frac{1}{x\sigma\sqrt{2\pi}} e^{\frac{-\ln\left(\frac{x}{x_A(t)}\right)^2}{2\sigma^2}} \tag{28}$$

Finally, the probability density function for the fluorescence of silent cells is a convolution of $T_S$'s gamma distribution for silencing and a log-normal distribution with a median of $x_S\left(t, t_S\right)$. The convolution occurs over $t_S$ from time 0 to $t$. This step can be thought of as a weighted sum of log-normal distributions with different medians based on when silencing occurred. Because the weights only sum to the probability of silencing up to $t$, the integral must be normalized by division by G(t). This normalization ensures that $f_S\left(x,t\right) =$ integrates to one:

$$f_S\left(x,t\right) = \frac{1}{G\left(t\right)} \frac{1}{x\sigma\sqrt{2\pi}} \int_0^t g\left(t_S\right) e^{\frac{-\ln\left(\frac{x}{x_S(t,t_S)}\right)^2}{2\sigma^2}} dt_S \tag{29}$$

Again, we used the same shape parameter,, so that when the fluorescence decays to background levels, the distribution for fluorescence of background silent cells matches the distribution for silent cells:

$$\lim_{t\to\infty} f_S\left(x,t\right) = f_B\left(x\right) \tag{30}$$

We have now defined the weights and probability density functions for each of the three possible states in terms of some new parameters and the equations derived in the deterministic component, which concludes the probabilistic component of the derivation. The additional constant parameters of the model are:

$P_B$- the fraction of background silent cells
$k$- the shape parameter of $T_S$'s gamma distribution
$\tau_S$- the mean silencing time from of $T_S$'s gamma distribution in days
$\sigma$- the shape parameter for all log-normal fluorescence distributions

### Fitting method

Fitting our probabilistic model to the flow cytometry data allows us to estimate the relevant silencing parameters. Of particular interest are the interference parameter, γ and the mean silencing times for

mCitrine and mCherry, μS, C and μS, M as well as the difference between the two, or ΔμS. Our model fitting was conducted in Python 3 Jupyter notebooks.γ,

First, we defined functions F_o(t,params*) and F_s(t,t_l,params*) to calculate $x_A(t)$ and $x_S(t, t_S)$ from the deterministic portion of the derivation respectively. Within these functions, we set the mRNA half-life ($\beta_m$) to four hours as was previously determined experimentally (*Bintu et al., 2016*). Then we defined $f(x,t)$ to calculate the probability distribution derived in the probabilistic portion of the derivation. Although the derivation describes the distribution on a linear scale, $f(x,t)$ calculates the distribution on a $log_{10}$ scale as this scale is more commonly used to visualize and work with flow cytometry data.

With functions defined to calculate the model solution given a set of parameters, we next defined an error function to calculate error between the data and the model. To calculate this error, the fluorescence values are binned into 100 equally spaced bins for each timepoint in the data. The probability densities of each bin are calculated. Then, we evaluate the model with the given parameters at the midpoint of each bin. The error is calculated by subtracting the bin densities from the solution to the model's probability density function. The error arrays from each timepoint are appended to generate a single array of errors for the whole time course.

The mCitrine and mCherry reporters were each fit separately and had differences in the parameters used. Therefore, we used the general error function described above to define two separate error functions for mCitrine and mCherry. Since we never saw a spike in mCitrine expression, the interference parameter, γ, was intentionally set to one and the spike delay, $t_{S,C}$ in the deterministic derivation for mCherry, was set to zero. Fixing these parameters eliminates the reporter spike and allows us to use the same set of equations to describe both mCitrine and mCherry. Additionally, we set the shape parameter of $T_S$'s gamma distribution for mCitrine to one. This constrains the gamma distribution to an exponential distribution, which is more suitable for mCitrine because this reporter silences quickly and reliably upon induction (*Figure 3*, time-lapse movie results).

The mCherry fit was conducted after the mCitrine fit as some of the parameters are carried from mCitrine to mCherry. One of these parameters is the protein dilution $\beta_p$, which can be more reliably calculated in the mCitrine fit due to higher dynamic range and more uniform silencing. Since both the mCitrine and mCherry reporters are fused to H2B, they are stable and do not degrade significantly over the course of the experiment. Therefore, $\beta_p$ is governed by dilution due to cell division and is the same for both mCitrine and mCherry. In addition, $\tau_{S,C}$ was calculated in the mCitrine fit and then used as $t_{S,C}$ in the mCherry deterministic equations.

There is one additional consideration for the mCherry fits. Due to fast mCherry silencing by KRAB, it was difficult estimate both the interference factor, γ, and the mean silencing delay, $\mu_{S,M}$. Therefore, we used the γ's estimated from the slower HDAC4 silencing time courses for each reporter when estimating the parameters in the KRAB silencing data. We believe this is acceptable because transcriptional interference is a property of the reporter and not the CR. The exception is that for the K562 insulator lines we did not fit the model to the HDAC4 data because we do not see complete silencing of mCherry. For these HDAC4 lines we estimated γ by taking the ratio of the maximum mode of the fluorescence in the time course to the mode on day zero and used this estimated γ for the KRAB fits.

One final adjustment to the model was made to fit non-saturating dox concentrations (100 ng/mL and 200 ng/mL). When dox concentration was low, we did not observe silencing of mCitrine in all cells as was the case for 1000 ng/ml, and accordingly did not observe subsequent spreading of silencing to mCherry in those cells. To account for only a fraction of cells being silenced, an additional parameter was added to the model to represent cells that are always active, $P_{aa}$. These cells do not silence, and their fluorescence distribution is fixed at active-state levels. Consequently, only $1 - P_{aa}$ are involved in silencing and are subject to the components described by *equation (21)*.

To estimate the unconstrained parameters for each reporter, we fit the model to the data using a least-squares approach, scipy.optimize.least_squares(). This function solves for the parameters that minimize the sum of the squares of the error array that we defined. Initial guesses for each parameter were reasonably estimated by manual inspection of the data where possible. Bounds for each parameter were set generously as to avoid bounding of the solution where possible. The fits were performed independently for each replicate or single clone and took several hours for each cell line. However, the resulting fits were satisfactory by visual inspection, and the parameters were in accordance with those found in time course movie analysis.

The 90% confidence interval (CI) for the parameters extracted from fitting daily flow cytometry data to the model, was estimated using a t-distribution with t-score 2.92 and sample standard deviation from all replicates.

## Acknowledgements

We thank the members of the Bintu lab for discussions and helpful feedback, as well as Mike V Van for cloning some CR plasmids. We especially thank Stanley Qi's Lab for use of the Cytoflex cytometer, Mitsuo Oshimura's Lab for the MI-HAC and phiC31 integrase, and Brooks Taylor (Mary Teruel's Lab) for use of his cell-tracking software.

## Additional information

### Funding

| Funder | Grant reference number | Author |
|---|---|---|
| Burroughs Wellcome Fund | Career Awards at the Scientific Interface | Lacramioara Bintu |
| NIH Office of the Director | NIGMS R35MI128947 | Lacramioara Bintu |
| NIH Office of the Director | Training Grant in Biotechnology 5T32GM008412 | Sarah Lensch |
| Japan Society for the Promotion of Science | Postdoctoral Fellowship | Taihei Fujimori |
| NIH Office of the Director | T32 Molecular Pharmacology Training Grant 5T32GM113854 | Joydeb Sinha |
| National Science Foundation | GRFP | Michaela M Hinks |
| Stanford University | VPGE Graduate Fellowship | Michaela M Hinks |
| Stanford University | VPGE Edge Fellowship | Michaela M Hinks |
| NIH Office of the Director | T32 Training Grant 5T32GM007365-45 | Adi Mukund |

The funders had no role in study design, data collection and interpretation, or the decision to submit the work for publication.

### Author contributions

Sarah Lensch, Conceptualization, Formal analysis, Investigation, Methodology, Project administration, Validation, Visualization, Writing – original draft, Writing - review and editing; Michael H Herschl, Formal analysis, Investigation, Methodology, Validation, Writing – original draft; Connor H Ludwig, Formal analysis, Investigation, Writing – original draft; Joydeb Sinha, Formal analysis, Investigation; Michaela M Hinks, Investigation, Writing – original draft; Adi Mukund, Visualization, Writing – original draft, Writing - review and editing; Taihei Fujimori, Methodology; Lacramioara Bintu, Conceptualization, Funding acquisition, Investigation, Methodology, Resources, Supervision, Writing – original draft, Writing - review and editing

### Author ORCIDs

Sarah Lensch http://orcid.org/0000-0003-1450-3242
Adi Mukund http://orcid.org/0000-0003-2752-1458
Lacramioara Bintu http://orcid.org/0000-0001-5443-6633

### Decision letter and Author response

Decision letter https://doi.org/10.7554/eLife.75115.sa1
Author response https://doi.org/10.7554/eLife.75115.sa2

## Additional files

### Supplementary files
• Transparent reporting form

### Data availability
Plasmids generated in this paper have been deposited on Addgene on the Bintu Lab page https://www.addgene.org/browse/article/28223311/ (see details in the Key Resources Table of the Materials and Methods section). CUT&RUN data has been deposited on the Gene Expression Omnibus (GEO) database, Accession code: GSE189540. Scripts for CUT&RUN analysis, time-lapse microscopy analysis and model for time evolution of fluorescence distributions have been uploaded to Github and are referenced in the text: Lensch S., Ludwig C. Herschl M., Sinha J. 2022. Spreading_Lensch_2022. Github. https://github.com/bintulab/Spreading_Lensch_2022, e427d21 (copy archived at swh:1:rev:e427d21074b63ad6bd708ae497a1b182ecd286fe).

The following dataset was generated:

| Author(s) | Year | Dataset title | Dataset URL | Database and Identifier |
|---|---|---|---|---|
| Lensch S, Ludwig CH, Sinha J, Bintu L | 2021 | Changes in histone modifications measured by CUT&RUN after recruitment of chromatin regulators to dual-gene synthetic reporters | https://www.ncbi.nlm.nih.gov/geo/query/acc.cgi?acc=GSE189540 | NCBI Gene Expression Omnibus, GSE189540 |

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

# Appendix 1

**Appendix 1—table 1.** Statistics of reporters after silencing.

Percent of cells off with standard error of mean (SEM) for each gene in the -dox and +dox condition after the number of days indicated, with threshold drawn in *Figure 1* (dotted line). Welch's unequal variances T-test was run on the percent cells off from each replicate (n=3) and p-value is shown for comparison of -dox versus +dox for each gene.

| | % mCitrine Off | | | % mCherry Off | | |
|---|---|---|---|---|---|---|
| | -dox | +dox | p-value | -dox | +dox | p-value |
| CHO KRAB No Spacer (day 5) | 15±2 | 100±0.03 | 0.00055 | 15.3±2.5 | 100±0.03 | 0.00084 |
| CHO KRAB 5kb Spacer (day 5) | 23.2±6.9 | 99.9±0.1 | 0.00789 | 39.8±8.8 | 96.9±1.9 | 0.01916 |
| CHO HDAC4 No Spacer (day 5) | 9.6±2.1 | 99.6±0.3 | 0.00044 | 11±2.0 | 76.2±6.9 | 0.0071 |
| CHO HDAC4 5kb Spacer (day 5) | 11.3±2.9 | 99.9±0.03 | 0.00109 | 32.2±3.7 | 68.1±0.9 | 0.00764 |
| K562 KRAB No Spacer (day 5) | 2.6±0.2 | 95.9±2.0 | 0.00039 | 6.8±1.1 | 87.7±4.5 | 0.00198 |
| K562 KRAB 5kb Spacer (day 5) | 4.4±0.2 | 97±1.2 | 0.00012 | 12.3±1.4 | 86.1±4.3 | 0.00148 |
| K562 HDAC4 No Spacer (day 20) | 10±6.6 | 98.3±0.3 | 0.0056 | 30.5±7.2 | 67.1±4.8 | 0.01784 |
| K562 HDAC4 5kb Spacer (day 20) | 9.4±5.4 | 99.3±0.09 | 0.00352 | 24.2±5.1 | 83.8±1.1 | 0.00555 |
| HEK HDAC4 5kb Spacer (day 7) | 1.3±0.2 | 86.7±3.5 | 0.00167 | 2.5±0.5 | 73.5±3.8 | 0.00256 |

**Appendix 1—table 2.** Delay times between mCitrine and mCherry silencing from model fit to flow cytometry data.

The 90% confidence interval (CI) was estimated using the t-distribution.

| | | Reporter | $\Delta\tau s$ (hour) | 90% CI |
|---|---|---|---|---|
| CHO-K1 | KRAB | NS | -1.34 | ±0.59 |
| | | 1.2 kb | 0.75 | ±1.26 |
| | | 5kb | 2.78 | ±2.82 |
| | | SH | -4.36 | ±8.96 |
| | | SC | -4.29 | ±6.75 |
| | | DH | -0.78 | ±10.36 |
| | | DC | -2.92 | ±7.14 |
| | HDAC4 | NS | 58.4 | ±28.2 |
| | | 1.2 kb | 54.9 | ±34.8 |
| | | 5kb | 44.5 | ±1.1 |
| | | SH | 64 | ±32.5 |
| | | SC | 67 | ±28.9 |
| | | DH | 69.1 | ±22.6 |
| | | DC | 59.9 | ±17.9 |

*Appendix 1—table 2 Continued on next page*

*Appendix 1—table 2 Continued*

|  |  | Reporter | Δτs (hour) | 90% CI |
|---|---|---|---|---|
| K562 | KRAB | NS | 2.1 | ±2.91 |
|  |  | 5kb | 13.3 | ±9.1 |
|  |  | SH | 1.29 | ±0.37 |
|  |  | SC | 5.86 | ±2.58 |
|  |  | DH | -1.06 | ±2.25 |
|  |  | DC | 6.76 | ±2.52 |
|  | HDAC4 | NS | 261 | ±177 |
|  |  | 5kb | 130 | ±32 |

# Appendix 2

## Transcriptional Interference

We noticed an increase in mCherry expression after 5 days of HDAC4 recruitment in K562 with the NS reporter (*Figure 1H*). We wondered if this spike in mCherry fluorescence was due to ceasing of transcriptional interference on the pRSV-mCherry from the upstream pEF promoter. Transcriptional interference can occur through promoter occlusion as has been previously measured (*Shearwin et al., 2005*; *Adhya and Gottesman, 1982*; *Eszterhas et al., 2002*).We hypothesized that if transcription termination failed at the mCitrine polyA, Pol II would run on into the RSV promoter and hinder transcription initiation of the mCherry gene (*Figure 1—figure supplement 4A*). Upon transcriptional silencing of the upstream pEF promoter, regular transcription initiation at pRSV can resume, leading to the observed temporary increase in mCherry expression prior to chromatin mediated silencing of pRSV-mCherry (*Figure 1—figure supplement 4B*). To test if any transcripts span from the mCitrine gene body across pRSV into the mCherry gene body, we performed PCR on cDNA from K562 cells after different durations of HDAC4 recruitment (primers in *Appendix 2—table 1*). This revealed that indeed, in cells where pEF has not yet been silenced by HDAC4, there is transcriptional run-on from pEF over pRSV, and that run-on does not occur in cells where mCherry is increased (*Figure 1—figure supplement 4B*). This suggests that the observed increase in mCherry production is due to pEF silencing and transcriptional interference no longer occurring. We do not see this effect with KRAB because silencing spreads more quickly than for HDAC4 and does not leave sufficient time for transcription to initiate at pRSV before both genes are silenced.

To experimentally measure the dynamics of transcriptional interference, we performed RT-qPCR (primers in *Appendix 2—table 1*) against different elements of the reporter after 0, 1 and 5 days of HDAC4 recruitment in CHO-K1 cells with the 5 kb reporter (*Figure 1—figure supplement 4C&D*). Our data confirmed transcription of the Our data confirmed transcription of the 5 kb lambda region and revealed that mCitrine and lambda mRNA levels both decreased after one day of HDAC4 recruitment. This further suggests that transcription runs on from mCitrine through the 5 kb lambda into mCherry, preventing transcription initiation at pRSV. Interestingly, mCherry mRNA abundance did not change between 0 and 1 day of HDAC4 recruitment, even though the mCherry protein level significantly increased after 1 day of HDAC4 recruitment. Since it is not possible to distinguish the initiation point of transcripts by RT-qPCR, the measured mCherry mRNA abundance comes from a combination of transcripts initiated at pRSV and transcripts initiated at pEF that run through mCherry. It should only be possible to translate mCherry from transcripts initiated at pRSV since the mCherry in the run-on transcript would not be in the correct reading frame. Therefore, the observed increase in mCherry protein abundance may result from an increase in the fraction of translatable mCherry mRNA as pEF is silenced. Further experiments would be needed to confirm this.

We noticed that in the absence of dox pRSV-mCherry expression for the SH reporter is much higher than for the 5 kb lambda reporter in CHO-K1 (*Figure 3—figure supplement 3*). We also do not observe an increase in mCherry expression upon HDAC4 recruitment in CHO-K1 cells with insulators (*Figure 1—figure supplement 4E*), unlike in the 5 kb lambda reporter (*Figure 1—figure supplement 4D*). We performed PCR on cDNA to test if any transcripts span from the mCitrine gene body across the cHS4 insulator and pRSV into the mCherry gene body at different durations of HDAC4 recruitment in CHO-K1 (primers in *Appendix 1—table 2*). The run-on transcript was not present at day 0 or after 1 or 5 days of HDAC4 recruitment in the SH insulator by PCR (*Figure 1—figure supplement 4F*). This suggests that the single cHS4 insulator, and possibly the cHS4 core sequence, can terminate transcription. This effect is in agreement with previous studies which have shown that insulator sequences can enhance gene expression and reduce promoter interference (*Tian and Andreadis, 2009*).

**Appendix 2—table 1.** Primers used for investigating transcriptional interference.

| Assay | Locus | Name | Sequence (5'->3') |
|---|---|---|---|
| PCR | Beta-actin (human genome) | ACTB-F1 | CATGTACGTTGCTATCCAGGC |
| PCR | | ACTB-R1 | CTCCTTAATGTCACGCACGAT |
| PCR | mCitrine | citrine3prime_fwd | GTAAGTGTACCCAATTCGCCCTATAGTGAG |

*Appendix 2—table 1 Continued on next page*

*Appendix 2—table 1 Continued*

| Assay | Locus | Name | Sequence (5'->3') |
|-------|-------|------|-------------------|
| PCR | mCherry | mCherry5prime_reverse | TCCTCGCCCTTGCTCACCAT |
| qPCR | mCitrine | F_cit_Set2 | CGGCGACGTAAACGGCCACAAGTTCAG |
| qPCR | | R_cit_Set2 | CTTGCCGGTGGTGCAGATGAA |
| qPCR | 5kb lambda | 5kb_lambda_Set1_F | CCACCTGTTACTGGTCGATTTA |
| qPCR | | 5kb_lambda_Set1_R | GATATTCCCACCTCCGGTTAAG |
| qPCR | mCherry | mCherry_Set1_F | AGGACGGCGAGTTCATCTA |
| qPCR | | mCherry_Set1_R | CCCATGGTCTTCTTCTGCATTA |
| qPCR | Beta-actin (CHO-K1 genome) | bActin_F | ACTGGGACGATATGGAGAAG |
| qPCR | | bActin_R | GGTCATCTTTTCACGGTTGG |

## Appendix 3

### Background Silencing

The extent to which background silencing is mitigated is influenced by the insulator configuration, cell type, and the CR that is over-expressed. By monitoring mCitrine and mCherry expression over time without CR recruitment, we are able to quantify the fraction of cells that spontaneously silence either mCitrine or mCherry (*Figure 4—figure supplement 5*). We calculated the rate of background silencing over time for each reporter by performing a least squares regression to the fraction of background silenced cells for each clone or replicate (*Figure 4—figure supplement 5*). For both KRAB and HDAC4 in CHO-K1 cells, all the insulator configurations reduced background silencing of the mCitrine and mCherry genes in comparison to the NS and 5 kb reporters. By contrast, the insulators increase the rate of background silencing in some configurations in K562 cells, and this effect is dependent on the CR present. With KRAB, all the insulator configurations lead to slightly higher background silencing than the NS and 5 kb reporters, and the level of silencing is similar for mCitrine and mCherry. With HDAC4, the levels of background silencing differ between mCitrine and mCherry, and the SH and DH insulators have strikingly high levels of mCherry background silencing. In only switching either the CR or cell type in a controlled system, we show that insulators have vastly different effects on neighboring genes.

