## [Editor Report]

This study describes a novel approach to investigate how the transcriptional repressors KRAB and HDAC4 repress gene expression, how repression spreads and the role of insulator elements in blocking the spread of repression and in reactivation of repressed genes. Despite some inherent limitations resulting from the use of artificial reporters compared to previous genomic studies, it addresses the question of repression in a rather novel manner adding a dimension of time and single cell resolution. The results allow modeling of the coordinated repression or activation of closely linked genes and should be of wide interest to researchers interested in chromatin, gene expression and synthetic biology. The authors have made a thorough revision of the paper and have addressed all of the major issues raised by the referees. They have better discussed their results in the context of previous studies and they have added new data with EZH2 inhibitors that support a role for PRC2 in HDAC4-mediated repression.

---

## [Decision Letter]

**Decision letter after peer review:**

Thank you for submitting your article "Dynamic spreading of chromatin-mediated gene silencing and reactivation between neighboring genes in single cells" for consideration by *eLife*. Your article has been reviewed by 3 peer reviewers, including Irwin Davidson as Reviewing Editor and Reviewer #1, and the evaluation has been overseen by Kevin Struhl as the Senior Editor. The following individual involved in review of your submission has agreed to reveal their identity: Didier Trono (Reviewer #2).

While the referees found the paper reported interesting observations, a major issue raised by all three referees regards the generalities drawn from a very limited set of observations (2 cell lines, 2 genomic loci, 2 chromatin modifiers, 2 promoters, a single distance of 5 kb as spacer, one insulator). In addition, previously published data was not discussed in the context of this current study. For example, Amabile et al. 2015 and Groner et al. 2010 (both referenced but not much commented on) were much broader and already made many of the points stated in the present work.

A major revision of the manuscript would considerably gain from placing its results in the context of the sum of previous studies, discussing specifically what the current study adds to the field.

Additional data.

The authors show that HDAC4 repression is associated with gain of H3K27me3. To investigate the role of PRC2 in this process and the requirement for the K27me3 mark in HDAC4 repression, the authors should test silencing of the locus in the presence of EZH2 inhibitor. In addition, there may be published data / databases where HDAC inhibition triggers silencing likely via H3K27me3 deposition may be seen in broader contexts.

Particular attention should be paid to the following issues.

1. The authors should mention in the text of the paper what is known about the differences in half-life/stability of the mCitrine and mCherry reporter proteins. This is important to assess the kinetics of repression. While imaging and flow cytometry allow assessment of their expression at the single cell level.

2. The authors do not show kinetics and overall expression of the Dox-induced KRAB and HDAC proteins and how these levels compare with the endogenous cellular levels. If as may be expected, the levels are much higher than endogenous, how does this affect the interpretation of the results in terms of kinetics and spreading of repression?

3. The authors should better explain why they chose KRAB and HDAC4 for their experiments. They should also comment on the histone deacetylase activity recruited by the KRAB-binding KAP1. KRAB is not a protein, but a domain, which often but not always recruits KAP1, and some KRAB-containing proteins also harbour other domains and some are not repressors. This should be explained as well to help the non-specialists situate the relevance of the work in a broader context.

4. Line 74: "a set of mammalian cell lines" could be advantageously replaced by "two mammalian cell lines".

5. A statistical test should be performed to qualify the yellow and red subpopulations.

6. A spacing of 0 or 5kb hardly qualifies as "various spacer lengths" as stated on line 79.

7. Lines 112-114: spreading from the HDAC4 docking site takes 20 days, is observed in only a fraction of cells, and the interpretation of the results is complicated by promoter interference intrinsic to the vector design (spreading is more efficient in the 5kb- than in the 0kb-spacing configuration). The data thus do not warrant the statement "these results show for the first time that silencing mediated by a histone deacetylase can also extend to nearby genes".

8. The models presented in 2F and 2G are not clear: The way the picture is drawn, it seems that loss of H3Kac is a temporal prerequisite of H3K9me3 and H3K27me3 deposition in both models. This might be true for HDAC4 (although the authors create some controversy on this particular point later in the manuscript), but not for KRAB.

9. Line 299: reference to the figure should be corrected (Figure 3GandH Figure 4GandH, Figure 4—figure supplement 2).

10. Lines 331-333: "These results suggest that the insulators can help the downstream gene remain active in conditions where its silencing is already slow, such as after indirect silencing by HDAC4 recruitment in K562." Again, this hypothesis relies on so few observations (one locus, one insulator, one cell line, one CR, one spacer) that it should be formulated with much caution.

11. Lines 510-515: one cannot say that the referenced previous studies yielded "disparate" results, notably on the extent of KRAB-induced heterochromatin spreading. Compared to the present work, they covered far greater variety of settings relative to the strengths of the targeted promoters, their distance from the KRAB docking site and their immediate or broader chromatin environment, so that a broad range of phenotypes was expected.

12. The authors should integrate in their thinking results reported ten years ago (BMC Genomics 12:378, 2011). This work reported an extensive analysis of chromatin features conducive or not of KRAB-induced long-range silencing, making interesting conclusions on the "repression-primed" state of the most sensitive promoters and their paradoxical location in a surrounding of highly expressed genes, and on the failure of CTCF recruitment to exert any barrier effect on H3K9me3 spreading.

13. The authors should also comment on KRAB zinc finger protein genes, many of which recruit KAP1 at their 3' end and are grouped in tightly packed clusters, yet can escape self- or mutually inflicted silencing. And on the interesting observation by K. Helin's group that the H3K9me1/2 demethylase PHF8 is recruited to the promoter of many of these genes, perhaps explaining this phenomenon (Mol Cell 2010 38: 165).

14. The potential roles of histone acetylation functioning as insulator sequences require more rigorous testing and validation. Perhaps the author should mine some published genomic data to find potential correlations between various histone acetylation marks and heterochromatin vs euchromatin compartments in development or disease conditions.

*Reviewer #1 (Recommendations for the authors):*

The authors should mention in the text of the paper what is known about the differences in half-life/stability of the mCitrine and mCherry reporter proteins. This is important to assess the kinetics of repression. While imaging and flow cytometry allow assessment of their expression at the single cell level, additional RT-qPCR experiments for mRNA levels for each gene would strengthen the analyses.

The authors do not show kinetics and overall expression of the Dox-induced KRAB and HDAC proteins and how these levels compare with the endogenous cellular levels. If as may be expected, the levels are much higher than endogenous, how does this affect the interpretation of the results in terms of kinetics and spreading of repression?

The authors have investigated the role of the cSH4 insulator element, can they better justify why this one was chosen. As the properties of insulator elements are not as well described as promoters or enhancers, experiments with other insulators would be required before general statements can be made on their ability to block repression or contribute to reactivation of gene expression.

It is now well accepted that regulation of gene expression is mainly driven by enhancer elements. The reporter plasmid used in this study comprises only proximal promoter elements. Do the authors know whether their promoter activity is driven/influenced by distal enhancers located around the integration sites? More specifically it would be informative to generate a reporter with a strong enhancer element and assess how this impacts repression. Insertion of an enhancer between the two genes may reveal whether it can block the spreading of repression to the mCHERRY gene.

*Reviewer #2 (Recommendations for the authors):*

Introduction:

1. The authors should better explain why they chose KRAB and HDAC4 for their experiments. They should also comment on the histone deacetylase activity recruited by the KRAB-binding KAP1. KRAB is not a protein but a domain, which often but not always recruits KAP1, and some KRAB-containing proteins also harbour other domains and some are not repressors. This should be explained as well to help the non-specialists situate the relevance of the work in a broader context.

Regarding Figure 1

1. Line 74: "a set of mammalian cell lines" could be advantageously replaced by "two mammalian cell lines".

2. A statistical test should be performed to qualify the yellow and red subpopulations.

3. A spacing of O or 5kb hardly qualifies as "various spacer lengths" as stated on line 79.

4. Lines 111-112: "Moreover, in K562 cells, HDAC4-mediated silencing of mCherry was much slower, taking up to 20 days, compared to 5 days in CHO-K1 cells (Figure 1HandJ). Any lead to explain the difference? Shouldn't other chromosomal loci be tested before attributing the difference to the cellular background?

5. Lines 112-114: spreading from the HDAC4 docking site takes 20 days, is observed in only a fraction of cells, and the interpretation of the results is complicated by promoter interference intrinsic to the vector design (spreading is more efficient in the 5kb- than in the 0kb-spacing configuration). The data thus do not warrant the statement "these results show for the first time that silencing mediated by a histone deacetylase can also extend to nearby genes". Rare are the pairs of genes, the promoters of which are at such short distances.

Regarding Figure 2

1. The models presented in 2F and 2G are not clear: The way the picture is drawn, it seems that loss of H3Kac is a temporal prerequisite of H3K9me3 and H3K27me3 deposition in both models. This might be true for HDAC4 (although the authors create some controversy on this particular point later in the manuscript), but not for KRAB.

2. Lines 181-183: "… that can lead to spreading of these chromatin modifications (Figure 2F). This observation suggests that silencing of the RSV promoter is indirectly due to PRC2 action, but it can only take place once HDAC4 removes acetylation at the locus." Artificial recruitment of PRC2 at this locus in the absence of HDAC4 could be used to validate this hypothesis.

Regarding Figure 3

The authors used time-lapse movies to analyse the spreading of epigenetic marks with high temporal resolution. They conclude that KRAB-mediated spreading of transcriptional silencing is distance-dependent in CHO-K1 cells while HDAC4-mediated spreading is distance-independent at these length scales.

1. A limitation of time-lapse movies is the low output. However, the authors developed a mathematical model to optimize the information recovered from "high output" data (flow cytometry combined with gene expression). The output of this method is nicely consistent with time-lapse movies opening perspectives to use this system on a larger scale. A useful contribution to the field!

2. There should be published data / databases allowing to verify that in broader contexts HDAC inhibition triggers silencing likely via H3K27me3 deposition (Line 176), rather than it being a peculiarity of the dual system used here.

3. Further down (lines 178-180) it is stated that "This repressive modification, which is associated with polycomb repressive complex 2 (PRC2), has not been reported in association with HDAC4, and was not observed after HDAC4 recruitment at a pEF-mCitrine reporter flanked by insulators in the same locus of CHO-K1 cells". To consolidate their hypothesis, the authors could test silencing of the locus in the presence of EZH2 inhibitor to see in which extent it depends on PRC2, and measure PRC2 enrichment by CUTandTAG or CUTandRUN.

4. Lines 223-226: "The observation that the delays of pRSV-mCherry silencing do not increase with distance (Figure 3EandJ) suggests that its silencing initiates at the RSV promoter, most likely by the action of PRC2 (Figure 2D, EandG).". If the silencing starts from the RSV promotor, why is it different without the 5Kb spacer? Can the dynamics of H3K27me3 deposition at the RSV promotor be assessed on both constructs? What would be the mechanism leading to the recruitment of PRC2 at the RSV instead of the pEF since HDAC4 is recruited at pEF? What is the effect of inverting the two promoters?

Regarding Figure 4

In this figure, the authors document the lack of effect of cHS4 insulators on KRAB/HDAC4-induced silencing in CHO cells, but observe some impact on the latter in K562 cells.

1. Line 299: reference to the figure should be corrected (Figure 3GandH Figure 4GandH, Figure 4—figure supplement 2).

2. Lines 331-333: "These results suggest that the insulators can help the downstream gene remain active in conditions where its silencing is already slow, such as after indirect silencing by HDAC4 recruitment in K562." Again, this hypothesis relies on so few observations (one locus, one insulator, one cell line, one CR, one spacer) that it should be formulated with much caution.

Regarding Figure 5

Here the authors investigate the re-activation of the genes in their system. Again, they formulate overly general conclusions based on a very limited set of parameters. Lines 395-398: "These results show that the extent of epigenetic memory depends strongly not only on the chromatin regulator recruited, but also on the configurations of promoters and insulators at the target locus, with stronger promoters closely surrounded by core insulators diminishing memory".

Regarding Figure 6 (Model)

The model is fine as long as it is clearly restricted to the parameters tested here, and not proposed as a paradigm.

Discussion

1. Lines 510-515: one cannot say that the referenced previous studies yielded "disparate" results, notably on the extent of KRAB-induced heterochromatin spreading. Compared to the present work, they covered far greater variety of settings relative to the strengths of the targeted promoters, their distance from the KRAB docking site and their immediate or broader chromatin environment, so that a broad range of phenotypes was expected, without this heterogeneity being shocking (which is what the term "disparate" suggests).

2. Lines 560-561: "By monitoring gene reactivation dynamics after release of the CRs, we conclude that activation can also spread to nearby genes in a distance-dependent manner." Only one distance (5Kb) was tested, which pales with the thousands of distances evaluated in some previously published studies.

3. Lines 571-572: "They suggest that genes in close proximity (<5-10kb) can respond to signals in a coordinated manner, during both silencing and activation". Right, although confirmatory of a large series of observations previously made on co-regulated gene clusters.

4. Lines 579- 582: "Finally, this experimental and theoretical framework can serve as a starting point for measuring and modeling the effects of targeting various epigenetic editors at endogenous loci in order to guide the time necessary for efficient on-target effects without unwanted off-target spreading". The authors fail to demonstrate convincingly that their system is really suited for the kind of high throughput analyses that would be necessary to consolidate, give a general value or even validate some of their conclusions.

5. The authors should integrate in their thinking results reported ten years ago (BMC Genomics 12:378, 2011). This work proceeded to an extensive analysis of chromatin features conducive or not of KRAB-induced long-range silencing, making interesting conclusions on the "repression-primed" state of the most sensitive promoters and their paradoxical location in a surrounding of highly expressed genes, and on the failure of CTCF recruitment to exert any barrier effect on H3K9me3 spreading.

6. The authors should also comment on KRAB zinc finger protein genes, many of which recruit KAP1 at their 3' end and are grouped in tightly packed clusters, yet can escape self- or mutually inflicted silencing. And on the interesting observation by K. Helin's group that the H3K9me1/2 demethylase PHF8 is recruited to the promoter of many of these genes, perhaps explaining this phenomenon (Mol Cell 2010 38: 165).

*Reviewer #3 (Recommendations for the authors):*

The authors should include a more critical discussion of the differences of this model with prior studies where KRAB domains were recruited to natural genomic sequences.

The potential roles of histone acetylation functioning as insulator sequences require more rigorous testing and validation. Perhaps the author should mine some published genomic data to find potential correlations between various histone acetylation marks and heterochromatin vs euchromatin compartments in development or disease conditions.

---

## [Author Response]

While the referees found the paper reported interesting observations, a major issue raised by all three referees regards the generalities drawn from a very limited set of observations (2 cell lines, 2 genomic loci, 2 chromatin modifiers, 2 promoters, a single distance of 5 kb as spacer, one insulator). In addition, previously published data was not discussed in the context of this current study. For example, Amabile et al. 2015 and Groner et al. 2010 (both referenced but not much commented on) were much broader and already made many of the points stated in the present work.A major revision of the manuscript would considerably gain from placing its results in the context of the sum of previous studies, discussing specifically what the current study adds to the field.

We agree with these comments, and have made major edits throughout the entire text of the manuscript to decrease the generalities of our claims and restrict them to our system, describe in more detail what has been shown in previous studies, motivate our experimental design choices better, emphasize the novel results from our work, and place our results in the context of previous work. Please see point-by-point responses to each reviewer.

Additional data.The authors show that HDAC4 repression is associated with gain of H3K27me3. To investigate the role of PRC2 in this process and the requirement for the K27me3 mark in HDAC4 repression, the authors should test silencing of the locus in the presence of EZH2 inhibitor. In addition, there may be published data / databases where HDAC inhibition triggers silencing likely via H3K27me3 deposition may be seen in broader contexts.

We added the new required data: To further test this hypothesis of PRC2-mediated silencing of the downstream pRSV-mCherry gene, we measured HDAC4-mediated silencing in the presence of EZH2 inhibitor Tazemetostat (Taz). We found that Taz blocked pRSV-mCherry silencing, while not affecting the pEF-mCitrine silencing. These results support the idea that pRSV-mCherry silencing we observe after HDAC4-mediated silencing of the pEF-mCitrine results from PRC2 mediated silencing of the pRSV-mCherry. We have added these data to Figure 2F-H as well as new supplemental Figure 2 —figure supplement 3. Additions to the main text include:

– Line 180, added text: “To test whether the silencing of mCherry was mediated by PRC2, we recruited HDAC4 in the presence of Tazemetostat (Taz), a small molecule inhibitor of EZH2, which is the H3K27 methyltransferase in PRC2 (Figure 2F). First, we tested Taz in the NS reporter line with recruitment of EED, the H3K27 reader in PRC2, and showed that inhibition reduced silencing of mCitrine and mCherry (Figure 2 —figure supplement 3A-B). We found that in the presence of dox and Taz while the mCitrine gene was still silenced by HDAC4, the mCherry gene was not silenced and even slightly increased in expression in the 5kb reporter (Figure 2G-H) and the NS reporter (Figure 2 —figure supplement 3C).

– Line 419, add to sentence: “loss of pRSV-mCherry silencing upon PRC2 inhibition

(Figure 2G-H), and the fact that the silencing delay is distance-independent.”

– Line 536, add to sentence: “which was confirmed by the loss of mCherry silencing when PRC2 was inhibited (Figure 2F-H).“

– Line 21, edit sentence in abstract to reflect the role of PRC2: “Silencing by histone deacetylase HDAC4 of the upstream gene can also facilitate background silencing of the downstream gene by PRC2, but with a days-long delay that does not change with distance.”

– Edits to legend of Figure 2:

– (F) Addition of Tazemetostat (Taz) inhibits the EZH2 methyltransferase from PRC2. (G) Fluorescence distributions of mCitrine (left) and mCherry (right) measured by flow cytometry either without CR recruitment (-dox, grey) or after 7 days of HDAC4 recruitment (+dox+DMSO, purple) and with Taz(+dox+Taz, pink) at the 5kb reporter in CHO-K1. (H) Percentages of cells OFF normalized by the -dox+DMSO condition, based on threshold shown in panel G (dotted line).

– (I) In the absence of CR recruitment, both genes have H3Ac across the reporter (top). Upon recruitment of KRAB (bottom left), we see a loss of H3Kac and gain of H3K9me3 across the dual-gene reporter through both DNA looping from rTetR-KRAB as well as positive feedback loops for spread of methylation, resulting in a distance-dependent delay of transcriptional silencing between two genes. Upon recruitment of HDAC4 (bottom right), we see a loss of H3Kac as well as a gain of H3K27me3 across the reporter through positive feedback loops for spread of methylation by PRC2, resulting in distance-independent delay of transcriptional silencing between two genes.

New figure legend:

Figure 2 —figure supplement 3. EZH2 inhibitor Tazemetostat affects silencing after recruitment of EED and HDAC4.

(A) Addition of Tazemetostat (Taz) inhibits the EZH2 methyltransferase from PRC2. (B-C) Fluorescence distributions of mCitrine (top left) and mCherry (bottom left) measured by flow cytometry either without CR recruitment (-dox, grey) or after 7 days of (B) EED or (C) HDAC4 recruitment (+dox+DMSO, purple) and with Taz(+dox+Taz, pink) at the NS reporter in CHO-K1. Percentages of cells OFF normalized by the -dox+DMSO condition (right), based on threshold (dotted line).

With respect to the connection between HDAC inhibition, loss of acetylation, and H3K27me3 deposition in broader contexts, we have modified the manuscript to describe published results which support our hypothesis that acetylation removal can lead to PRC2associate chromatin modifications:

– Added two sentences at line 183: “ In accordance with our hypothesis, previous studies have shown that by co-ChIP of primary cells H3K27me3 and H3Ac are negatively correlated (Weiner et al. 2016), that lack of transcription and histone deacetylation both drive an increase in H3K27me3 deposition steadily over time (Hosogane et al. 2016), and that recruitment of HDAC1 leads to repressed chromatin associated with a strong increase in H3K27me3 (Song et al. 2016). Additionally, knockdown of p300, a histone acetyltransferase, resulted in a global decrease of H3K27Ac in tandem with an increase in H3K27me3 (Martire et al. 2020).”

– In Discussion section, line 536 added: “In accordance with our observations, previous studies have shown that H3K27me3 and H3Ac are negatively correlated on individual nucleosomes (Weiner et al. 2016), and specifically that deacetylation by either lack of transcription or recruitment of HDAC1 leads to increases in H3K27me3 (Hosogane et al. 2016; Song et al. 2016). Additionally, studies where cells are treated with HDAC inhibitor show in an increase in histone acetylation which is correlated with reduced recruitment of EZH2, EED and Suz12 proteins from the PRC2 as well as a decrease in H3K27me3 (Reynolds et al. 2012; Wang et al. 2012), showing the bidirectionality of the inverse correlation between H3K27me and H3K27Ac (Pasini et al. 2010). This supports our observation that low levels of acetylation allow PRC2 to work more effectively.”

Particular attention should be paid to the following issues.1. The authors should mention in the text of the paper what is known about the differences in half-life/stability of the mCitrine and mCherry reporter proteins. This is important to assess the kinetics of repression. While imaging and flow cytometry allow assessment of their expression at the single cell level.

We agree with the reviewer that the half-life/stability of the reporter proteins are important parameters for the kinetics of repression. Our reporter genes are fusions between H2B and a fluorescent protein: H2B-mCitrine and H2B-mCherry. Fusions between H2B and fluorescent proteins (H2B-FPs) are used for studying chromatin dynamics due to their high signal to noise ratio from nuclear localization and incorporation into nucleosomes without inhibiting cell cycle progression (Kanda et al. 1998; Hadjantonakis and Papaioannou 2004; Stewart et al. 2009; Fraser et al. 2005). H2B-FPs are very stable, with very long protein halflives on the order of weeks; therefore degradation is mostly driven by cell division (Morcos et al. 2020). At the mRNA level, we have previously shown that the half-life of H2B-mCitrine mRNA transcript is about 3.9 hours as measured by inhibiting transcription with actinomycin D and quantifying mRNA overtime (Bintu et al. 2016). Although we have not measured the lifetime of the H2B-mCherry mRNA our video data and our model from daily flow cytometry data shows that the delay between mCitrine and mCherry silencing (δ T) is close to zero in the NS reporters (Figure 3Band3I), suggesting that mCitrine and mCherry lifetimes are very close together. Small differences in mRNA stability could change silencing times of each gene but ultimately would shift the δ T delay equally for all reporter constructs, and therefore not change our conclusion that the delay increases with distance, which is also corroborated by our video results which measure silencing at the protein level.

We have added this information into the main text as follows:

– Line 73-76, first sentence edited:

– Original: “ To study the spreading of transcriptional silencing and chromatin modifications, we engineered a set of mammalian cell lines containing two neighboring fluorescent reporter genes and a CR, with each gene driven by constitutive promoters (Figure 1A-B, Figure 1—figure supplement 1).”

-New: “To study the spreading of transcriptional silencing and chromatin modifications, we site-specifically integrated reporters consisting of two neighboring fluorescent reporter genes (mCitrine and mCherry) with each gene driven by a constitutive promoters (Figure 1A-B, Figure 1—figure supplement 1). Both fluorescent proteins are fused to H2B; this approach increases the signal to noise ratio due to nuclear localization and is commonly used for studying chromatin dynamics (Kanda et al. 1998; Hadjantonakis and Papaioannou 2004; Stewart et al. 2009; Fraser et al. 2005). In each reporter cell line, we also stably integrated a recruitable rTetR fused to either KRAB or HDAC4.”

Line 195, added sentence: “Since H2B fusions to fluorescent proteins have long halflives on the order of weeks, their loss is mostly driven by cell division (Morcos et al. 2020), which allows the silencing of the gene to be detected based on the change in fluorescent protein accumulation. Stitching together single-cell traces across cell divisions results in cumulative traces of fluorescent protein levels over time for specific lineages (Figure 3A, Figure 3 —figure supplement 1). The slope of these traces is then proportional to promoter activity, and its decrease can be used to determine silencing of a gene(Materials and methods).”

Line 218, add to sentence:

“… especially if we consider the additional time necessary for mRNA degradation of about ~4 hours included in the flow fits (Materials and methods), as determined previously by inhibiting transcription with actinomycin D and quantifying the decay of H2B-mCitrine mRNA over time(Bintu et al. 2016). Since we observed a delay time between mCherry and mCitrine silencing of less than 4 hrs after KRAB recruitment (Figure 3B), we assumed that the H2B-mCherry mRNA degrades at a similar rate as H2B-mCitrine.”

2. The authors do not show kinetics and overall expression of the Dox-induced KRAB and HDAC proteins and how these levels compare with the endogenous cellular levels. If as may be expected, the levels are much higher than endogenous, how does this affect the interpretation of the results in terms of kinetics and spreading of repression?

Indeed, we would expect the expression levels of our chromatin regulators to be higher than endogenous levels, and we also expect the amount of rTetR-CR recruited at the reporter to affect the dynamics of silencing.. In fact, with our dox inducible system we are able to tune the number of rTetR-KRAB or rTetR-HDAC4 proteins that can be recruited to the reporter locus by altering dox concentration. As shown in Figure 4 – supplement 4, at lower dox concentrations, in the insulator reporters silencing of mCherry still occurs in cells where mCitrine is silenced. We find that although the fraction of the population that silences mCitrine is lower, in the cells with mCitrine silent, silencing spreads to the downstream mcherry within 5 days for both HDAC4 and KRAB. We have also collected additional flow cytometry data with KRAB recruitment to NS and 5kb reporters at lower dox concentrations. As seen with the insulator reporters, we find that lower dox concentration reduces the fraction of cells that silence but that in cells where mCitrine is silenced, mCherry silencing follows in both NS and 5kb reporters. Although the lower dox concentration slows down the rate of spreading, it does not alter coordinated dynamics of the two genes. In the NS reporter, both genes silence simultaneously at 4 ng/mL (Figure 1 —figure supplement 3A). In the 5kb reporter after 5 days at lower dox concentrations, mCitrine still silences first with slower spreading to mCherry, this progression can be seen in the 4 ng/mL and 10 ng/mL concentrations respectively (Figure 1 —figure supplement 3B).

We have now added this additional data as Figure 1 —figure supplement 3, and it is referenced in the main text as follows:

– Line 107, added sentence: “Tuning the probability of of KRAB recruitment at the reporter by varying dox concentration, we found that spreading of silencing still occurs in both the NS and the 5kb reporters, and the percentage of cells with both genes silenced increases with dox concentration (Figure 1 —figure supplement 3).”

New figure legend:

Figure 1 —figure supplement 3. Silencing by KRAB at lower dox concentrations.

(A-B) 2D scatter plots of mCitrine and mCherry fluorescence measured by flow cytometry in CHO-K1 cells with KRAB recruitment for 5 days at 4 ng/mL, 10 ng/mL and 1000 ng/mL in the (A) NS reporter or (B) 5kb reporter. Percent of cells in each quadrant is shown with standard error of mean (n=2).

3. The authors should better explain why they chose KRAB and HDAC4 for their experiments. They should also comment on the histone deacetylase activity recruited by the KRAB-binding KAP1. KRAB is not a protein, but a domain, which often but not always recruits KAP1, and some KRAB-containing proteins also harbour other domains and some are not repressors. This should be explained as well to help the non-specialists situate the relevance of the work in a broader context.

Indeed, we have modified the paragraphs introducing KRAB and HDAC4 to better reflect why we chose them, as well as the known biology of KRAB as follows (additions highlighted):

New KRAB paragraph (lines 46-52): “The Kruppel associated box (KRAB) repressive domain from zinc finger 10 is commonly used in synthetic biology applications (Nakamura et al. 2021), is one of the strongest repressor domains in human cells (Witzgall et al. 1994; Margolin et al. 1994; Alerasool et al. 2020; Tycko et al. 2020), and is associated with spreading of heterochromatin and epigenetic memory through positive feedback mechanisms 8,15–22. KRAB-mediated gene silencing operates through recruitment of cofactors, including KAP1, HP1, and SETDB1, that read and write the repressive histone modification, histone 3 lysine 9 trimethylation (H3K9me3), creating a positive feedback loop that establishes stable gene silencing (Ayyanathan et al. 2003). Through this type of feedback mediated by the ability of KAP1 to recruit HP1, spreading of epigenetic effects beyond the target locus can affect nearby genes in a distance-dependent manner (Groner et al. 2010).Targeted recruitment of the KRAB domain from ZNF10 has been shown to repress gene expression, and lead to loss in histone 3 acetylation and gain of H3K9me3 across several tens to hundreds of kilobases around the target gene depending on the method and duration of recruitment (Amabile et al. 2016; Groner et al. 2010; Feng et al. 2020). However, the spatial-temporal dynamics of these effects and the capacity of the commonly used cHS4 insulator to influence KRAB-mediated spreading of silencing have not been systematically characterized. In addition, the dynamics of reactivation at neighboring genes after removal of KRAB has also not been well characterized in a synthetic system with variable distance or insulators. Understanding the drivers of reactivation could also be important for diseases, where developmentally silenced genes reactivate(Das and Chadwick 2021).”

New HDAC4 paragraph (previously at line 108): We wanted to compare the effects of KRAB with those of another fast silencer that is not associated with histone methylation positive feedback, and instead is only associated with removal of acetylation. We have previously shown that HDAC4 recruitment leads to fast gene silencing, comparable to that of KRAB(Bintu et al. 2016). Therefore, we expected silencing of the targeted gene to occur at a similar rate as with KRAB, while differences in the downstream gene silencing would be due to the lack of positive feedback with HDAC4 recruitment.

4. Line 74: "a set of mammalian cell lines" could be advantageously replaced by "two mammalian cell lines".

We agree, it is better to specify exactly how many cell types we have tested. We have added additional data with another human cell line, HEK293T with the reporters integrated at the AAVS1 locus, as a new supplemental figure to Figure 1. As a result, we now have three cell lines, and, as suggested, we modified the text to remove “a set of mammalian cell lines” from the first sentence, and replace it with: “we site-specifically integrated our reporters in three cell types”, and also added “and HEK293T cells” to line 84.

5. A statistical test should be performed to qualify the yellow and red subpopulations.

We have now included a statistical test between the fraction of cells off in +dox/-dox for mCitrine and mCherry cell population replicates. The results from the statistical tests is now included in Appendix 1- table 1, which includes percent of cells off in each condition with the standard error of mean (SEM) and p-value from a Welch’s unequal variances T-test comparing the percent of cells off in -dox versus +dox for each gene. All comparisons were shown to be statistically significant (p<0.05), indicating that silencing of both genes is reproducible at the indicated timepoints.

This is referenced in the main text (line 112, added sentence) and the legend for Figure 1: “ A Welch’s unequal variances T-test comparing the percent of cells off in -dox versus +dox for each gene, showed that the percent of cells silenced were statistically significant and reproducible for NS and 5kb reporters in all cell types tested (Appendix 1 - table 1).”

Note, that we also added a third replicate to the 20 day recruitment of HDAC4, which in our original visualization only had 2 replicates (Figure 1H,J).

New table legend:

Appendix 1 - table 1. Statistics of reporters after silencing. Percent of cells off with standard error of mean (SEM) for each gene in the -dox and +dox condition after the number of days indicated, with threshold drawn in Figure 1 (dotted line). Welch’s unequal variances T-test was run on the percent cells off from each replicate (n=3) and p-value is shown for comparison of -dox versus +dox for each gene.

6. A spacing of 0 or 5kb hardly qualifies as "various spacer lengths" as stated on line 79.

We deleted the “various spacer lengths”. Instead, the text reads: “we separated the two fluorescent genes by either no spacer (NS) (Figure 1A) or 5kb λ phage DNA (5kb) (Figure 1B).”

7. Lines 112-114: spreading from the HDAC4 docking site takes 20 days, is observed in only a fraction of cells, and the interpretation of the results is complicated by promoter interference intrinsic to the vector design (spreading is more efficient in the 5kb- than in the 0kb-spacing configuration). The data thus do not warrant the statement "these results show for the first time that silencing mediated by a histone deacetylase can also extend to nearby genes".

It is true that most protein coding genes in mammalian cells are generally spaced farther apart. However these distances are common, and in fact desired, for synthetic circuits, including expression of antibiotic markers. We have modified the text to reflect this emphasis:

“These results show that in a synthetic reporter system, where genes are separated by short distances up to 5kb, silencing mediated by a histone deacetylase can also extend to a nearby gene.”

In addition, we have tested the effects of HDAC4 in a different human cell line (HEK293T) with the 5kb reporter integrated at the AAVS1 locus, and the silencing of mCherry occurs in 7 days, much faster than in K562 and more similar to CHO-K1. We added these data as a Figure 1 —figure supplement 2, as mentioned above.

8. The models presented in 2F and 2G are not clear: The way the picture is drawn, it seems that loss of H3Kac is a temporal prerequisite of H3K9me3 and H3K27me3 deposition in both models. This might be true for HDAC4 (although the authors create some controversy on this particular point later in the manuscript), but not for KRAB.

Indeed, good point, the removal of histone acetylation need not happen first, we have changed this cartoon to remove the implication that acetylation is a temporal prerequisite to methylation (see new Figure 2I).

9. Line 299: reference to the figure should be corrected (Figure 3GandH Figure 4GandH, Figure 4—figure supplement 2).

Thank you, we have corrected this error.

10. Lines 331-333: "These results suggest that the insulators can help the downstream gene remain active in conditions where its silencing is already slow, such as after indirect silencing by HDAC4 recruitment in K562." Again, this hypothesis relies on so few observations (one locus, one insulator, one cell line, one CR, one spacer) that it should be formulated with much caution.

We have rephrased this sentence as follows:

“These results suggest that in K562, where silencing by HDAC4 recruitment has slower dynamics compared to CHO, the cHS4 insulators can help the downstream gene mCherry remain active.”

11. Lines 510-515: one cannot say that the referenced previous studies yielded "disparate" results, notably on the extent of KRAB-induced heterochromatin spreading. Compared to the present work, they covered far greater variety of settings relative to the strengths of the targeted promoters, their distance from the KRAB docking site and their immediate or broader chromatin environment, so that a broad range of phenotypes was expected.

We have rephrased this sentence as follows:

“Our results help frame the extent of spreading of KRAB-mediated silencing as a dynamic process that depends on the time and strength of recruitment as well as the level of activation at the target locus, which can be influenced by neighboring promoters and insulators.”

12. The authors should integrate in their thinking results reported ten years ago (BMC Genomics 12:378, 2011). This work reported an extensive analysis of chromatin features conducive or not of KRAB-induced long-range silencing, making interesting conclusions on the "repression-primed" state of the most sensitive promoters and their paradoxical location in a surrounding of highly expressed genes, and on the failure of CTCF recruitment to exert any barrier effect on H3K9me3 spreading.

We have integrated the findings from this work into our discussion:

– Line 520, added text: “Additionally, in our system, we see that silencing is slower and occasionally inhibited by the presence of the cHS4 insulator, which contains high levels of acetylation. In our model, we propose that genes in active regions can also cooperate with neighboring genes in addition to insulators to oppose silencing by chromatin regulators. However, there are other factors at play in targeting of endogenous loci which affect gene silencing. In previous studies, targeting dCas9KRAB to hundreds of repeated sgRNA sites forms a large domain of H3K9me3 heterochromatin on the order of megabases in a few days, but does not result in widespread gene silencing, rather the silencing of genes is controlled by the loss of H3K27Ac and H3K4me3 (Feng et al. 2020). Additionally previous studies have also shown that while genes that are more susceptible to KRAB-mediated silencing with KAP1 have higher levels of repressive histone marks at the promoter and gene body, though surprisingly they have higher levels of active histone marks surrounding the gene (Meylan et al. 2011), indicating that the genomic locus affects silencing by KRAB. Therefore, more work needs to be done to understand how the genomic and epigenetic environment surrounding a group of genes can affect their repressibility.

13. The authors should also comment on KRAB zinc finger protein genes, many of which recruit KAP1 at their 3' end and are grouped in tightly packed clusters, yet can escape self- or mutually inflicted silencing. And on the interesting observation by K. Helin's group that the H3K9me1/2 demethylase PHF8 is recruited to the promoter of many of these genes, perhaps explaining this phenomenon (Mol Cell 2010 38: 165).

We have added this to the discussion by editing sentence on line 521-523:

Original: “First, in mammalian genetic circuits where multiple genes need to be integrated in the same cell, genes controlled by KRAB need to be placed far away from other genes to avoid unwanted interference and feedback.”

New: “First, in mammalian genetic circuits where multiple genes need to be integrated together in a cell, genes controlled by KRAB need to either be placed far away from other genes to avoid unwanted interference and feedback, or other mechanisms of escaping silencing potentially through recruitment of H3K9me demethylases such as PHF8 should be considered (Kleine-Kohlbrecher et al. 2010).”

14. The potential roles of histone acetylation functioning as insulator sequences require more rigorous testing and validation. Perhaps the author should mine some published genomic data to find potential correlations between various histone acetylation marks and heterochromatin vs euchromatin compartments in development or disease conditions.

We agree that using genomic data to connect histone modifications with insulators which help protect euchromatin from heterochromatin is a good approach. A recent paper has done this and used ENCODE ChIP-seq data to identify insulators, and found candidates which were then screened for their ability to prevent transgene silencing. They additionally show that when tested in the presence of an HDAC inhibitor, the expression of the insulated transgene increased. We have added this in our discussion along with more references to previous studies showing that the HS4 insulator’s capabilities are associated with acetylation and previous models that have characterized the role of insulators as activating forces which combat repressive forces:

– Line 547-549, add to sentence: “We can explain the function of insulators on spreading and background silencing by thinking of them as areas in an active chromatin state, that can for example recruit additional writers of acetylation(Zhao and Dean 2004), and other properties similar to promoters(Raab and Kamakaka 2010).”

– Line 549-550, add: “It has been shown that the core region of the HS4 insulator has high levels of acetylation (Mutskov et al. 2002; Zhao and Dean 2004), and it has been previously proposed that the HS4 insulator protects from spreading of H3K9me3 through maintenance of high levels of acetylation (Litt et al. 2001; Mutskov et al. 2002; Ghirlando et al. 2012). A recent study (Rudina and Smolke 2019) used ChIPseq ENCODE data of histone modifications, including the presence of active acetylation marks and absence of repressive methylation marks, to identify candidate insulator sequences which were then screened for their ability to prevent background transgene silencing. Additionally, when these insulators were tested in the presence of an HDAC inhibitor, the expression of the insulated transgene increased (Rudina and Smolke 2019).

– Line 550, new paragraph: “Previous models have proposed the idea that insulators, similar to enhancers and transcription factors, are anti-silencers which oppose repressive forces (Fourel et al. 2004), and transcriptional activators have been shown to insulate against heterochromatin and even drive reactivation (Sutter et al. 2003). Our findings are in line with these models, and we further propose that active regions, such as insulators and promoters, cooperate with each other to inhibit gene silencing and drive gene reactivation.”

We have also added text to reference previous work that has shown strong correlations between acetylation and euchromatin, line 127:

– “Previous studies looking at histone modifications across the genome in different cell types have shown that active promoters and strong enhancers in euchromatin have high levels of acetylation (Ernst et al. 2011), which is in agreement with the general histone code by which acetylation confers open and active regions of euchromatin, while methylation results in silenced and compact heterochromatin (Jenuwein and Allis 2001).”

We have added information about facioscapulohumeral dystrophy (FSHD), which is a disease that is potentially caused by loss of spreading of heterochromatin causing aberrant gene activation:

– Line 568, add to sentence: “and perhaps in understanding aging where loss of heterochromatin can lead to changes in gene expression (Villeponteau 1997) or in diseases such as facioscapulohumeral dystrophy (FSHD). FSHD is a neuromuscular disorder that is potentially caused by loss of spreading of heterochromatin causing aberrant gene activation(Kleinjan and van Heyningen 1998; Gabellini et al. 2002; Hahn et al. 2010).”

Reviewer #1 (Recommendations for the authors):The authors should mention in the text of the paper what is known about the differences in half-life/stability of the mCitrine and mCherry reporter proteins. This is important to assess the kinetics of repression. While imaging and flow cytometry allow assessment of their expression at the single cell level, additional RT-qPCR experiments for mRNA levels for each gene would strengthen the analyses.

We agree with the reviewer that the half-life/stability of the reporter proteins are important parameters for the kinetics of repression. Our reporter genes are fusions between H2B and a fluorescent protein: H2B-mCitrine and H2B-mCherry. Fusions between H2B and fluorescent proteins (H2B-FPs) are used for studying chromatin dynamics due to their high signal to noise ratio from nuclear localization and incorporation into nucleosomes without inhibiting cell cycle progression (Kanda et al. 1998; Hadjantonakis and Papaioannou 2004; Stewart et al. 2009; Fraser et al. 2005). H2B-FPs are very stable, with very long protein halflives on the order of weeks; therefore degradation is mostly driven by cell division (Morcos et al. 2020). At the mRNA level, we have previously shown that the half-life of H2B-mCitrine mRNA transcript is about 3.9 hours as measured by inhibiting transcription with actinomycin D and quantifying mRNA overtime (Bintu et al. 2016). Although we have not measured the lifetime of the H2B-mCherry mRNA our video data and our model from daily flow cytometry data shows that the delay between mCitrine and mCherry silencing (δ T) is close to zero in the NS reporters (Figure 3Band3I), suggesting that mCitrine and mCherry lifetimes are very close together. Small differences in mRNA stability could change silencing times of each gene but ultimately would shift the δ T delay equally for all reporter constructs, and therefore not change our conclusion that the delay increases with distance, which is also corroborated by our video results which measure silencing at the protein level.

We have added this information into the main text as follows:

– Line 73-76, first sentence edited:

– Original: “ To study the spreading of transcriptional silencing and chromatin modifications, we engineered a set of mammalian cell lines containing two neighboring fluorescent reporter genes and a CR, with each gene driven by constitutive promoters (Figure 1A-B, Figure 1—figure supplement 1).”

– New: “To study the spreading of transcriptional silencing and chromatin modifications, we site-specifically integrated reporters consisting of two neighboring fluorescent reporter genes (mCitrine and mCherry) with each gene driven by a constitutive promoters (Figure 1A-B, Figure 1–figure supplement 1). Both fluorescent proteins are fused to H2B; this approach increases the signal to noise ratio due to nuclear localization and is commonly used for studying chromatin dynamics (Kanda et al. 1998; Hadjantonakis and Papaioannou 2004; Stewart et al. 2009; Fraser et al. 2005). In each reporter cell line, we also stably integrated a recruitable rTetR fused to either KRAB or HDAC4.”

– Line 195, added sentence: “Since H2B fusions to fluorescent proteins have long halflives on the order of weeks, their loss is mostly driven by cell division (Morcos et al. 2020), which allows the silencing of the gene to be detected based on the change in fluorescent protein accumulation. Stitching together single-cell traces across cell divisions results in cumulative traces of fluorescent protein levels over time for specific lineages (Figure 3A, Figure 3 —figure supplement 1). The slope of these traces is then proportional to promoter activity, and its decrease can be used to determine silencing of a gene(Materials and methods).”

– Line 218, add to sentence:

“… especially if we consider the additional time necessary for mRNA degradation of about ~4 hours included in the flow fits (Materials and methods), as determined previously by inhibiting transcription with actinomycin D and quantifying the decay of H2B-mCitrine mRNA over time(Bintu et al. 2016). Since we observed a delay time between mCherry and mCitrine silencing of less than 4 hrs after KRAB recruitment (Figure 3B), we assumed that the H2B-mCherry mRNA degrades at a similar rate as H2BmCitrine.”

The authors do not show kinetics and overall expression of the Dox-induced KRAB and HDAC proteins and how these levels compare with the endogenous cellular levels. If as may be expected, the levels are much higher than endogenous, how does this affect the interpretation of the results in terms of kinetics and spreading of repression?

Indeed, we would expect the expression levels of our chromatin regulators to be higher than endogenous levels, and we also expect the amount of rTetR-CR recruited at the reporter to affect the dynamics of silencing.. In fact, with our dox inducible system we are able to tune the number of rTetR-KRAB or rTetR-HDAC4 proteins that can be recruited to the reporter locus by altering dox concentration. As shown in Figure 4 – supplement 4, at lower dox concentrations, in the insulator reporters silencing of mCherry still occurs in cells where mCitrine is silenced. We find that although the fraction of the population that silences mCitrine is lower, in the cells with mCitrine silent, silencing spreads to the downstream mcherry within 5 days for both HDAC4 and KRAB. We have also collected additional flow cytometry data with KRAB recruitment to NS and 5kb reporters at lower dox concentrations. As seen with the insulator reporters, we find that lower dox concentration reduces the fraction of cells that silence but that in cells where mCitrine is silenced, mCherry silencing follows in both NS and 5kb reporters. Although the lower dox concentration slows down the rate of spreading, it does not alter coordinated dynamics of the two genes. In the NS reporter, both genes silence simultaneously at 4 ng/mL (Figure 1 —figure supplement 3A). In the 5kb reporter after 5 days at lower dox concentrations, mCitrine still silences first with slower spreading to mCherry, this progression can be seen in the 4 ng/mL and 10 ng/mL concentrations respectively (Figure 1 —figure supplement 3B).

We have now added this additional data as Figure 1 —figure supplement 3, and it is referenced in the main text as follows:

– Line 107, added sentence: “Tuning the probability of of KRAB recruitment at the reporter by varying dox concentration, we found that spreading of silencing still occurs in both the NS and the 5kb reporters, and the percentage of cells with both genes silenced increases with dox concentration (Figure 1 —figure supplement 3).”

New figure legend:

Figure 1 —figure supplement 3. Silencing by KRAB at lower dox concentrations.

(A-B) 2D scatter plots of mCitrine and mCherry fluorescence measured by flow cytometry in CHO-K1 cells with KRAB recruitment for 5 days at 4 ng/mL, 10 ng/mL and 1000 ng/mL in the (A) NS reporter or (B) 5kb reporter. Percent of cells in each quadrant is shown with standard error of mean (n=2).

The authors have investigated the role of the cSH4 insulator element, can they better justify why this one was chosen. As the properties of insulator elements are not as well described as promoters or enhancers, experiments with other insulators would be required before general statements can be made on their ability to block repression or contribute to reactivation of gene expression.

We agree with this comment and we have:

1) modified the text to better explain why we chose the cHS4 element , and 2) edited the conclusions we draw about the ability to block repression to refer only to the cHS4 insulator and its derivatives we have tested as indicated below.

We have added more reasoning as to why we chose to use the cHS4 insulator:

– Line 44, add sentence: “The cHS4 insulator is a commonly used insulator in synthetic biology in mammalian cells (Recillas-Targa et al. 2004), as it has been shown act as a barrier against heterochromatin (Recillas-Targa et al. 2002; BurgessBeusse et al. 2002), thus preventing transgene silencing in many cell lines including CHO and K562 (Pikaart et al. 1998; Walters et al. 1999; Mutskov et al. 2002; RecillasTarga et al. 2004; Zhang et al. 2017).”

– Line 276, edit first part of sentence to : “The cHS4 insulator is a commonly used insulator for protecting from transgene silencing in many cell lines(Recillas-Targa et al. 2004); …”

We have clarified the conclusions are restricted to the cHS4 insulator:

– Line 62: “different insulator arrangements” changed to “the cHS4 insulator and its core region”

– Lines 285, 289, 291, 298, 302, 318, 320, 322, 324, 326, 334, 337, 346, 354, 372, 385,

389, 395, 504 specify: “cHS4 insulator(s)”/”insulator configurations” in place of “insulator(s)”

Line 473: “Similar to promoters, the core region of the cHS4 insulator”

– Modified Figure 6B-D and 6HandI to replace “insulator core” with “cHS4 insulator core”.

In Figure 6 legend, we changed “Insulators” to “The core element of the cHS4

insulator”

Line 555, edit sentence:

– Original: “Screening different insulators in a similar recruitment assay in the future could help go beyond the binary classification of sequences as insulators or non-insulators and actually quantify how well they perform against different mechanisms of silencing.”

– New: “Screening different sequences with insulating function, such as Matrix Attachment Regions (MARs) and ubiquitous chromatin opening elements

(UCOEs)(Rudina and Smolke 2019), along with other insulators such as H19Igf2 and SNS (Kaffer et al. 2000; Di Simone et al. 2001), in a similar recruitment assay in the future could help go beyond the binary classification of sequences as insulators or non-insulators and help specify and quantify how well they perform against different mechanisms of silencing.”

It is now well accepted that regulation of gene expression is mainly driven by enhancer elements. The reporter plasmid used in this study comprises only proximal promoter elements. Do the authors know whether their promoter activity is driven/influenced by distal enhancers located around the integration sites? More specifically it would be informative to generate a reporter with a strong enhancer element and assess how this impacts repression. Insertion of an enhancer between the two genes may reveal whether it can block the spreading of repression to the mCHERRY gene.

We agree this would be an interesting idea to test, potentially in a high-throughput screen that could look at different enhancers as well as different insulators. We have added this idea to the discussion:

– Line 559, add sentence: “Additionally, it would also be interesting to test enhancers between the two reporter genes to study whether they can also block spreading of silencing to the downstream gene.”

We’ve also highlighted previous work on the effect of recruiting KRAB to an enhancer:

– Line 516: “For example, when KRAB is recruited for a short period at an enhancer in the hemoglobin locus in K562, the extent of spreading of methylation is very small (<1 kb)(Thakore et al. 2015).”

Reviewer #2 (Recommendations for the authors):Introduction:1. The authors should better explain why they chose KRAB and HDAC4 for their experiments. They should also comment on the histone deacetylase activity recruited by the KRAB-binding KAP1. KRAB is not a protein but a domain, which often but not always recruits KAP1, and some KRAB-containing proteins also harbour other domains and some are not repressors. This should be explained as well to help the non-specialists situate the relevance of the work in a broader context.

Indeed, we have modified the paragraphs introducing KRAB and HDAC4 to better reflect why we chose them, as well as the known biology of KRAB as follows:

New KRAB paragraph (lines 46-52): “The Kruppel associated box (KRAB) repressive domain from zinc finger 10 is commonly used in synthetic biology applications (Nakamura et al. 2021), is one of the strongest repressor domains in human cells (Witzgall et al. 1994; Margolin et al. 1994; Alerasool et al. 2020; Tycko et al. 2020), and is associated with spreading of heterochromatin and epigenetic memory through positive feedback mechanisms 8,15–22. KRAB-mediated gene silencing operates through recruitment of cofactors, including KAP1, HP1, and SETDB1, that read and write the repressive histone modification, histone 3 lysine 9 trimethylation (H3K9me3), creating a positive feedback loop that establishes stable gene silencing (Ayyanathan et al. 2003). Through this type of feedback mediated by the ability of KAP1 to recruit HP1, spreading of epigenetic effects beyond the target locus can affect nearby genes in a distance-dependent manner (Groner et al. 2010).Targeted recruitment of the KRAB domain from ZNF10 has been shown to repress gene expression, and lead to loss in histone 3 acetylation and gain of H3K9me3 across several tens to hundreds of kilobases around the target gene depending on the method and duration of recruitment (Amabile et al. 2016; Groner et al. 2010; Feng et al. 2020). However, the spatial-temporal dynamics of these effects and the capacity of the commonly used cHS4 insulator to influence KRAB-mediated spreading of silencing have not been systematically characterized. In addition, the dynamics of reactivation at neighboring genes after removal of KRAB has also not been well characterized in a synthetic system with variable distance or insulators.Understanding the drivers of reactivation could also be important for diseases, where developmentally silenced genes reactivate(Das and Chadwick 2021).”

New HDAC4 paragraph (previously at line 108) : We wanted to compare the effects of KRAB with those of another fast silencer that is not associated with histone methylation positive feedback, and instead is only associated with removal of acetylation. We have previously shown that HDAC4 recruitment leads to fast gene silencing, comparable to that of KRAB (Bintu et al. 2016). Therefore, we expected silencing of the targeted gene to occur at a similar rate as with KRAB, while differences in the downstream gene silencing would be due to the lack of positive feedback with HDAC4 recruitment.

Regarding Figure 11. Line 74: "a set of mammalian cell lines" could be advantageously replaced by "two mammalian cell lines".

We agree, it is better to specify exactly how many cell types we have tested. We have added additional data with another human cell line, HEK293T with the reporters integrated at the AAVS1 locus, as a new supplemental figure to Figure 1. As a result, we now have three cell lines, and, as suggested, we modified the text to remove “ a set of mammalian cell lines” from the first sentence, and replace it with: “we site-specifically integrated our reporters in three cell types”, and also added “and HEK293T cells” to line 84.

2. A statistical test should be performed to qualify the yellow and red subpopulations.

We have now included a statistical test between the fraction of cells off in +dox/-dox for mCitrine and mCherry cell population replicates. The results from the statistical tests is now included as a Appendix 1 - table 1, which includes percent of cells off in each condition with the standard error of mean (SEM) and p-value from a Welch’s unequal variances T-test comparing the percent of cells off in -dox versus +dox for each gene. All comparisons were shown to be statistically significant (p<0.05), indicating that silencing of both genes is reproducible at the indicated timepoints.

This is referenced in the main text (line 112, added sentence) and the legend for Figure 1: “A Welch’s unequal variances T-test comparing the percent of cells off in -dox versus +dox for each gene, showed that the percent of cells silenced were statistically significant and reproducible for NS and 5kb reporters in all cell types tested (Appendix 1 - table 1).”

Note, that we also added a third replicate to the 20 day recruitment of HDAC4, which in our original visualization only had 2 replicates (Figure 1H,J).

New table legend:

Appendix 1 - table 1. Statistics of reporters after silencing. Percent of cells off with standard error of mean (SEM) for each gene in the -dox and +dox condition after the number of days indicated, with threshold drawn in Figure 1 (dotted line). Welch’s unequal variances T-test was run on the percent cells off from each replicate (n=3) and p-value is shown for comparison of -dox versus +dox for each gene.

3. A spacing of O or 5kb hardly qualifies as "various spacer lengths" as stated on line 79.

We deleted the “various spacer lengths”. Instead, the text reads: “we separated the two fluorescent genes by either no spacer (NS) (Figure 1A) or 5kb λ phage DNA (5kb) (Figure 1B).”

4. Lines 111-112: "Moreover, in K562 cells, HDAC4-mediated silencing of mCherry was much slower, taking up to 20 days, compared to 5 days in CHO-K1 cells (Figure 1HandJ). Any lead to explain the difference? Shouldn't other chromosomal loci be tested before attributing the difference to the cellular background?

We agree that in general we expect both genomic locus and cell type to be able to affect the speed of spreading, and we have emphasized in our discussion the interesting results at different genomic loci reported in (Meylan et al. 2011), please see response to Discussion point 5 below.

In the case of HDAC4 recruitment, we believe that the silencing of pRSV-mCherry is due to background silencing of the pRSV promoter by PRC2 in the absence of protective activation by pEF. pRSV promoter is stronger in K562 cells, as indicated by higher mCherry expression compared to CHO-K1 cells (Figure 3 —figure supplement 3), therefore it likely can resist this background silencing better. This is consistent with the fact that in K562 we also see faster reactivation of the pRSV-mcherry (Figure 5). In support of this point, we also added text pointing to previous work comparing the two promoters: “In line with our hypothesis, previous studies have shown that the pRSV promoter is more prone to transgene background silencing than the pEF promoter when integrated into mammalian genomes (Garrison et al. 2007).”

Moreover, we were curious what happens if we change the cellular background while still having the reporter integrated at the AAVS1 locus: so we created HEK293T lines with the 5kb reporter at the AAVS1 locus, and recruited either KRAB and HDAC4 (see new figure, Figure 1 - figure supplement 2). We find that KRAB-mediated silencing can consistently spread over 5kb, consistent with the CHO-HAC and K562 AAVS1 results. With HDAC4-mediated recruitment at the 5kb dual reporter in HEK293T at the AAVS1 locus, the majority of cells that silence mCitrine also silence the downstream mCherry within 7 days, similar to the CHO-HAC system, and much faster than in K562. These results suggest that cellular background can play a strong role in the speed of silencing and spreading, as all other factors in the systems are held constant such as reporter set up and genomic locus.

We have edited the paragraph from line 86-114 to reflect the addition of these new data:

Lines 86-90: “To investigate the spreading of transcriptional silencing, we first recruited KRAB upstream of the mCitrine gene for 5 days in CHO-K1, K562 and HEK293T cell lines with the addition of dox (Figure 1A-B). Using flow cytometry to measure fluorescence intensity, we observed silencing of both the upstream mCitrine gene and the downstream mCherry gene in the NS (Figure 1C-D) and 5kb (Figure 1E-F, Figure 1 - figure supplement 2A) reporter lines for the cell types tested.

–Line 112, add sentences: “ With HDAC4-mediated recruitment at the 5kb dual reporter in HEK293T at the AAVS1 locus, the majority of cells that silence mCitrine also silence the downstream mCherry within 7 days (Figure 1 - figure supplement 2B), similar to the CHO-HAC system, and much faster than in K562. These results suggest that cellular background can play a strong role in the speed of silencing and spreading, as all other factors in the systems are held constant such as reporter set up and genomic locus.”

New figure legend:

Figure 1 —figure supplement 2. Recruitment of KRAB and HDAC4 to 5kb reporter in HEK293T.

(A-B) Fluorescence distributions of mCitrine and mCherry measured by flow cytometry either without CR recruitment (-dox) or after 5 or 7 days of recruitment (+dox) of either (A) KRAB or (B) HDAC4 at the 5kb reporters in HEK293T. Biological replicates of multiclonal populations are shown as overlaid semi-transparent distributions (n=3).

5. Lines 112-114: spreading from the HDAC4 docking site takes 20 days, is observed in only a fraction of cells, and the interpretation of the results is complicated by promoter interference intrinsic to the vector design (spreading is more efficient in the 5kb- than in the 0kb-spacing configuration). The data thus do not warrant the statement "these results show for the first time that silencing mediated by a histone deacetylase can also extend to nearby genes". Rare are the pairs of genes, the promoters of which are at such short distances.

It is true that most protein coding genes in mammalian cells are generally spaced farther apart. However these distances are common, and in fact desired, for synthetic circuits, including expression of antibiotic markers. We have modified the text to reflect this emphasis:

“These results show that in a synthetic reporter system, where genes are separated by short distances up to 5kb, silencing mediated by a histone deacetylase can also extend to a nearby gene.”

In addition, we have tested the effects of HDAC4 in a different human cell line (HEK293T) with the 5kb reporter integrated at the AAVS1 locus, and the silencing of mCherry occurs in 7 days, much faster than in K562 and more similar to CHO-K1. We added these data as a Figure 1 —figure supplement 2, as mentioned above.

Regarding Figure 21. The models presented in 2F and 2G are not clear: The way the picture is drawn, it seems that loss of H3Kac is a temporal prerequisite of H3K9me3 and H3K27me3 deposition in both models. This might be true for HDAC4 (although the authors create some controversy on this particular point later in the manuscript), but not for KRAB.

Indeed, good point, the removal of histone acetylation need not happen first, we have changed this cartoon to remove the implication that acetylation is a temporal prerequisite to methylation (see new Figure 2I).

2. Lines 181-183: "… that can lead to spreading of these chromatin modifications (Figure 2F). This observation suggests that silencing of the RSV promoter is indirectly due to PRC2 action, but it can only take place once HDAC4 removes acetylation at the locus." Artificial recruitment of PRC2 at this locus in the absence of HDAC4 could be used to validate this hypothesis.

We agree that recruitment of PRC2 to our reporter would help support our hypothesis. We generated a CHO-K1 cell line where EED, from PRC2, is recruited upon addition of dox. We see that upon EED recruitment to the NS reporter that both genes are silenced (Figure 2 —figure supplement 3B). We go on further to test the effects of PRC2 inhibition on HDAC4mediated silencing (see comments to point 3 regarding figure 3).

Regarding Figure 3The authors used time-lapse movies to analyse the spreading of epigenetic marks with high temporal resolution. They conclude that KRAB-mediated spreading of transcriptional silencing is distance-dependent in CHO-K1 cells while HDAC4-mediated spreading is distance-independent at these length scales.1. A limitation of time-lapse movies is the low output. However, the authors developed a mathematical model to optimize the information recovered from "high output" data (flow cytometry combined with gene expression). The output of this method is nicely consistent with time-lapse movies opening perspectives to use this system on a larger scale. A useful contribution to the field!

Thank you!

2. There should be published data / databases allowing to verify that in broader contexts HDAC inhibition triggers silencing likely via H3K27me3 deposition (Line 176), rather than it being a peculiarity of the dual system used here.

We’ve modified the text to describe published results which support our hypothesis that acetylation removal can lead to PRC2-associate chromatin modifications: both here (line 183) and in our Discussion section:

– Added two sentences at line 183: “ In accordance with our hypothesis, previous studies have shown that by co-ChIP of primary cells H3K27me3 and H3Ac are negatively correlated (Weiner et al. 2016), that lack of transcription and histone deacetylation both drive an increase in H3K27me3 deposition steadily over time (Hosogane et al. 2016), and that recruitment of HDAC1 leads to repressed chromatin associated with a strong increase in H3K27me3 (Song et al. 2016). Additionally, knockdown of p300, a histone acetyltransferase, resulted in a global decrease of H3K27Ac in tandem with an increase in H3K27me3 (Martire et al. 2020).”

– In Discussion section, line 536 added: “In accordance with our observations, previous studies have shown that H3K27me3 and H3Ac are negatively correlated on individual nucleosomes (Weiner et al. 2016), and specifically that deacetylation by either lack of transcription or recruitment of HDAC1 leads to increases in H3K27me3 (Hosogane et al. 2016; Song et al. 2016). Additionally, studies where cells are treated with HDAC inhibitor show in an increase in histone acetylation which is correlated with reduced recruitment of EZH2, EED and Suz12 proteins from the PRC2 as well as a decrease in H3K27me3 (Reynolds et al. 2012; Wang et al. 2012), showing the bidirectionality of the inverse correlation between H3K27me and H3K27Ac (Pasini et al. 2010). This supports our observation that low levels of acetylation allow PRC2 to work more effectively.”

3. Further down (lines 178-180) it is stated that "This repressive modification, which is associated with polycomb repressive complex 2 (PRC2), has not been reported in association with HDAC4, and was not observed after HDAC4 recruitment at a pEF-mCitrine reporter flanked by insulators in the same locus of CHO-K1 cells". To consolidate their hypothesis, the authors could test silencing of the locus in the presence of EZH2 inhibitor to see in which extent it depends on PRC2, and measure PRC2 enrichment by CUTandTAG or CUTandRUN.

Thank you for this suggestion! To further test this hypothesis of PRC2-mediated silencing of the downstream pRSV-mCherry gene, we measured HDAC4-mediated silencing in the presence of EZH2 inhibitor Tazemetostat (Taz). We found that Taz blocked pRSV-mCherry silencing, while not affecting the pEF-mCitrine silencing. These results support the idea that pRSV-mCherry silencing we observe after HDAC4-mediated silencing of the pEF-mCitrine results from PRC2 mediated silencing of the pRSV-mCherry. We have added these data to Figure 2F-H as well as new supplemental Figure 2 —figure supplement 3. Additions to the main text include:

– Line 180, added text: “To test whether the silencing of mCherry was mediated by PRC2, we recruited HDAC4 in the presence of Tazemetostat (Taz), a small molecule inhibitor of EZH2, which is the H3K27 methyltransferase in PRC2 (Figure 2F). First, we tested Taz in the NS reporter line with recruitment of EED, the H3K27 reader in PRC2, and showed that inhibition reduced silencing of mCitrine and mCherry (Figure 2 —figure supplement 3A-B). We found that in the presence of dox and Taz while the mCitrine gene was still silenced by HDAC4, the mCherry gene was not silenced and even slightly increased in expression in the 5kb reporter (Figure 2G-H) and the NS reporter (Figure 2 —figure supplement 3C).

– Line 419, add to sentence: “loss of pRSV-mCherry silencing upon PRC2 inhibition

(Figure 2G-H), and the fact that the silencing delay is distance-independent.”

– Line 536, add to sentence: “which was confirmed by the loss of mCherry silencing when PRC2 was inhibited (Figure 2F-H).“

– Line 21, edit sentence in abstract to reflect the role of PRC2: “Silencing by histone deacetylase HDAC4 of the upstream gene can also facilitate background silencing of the downstream gene by PRC2, but with a days-long delay that does not change with distance.”

– Edits to legend of Figure 2:

(F) Addition of Tazemetostat (Taz) inhibits the EZH2 methyltransferase from PRC2. (G) Fluorescence distributions of mCitrine (left) and mCherry (right) measured by flow cytometry either without CR recruitment (-dox, grey) or after 7 days of HDAC4 recruitment (+dox+DMSO, purple) and with Taz(+dox+Taz, pink) at the 5kb reporter in CHO-K1. (H) Percentages of cells OFF normalized by the -dox+DMSO condition, based on threshold shown in panel G (dotted line).

(I) In the absence of CR recruitment, both genes have H3Ac across the reporter (top). Upon recruitment of KRAB (bottom left), we see a loss of H3Kac and gain of H3K9me3 across the dual-gene reporter through both DNA looping from rTetR-KRAB as well as positive feedback loops for spread of methylation, resulting in a distance-dependent delay of transcriptional silencing between two genes. Upon recruitment of HDAC4 (bottom right), we see a loss of H3Kac as well as a gain of H3K27me3 across the reporter through positive feedback loops for spread of methylation by PRC2, resulting in distance-independent delay of transcriptional silencing between two genes.

New figure legend:

Figure 2 —figure supplement 3. EZH2 inhibitor Tazemetostat affects silencing after recruitment of EED and HDAC4.

(A) Addition of Tazemetostat (Taz) inhibits the EZH2 methyltransferase from PRC2. (B-C) Fluorescence distributions of mCitrine (top left) and mCherry (bottom left) measured by flow cytometry either without CR recruitment (-dox, grey) or after 7 days of (B) EED or (C) HDAC4 recruitment (+dox+DMSO, purple) and with Taz(+dox+Taz, pink) at the NS reporter in CHO-K1. Percentages of cells OFF normalized by the -dox+DMSO condition (right), based on threshold (dotted line).

4. Lines 223-226: "The observation that the delays of pRSV-mCherry silencing do not increase with distance (Figure 3EandJ) suggests that its silencing initiates at the RSV promoter, most likely by the action of PRC2 (Figure 2D, EandG).". If the silencing starts from the RSV promotor, why is it different without the 5Kb spacer? Can the dynamics of H3K27me3 deposition at the RSV promotor be assessed on both constructs? What would be the mechanism leading to the recruitment of PRC2 at the RSV instead of the pEF since HDAC4 is recruited at pEF? What is the effect of inverting the two promoters?

We show that although the silencing dynamics of KRAB silencing are dependent on distance between the two genes, with HDAC4 recruitment the silencing of the pRSVmCherry gene is independent of distance. If PRC2 were brought by HDAC4 at the pEF, we would expect the pRSV-mCherry silencing to take longer in the 5kb construct than in the NS, just as we see with KRAB (as spreading of the modifications depends on distance).Therefore, our hypothesis is that PRC2 gets recruited at the pRSV promoter independently of HDAC4, at all times. We have illustrated this point more clearly in Figure 6 (by adding PRC2 above the background repressive arrow targeting the pRSV-mCherry). The pEF promoter when active helps prevent background silencing of the pRSV-mCherry gene by PRC2 (as shown in Figure 6D, see new annotation). Therefore, when HDAC4 silences the pEF-mCitrine gene, the pEF can no longer prevent the background silencing of the downstream pRSV-mCherry gene by PRC2 (Figure 6C, see new annotation). This hypothesis is supported by the additional PRC2 inhibitor data we collected (where mCherry doesn’t get silenced upon HDAC4 recruitment, see Figure 2F-H and Figure 2—figure supplement 3. In addition to changing Figure 6 to reflect the new insights gained from the inhibitor data, we also modified the main text to clarify the logic presented above, as follow:

– Editing the sentence on lines 223-226 to: “We have determined that pRSV-mCherry silencing after HDAC4 recruitment is mediated by PRC2 (Figure 2F-H). If PRC2 were recruited directly by HDAC4 near the pEF promoter, we would expect the silencing delay to increase with distance, as we see with KRAB; however, we see that the delay with HDAC4 recruitment is distance-independent. Therefore, the observation that the delays of pRSV-mCherry silencing do not increase with distance upon HDAC4mediated silencing (Figure 3EandJ), together with the loss of mCherry silencing but not mCitrine silencing upon HDAC4 recruitment in the presence of Taz (Figure 2F-H), suggests that the silencing of mCherry is initiated at the pRSV promoter, by the action of PRC2 (Figure 2D, EandI).”

– We have also clarified an earlier sentence on line 177-183 that was inadvertently misleading the reader to think we propose HDAC4 directly recruits PRC2:

Original: “This repressive modification, which is associated with polycomb repressive complex 2 (PRC2), has not been reported in association with HDAC4, and was not observed after HDAC4 recruitment at a pEF-mCitrine reporter flanked by insulators in the same locus of CHO-K1 cells. Like H3K9me3 (Figure 2E), H3K27me3 has reader-writer positive feedback that can lead to spreading of these chromatin modifications (Figure 2F). This observation suggests that silencing of the RSV promoter is indirectly due to PRC2 action, but it can only take place once HDAC4 removes acetylation at the locus.”

New version: “This repressive modification is associated with polycomb repressive complex 2 (PRC2), and not with HDAC4. We have previously observed at a pEF-mCitrine reporter flanked by insulators in the same locus of CHO-K1 cells that HDAC4 recruitment does not lead to deposition of H3K27me3 (Bintu et al). Therefore, we hypothesized that the pRSV promoter gets silenced not by HDAC4, but by PRC2 after the pEF-mCitrine gene is silenced. Once the H3K27me3 modification is present, like H3K9me3 , H3K27me3 also has reader-writer positive feedback that can lead to spreading of these chromatin modifications (Figure 2I).”

Measuring the dynamics of H3K27me3 deposition by CUTandRUN are difficult to capture and to quantify as a function of time for such a heterogeneous population where some cells silence pRSV at 2 days and some at 3-5 days. We see these silencing times in movies where we can measure the fluorescent protein production rate in single cells; however, we cannot sort on these rates, since it takes time for the H2B-FP to dilute after silencing. In a bulk measurement of histone modifications, cells that have silenced and likely have histone modifications are mixed with cells that are not yet silenced and do not have modifications present. So the signal would likely be an average measurement of a heterogeneous population and not reflective of the true dynamics of the spread of the histone modifications.

We have also added evidence from previous works showing that the pRSV promoter is more prone to transgene silencing than the pEF promoter:

– Line 419, add sentence: “In line with our hypothesis, previous studies have shown that the pRSV promoter is more prone to transgene background silencing than the pEF promoter when integrated into mammalian genomes (Garrison et al. 2007).”

Additionally we agree that changing the promoters would be an interesting experiment to further test and expand our hypotheses summarized in Figure 6, and have added this point to our discussion.

– Line 539, add sentence: “Testing reporters with inverted promoters as well as testing other promoter combinations and configurations would be informative for building synthetic constructs where genes are in close proximity.”

Regarding Figure 4In this figure, the authors document the lack of effect of cHS4 insulators on KRAB/HDAC4-induced silencing in CHO cells, but observe some impact on the latter in K562 cells.1. Line 299: reference to the figure should be corrected (Figure 3GandH Figure 4GandH, Figure 4—figure supplement 2).

Thank you, we have corrected this error.

2. Lines 331-333: "These results suggest that the insulators can help the downstream gene remain active in conditions where its silencing is already slow, such as after indirect silencing by HDAC4 recruitment in K562." Again, this hypothesis relies on so few observations (one locus, one insulator, one cell line, one CR, one spacer) that it should be formulated with much caution.

We have rephrased this sentence as follows: “These results suggest that in K562, where silencing by HDAC4 recruitment has slower dynamics compared to CHO, the cHS4 insulators can help the downstream gene mCherry remain active.”

Regarding Figure 5Here the authors investigate the re-activation of the genes in their system. Again, they formulate overly general conclusions based on a very limited set of parameters. Lines 395-398: "These results show that the extent of epigenetic memory depends strongly not only on the chromatin regulator recruited, but also on the configurations of promoters and insulators at the target locus, with stronger promoters closely surrounded by core insulators diminishing memory".

We have rephrased this sentence as follows: “These results show that the extent of epigenetic memory in our system depends on the chromatin regulator recruited and the configurations of promoters and insulators at the target locus, with the stronger pEF promoter closely surrounded by cHS4 insulators diminishing memory.”

Regarding Figure 6 (Model)The model is fine as long as it is clearly restricted to the parameters tested here, and not proposed as a paradigm.

We agree that the original text introducing the model in Figure 6 was phrased in terms that were too general. We have now clarified that this model is a way of summarizing and making sense of all our results, and removed all generalizations throughout this section of the results and Figure 6, as follows:

– Original, Lines 401-404: “We used our gene dynamics and chromatin modifications data to develop a generalizable kinetic model that captures distance-dependent silencing associated with a tethered chromatin regulator and distills the roles of promoters and insulators as elements associated with high reactivation rates.” New version: “We developed a kinetic model that summarizes our observations of gene dynamics and chromatin modifications as a competition between the distancedependent silencing rates associated with each tethered chromatin regulator and the reactivation rates driven by our promoters and insulators.”

– Lines 428: “In summary, in our system both silencing and reactivation rates are affected by the distance between two genes.”

– Line 463: “In our system, promoters can be thought of as regions that are associated with high rates of gene activation and can drive reactivation after silencing.

– Line 473: “Similar to promoters, the core region of the cHS4 insulator can be modeled as a DNA element that increases the rate of reactivation of nearby genes in our setup in a distance-dependent manner (Figure 6H). This action, combined with our promoter reactivation rates,”

– Line 483: “Thinking of the promoters and core insulators in our synthetic constructs as elements associated with higher reactivation rates”

– Modified Figure 6B-D and 6HandI to replace “insulator core” with “cHS4 insulator core”.

– In Figure 6 legend, we changed “Insulators” to “The core element of the cHS4 insulator”

This phrasing now better matches the overall text in this section and the rest of the figure, where we refer to the specific elements used in our experiments: pEF, pRSV, cHS4 insulator core, KRAB, HDAC4, rather than general promoters, insulators or chromatin regulator.

Along the same line, we modified our abstract to replace “We propose a new model” with “Our data can be described by a model”. In addition, we have modified the discussion heavily to clearly delineate how this model that summarizes our data connects with previous experimental results and previous models proposed in the literature (see individual responses below).

Discussion1. Lines 510-515: one cannot say that the referenced previous studies yielded "disparate" results, notably on the extent of KRAB-induced heterochromatin spreading. Compared to the present work, they covered far greater variety of settings relative to the strengths of the targeted promoters, their distance from the KRAB docking site and their immediate or broader chromatin environment, so that a broad range of phenotypes was expected, without this heterogeneity being shocking (which is what the term "disparate" suggests).

We have rephrased this sentence as follows: “ Our results help frame the extent of spreading of KRAB-mediated silencing as a dynamic process that depends on the time and strength of recruitment as well as the level of activation at the target locus, which can be influenced by neighboring promoters and insulators.”

2. Lines 560-561: "By monitoring gene reactivation dynamics after release of the CRs, we conclude that activation can also spread to nearby genes in a distance-dependent manner." Only one distance (5Kb) was tested, which pales with the thousands of distances evaluated in some previously published studies.

Increasing the distance between a target gene and the KRAB recruitment site has been shown to decrease the susceptibility of a gene to be silenced (Groner et al. 2010), which we have added in the silencing portion (see section in introduction and discussion). However, here we are specifically referring to reactivation of genes after release of the silencing CR (that is, in the epigenetic memory phase). To our knowledge there have not been studies looking at the effect of distance and the cHS4 insulator on reactivation.

We have also emphasized this aspect of reactivation in the introduction, line 40-42, edit sentence:

“ To understand the effects of intergenic distance and insulators on spreading, we studied the dynamics of spreading of gene silencing and reactivation after recruitment and release, respectively, of different types of CRs to a dual-gene reporter (Figure 1A-B).”

3. Lines 571-572: "They suggest that genes in close proximity (<5-10kb) can respond to signals in a coordinated manner, during both silencing and activation". Right, although confirmatory of a large series of observations previously made on co-regulated gene clusters.

We have rephrased this sentence to emphasize that this coordinated response, which has been studied on co-regulated gene clusters endogenously is also seen in synthetic dualgene constructs:

“They suggest that genes in close proximity (<5-10kb) in a synthetic system can respond changes in chromatin regulators recruited at the promoter of one gene in a coordinated manner, during both silencing and activation, similar to endogenous gene clusters where neighboring genes tend to be co-regulated,including cell cycle, circadian rhythm and housekeeping genes (Cho et al. 1998; Cohen et al. 2000; Boutanaev et al. 2002; Ueda et al. 2002; Lercher et al. 2002).”

We have also edited the first sentence of our abstract to:

“In mammalian cells genes that are in close proximity can be transcriptionally coupled: silencing or activating one gene can affect its neighbors.”

We have also added previous observations about co-regulated ZNF-KRAB clusters, line 524:

“… as may be the case in regulation of zinc finger gene clusters which are auto- and coregulated by KAP1 recruitment (O'Geen et al. 2007)”

4. Lines 579- 582: "Finally, this experimental and theoretical framework can serve as a starting point for measuring and modeling the effects of targeting various epigenetic editors at endogenous loci in order to guide the time necessary for efficient on-target effects without unwanted off-target spreading". The authors fail to demonstrate convincingly that their system is really suited for the kind of high throughput analyses that would be necessary to consolidate, give a general value or even validate some of their conclusions.

We agree that future work on systematically studying dynamics at endogenous loci would require a different experimental system, such as dCas9. We have edited this sentence accordingly: “Finally, our experimental results and theoretical framework highlight the need for further measuring and modeling the dynamics of gene expression after targeting various epigenetic editors at endogenous loci systematically, (for example, with KRAB-based tools such as CRISPRi) in order to determine the time necessary for efficient non-target effects without unwanted off-target spreading.”

5. The authors should integrate in their thinking results reported ten years ago (BMC Genomics 12:378, 2011). This work proceeded to an extensive analysis of chromatin features conducive or not of KRAB-induced long-range silencing, making interesting conclusions on the "repression-primed" state of the most sensitive promoters and their paradoxical location in a surrounding of highly expressed genes, and on the failure of CTCF recruitment to exert any barrier effect on H3K9me3 spreading.

We have integrated the findings from this work into our discussion:

– Line 520, added text: “Additionally, in our system, we see that silencing is slower and occasionally inhibited by the presence of the cHS4 insulator, which contains high levels of acetylation. In our model, we propose that genes in active regions can also cooperate with neighboring genes in addition to insulators to oppose silencing by chromatin regulators. However, there are other factors at play in targeting of endogenous loci which affect gene silencing. In previous studies, targeting dCas9-KRAB to hundreds of repeated sgRNA sites forms a large domain of H3K9me3 heterochromatin on the order of megabases in a few days, but does not result in widespread gene silencing, rather the silencing of genes is controlled by the loss of H3K27Ac and H3K4me3 (Feng et al. 2020). Additionally previous studies have also shown that while genes that are more susceptible to KRAB-mediated silencing with KAP1 have higher levels of repressive histone marks at the promoter and gene body, though surprisingly they have higher levels of active histone marks surrounding the gene (Meylan et al. 2011), indicating that the genomic locus affects silencing by KRAB. Therefore, more work needs to be done to understand how the genomic and epigenetic environment surrounding a group of genes can affect their repressibility.

6. The authors should also comment on KRAB zinc finger protein genes, many of which recruit KAP1 at their 3' end and are grouped in tightly packed clusters, yet can escape self- or mutually inflicted silencing. And on the interesting observation by K. Helin's group that the H3K9me1/2 demethylase PHF8 is recruited to the promoter of many of these genes, perhaps explaining this phenomenon (Mol Cell 2010 38: 165).

We have added this to the discussion by editing sentence on line 521-523:

Original: “First, in mammalian genetic circuits where multiple genes need to be integrated in the same cell, genes controlled by KRAB need to be placed far away from other genes to avoid unwanted interference and feedback.”

New: “First, in mammalian genetic circuits where multiple genes need to be integrated together in a cell, genes controlled by KRAB need to either be placed far away from other genes to avoid unwanted interference and feedback, or other mechanisms of escaping silencing potentially through recruitment of H3K9me demethylases such as PHF8 should be considered (Kleine-Kohlbrecher et al. 2010).”

Reviewer #3 (Recommendations for the authors):As noted in the public comments, the authors should include a more critical discussion of the differences of this model with prior studies where KRAB domains were recruited to natural genomic sequences.

We agree with the suggestion to discuss in more detail KRAB recruitment to natural genomic sequences. We have added in the main text (line 526):

– Previous work has shown that dCas9-KRAB targeting of promoters or enhancers results in silencing of the associated gene along with enrichment of H3K9me3 at the targeted locus (Kearns et al. 2015; Thakore et al. 2015; Klann et al. 2017; O'Geen et al. 2017; Das and Chadwick 2021). Targeting of promoters of endogenous genes appears to result in spreading of H3K9me3 across the gene body (Klann et al. 2017), while targeting enhancers can result in either H3K9me3 only at the enhancer (Thakore et al. 2015) or only at the target gene (Kearns et al. 2015). However, we are not aware of ChIP-seq data after dCas-KRAB recruitment to an endogenous promoter to determine the extent of spreading of H3K9me3 beyond the gene body.

– We have also added more details regarding how our model differs from previous KRAB studies

Line 571-572, original: “They suggest that genes in close proximity (<5-10kb) can respond to signals in a coordinated manner, during both silencing and activation...”

New: “They suggest that genes in close proximity (<5-10kb) in a synthetic gene system can respond to changes in chromatin regulators recruited at the promoter of one gene in a coordinated manner, during both silencing and activation…”

The potential roles of histone acetylation functioning as insulator sequences require more rigorous testing and validation. Perhaps the author should mine some published genomic data to find potential correlations between various histone acetylation marks and heterochromatin vs euchromatin compartments in development or disease conditions.

We agree that using genomic data to connect histone modifications with insulators which help protect euchromatin from heterochromatin is a good approach. A recent paper has done this and used ENCODE ChIP-seq data to identify insulators, and found candidates which were then screened for their ability to prevent transgene silencing. They additionally show that when tested in the presence of an HDAC inhibitor, the expression of the insulated transgene increased. We have added this in our discussion along with more references to previous studies showing that the HS4 insulator’s capabilities are associated with acetylation and previous models that have characterized the role of insulators as activating forces which combat repressive forces:

– Line 547-549, add to sentence: “We can explain the function of insulators on spreading and background silencing by thinking of them as areas in an active chromatin state, that can for example recruit additional writers of acetylation (Zhao and Dean 2004), and other properties similar to promoters(Raab and Kamakaka 2010).”

– Line 549-550, add: “It has been shown that the core region of the HS4 insulator has high levels of acetylation (Mutskov et al. 2002; Zhao and Dean 2004), and it has been previously proposed that the HS4 insulator protects from spreading of H3K9me3 through maintenance of high levels of acetylation (Litt et al. 2001; Mutskov et al. 2002; Ghirlando et al. 2012). A recent study (Rudina and Smolke 2019) used ChIP-seq ENCODE data of histone modifications, including the presence of active acetylation marks and absence of repressive methylation marks, to identify candidate insulator sequences which were then screened for their ability to prevent background transgene silencing. Additionally, when these insulators were tested in the presence of an HDAC inhibitor, the expression of the insulated transgene increased (Rudina and Smolke 2019).

– Line 550, new paragraph: “Previous models have proposed the idea that insulators, similar to enhancers and transcription factors, are anti-silencers which oppose repressive forces (Fourel et al. 2004), and transcriptional activators have been shown to insulate against heterochromatin and even drive reactivation (Sutter et al. 2003). Our findings are in line with these models, and we further propose that active regions, such as insulators and promoters, cooperate with each other to inhibit gene silencing and drive gene reactivation.”

We have also added text to reference previous work that has shown strong correlations between acetylation and euchromatin, line 127:

– “Previous studies looking at histone modifications across the genome in different cell types have shown that active promoters and strong enhancers in euchromatin have high levels of acetylation (Ernst et al. 2011), which is in agreement with the general histone code by which acetylation confers open and active regions of euchromatin, while methylation results in silenced and compact heterochromatin (Jenuwein and Allis 2001).”

We have added information about facioscapulohumeral dystrophy (FSHD), which is a disease that is potentially caused by loss of spreading of heterochromatin causing aberrant gene activation:

– Line 568, add to sentence: “and perhaps in understanding aging where loss of heterochromatin can lead to changes in gene expression (Villeponteau 1997) or in diseases such as facioscapulohumeral dystrophy (FSHD). FSHD is a neuromuscular disorder that is potentially caused by loss of spreading of heterochromatin causing aberrant gene activation (Kleinjan and van Heyningen 1998; Gabellini et al. 2002; Hahn et al. 2010).”